# Identifying the topographic signature of early Martian oceans

Abdallah S. Zaki[1,2 ✉] & Michael P. Lamb[1]

Planet-wide interpretations of shorelines suggest that Mars once hosted an early ocean covering one-third of its surface[1–9]. However, the elevations of these shorelines deviate from an equipotential surface by several kilometres, challenging that interpretation[3,7,10–12]. Here we investigate whether a planet that once hosted an ocean should be expected to leave discernible shorelines. We show that on Earth, the most prominent topographic signature of a global ocean is not a shoreline. Rather, it is a band of low slope and curvature values that comprises coastal plains and the continental shelf, with an elevation range of −410 m to −15 m. When applying a similar analysis to the Martian surface, we observe a comparably flat zone between approximately −1,800 m and −3,800 m elevation, potentially marking a partially preserved Martian coastal shelf. Although other processes, such as lava flows[13], might explain flat regions locally, a coastal shelf best explains the circumglobal band of flat topography, in addition to river delta deposits[4,14–17], coastal deposits[18], thick sequences of layered rock[19,20] and aqueously altered minerals[20,21], all observed within the putative coastal shelf zone. Our results support the presence of an ancient ocean on Mars and indicate that topographic shelves rather than shorelines may be better indicators of long-lived oceans.

The northern plains of Mars feature distinct geological boundaries between the southern highlands and northern lowlands extending for thousands of kilometres, which have been used to infer ancient ocean shorelines[1–9]. These are known as the Arabia (contact 1) and Deuteronilus (contact 2) shorelines[3,7]. However, several kilometres of deviation in the elevation of these proposed shorelines from equipotential surfaces called into question the interpretation of these features as shorelines and, consequently, the presence of a vast ocean covering one-third of the Martian surface[3,7,10]. Two explanations have been proposed to explain this long-wavelength deviation in putative shoreline elevation. The first is true polar wander, which formed a new equatorial bulge and redistributed mass from the palaeoequatorial bulge to the new equator, thereby deforming the surface topography and palaeoshorelines following Tharsis volcanic growth[11]. The second explanation is the Tharsis-induced deformation model, which proposes that the uplift from the formation of Tharsis altered the shape of Mars and geoid, leading to the deformation of the shorelines[12]. Moreover, misidentification of shoreline features or resurfacing due to tsunamis or lava flows has been raised as possibilities[10,13,22,23]. However, none of these explanations fully accounts for the global elevation deviations.

Previous work reconstructed Martian oceans by mapping topographic breaks[2,3,7,22], assuming they represent shorelines, in line with palaeolake reconstructions on Earth. However, reconstructing even geologically recent palaeolake levels based on putative shorelines is challenging because of erosion and deformation[24–28]. Moreover, it is unclear if a shoreline is the topographic fingerprint of a long-lived ocean. Rather than a distinct shoreline imprint, the main topographic feature of the modern ocean is the band of low-gradient terrain that bounds most continents at an elevation between tens and a few hundreds of metres below the sea level, and it comprises coastal plains and the continental shelf[29,30].

The origin of continental shelves is debated, and multiple erosion and deposition processes probably contribute to their formation. The processes include (1) fluvial deposition to build low-gradient deltaic and coastal plains, and the extension and contraction of these plains during sea-level fluctuations[31–34]; (2) wave bevelling to create a low-gradient platform and efficient offshore transport of sediment at depths above wave base through wave-supported suspensions[33,35]; and (3) the formation of buoyant freshwater river plumes over saline ocean water that move sediment long distances along the coast because of geostrophic steering[36]. On Earth, plate tectonics is responsible for differentiating the continental and oceanic lithosphere that has created continental margin relief[37,38] and also for developing the faulted basement platform during rift extension on which the continental shelf on passive margins is built[37,39]. Active tectonics can also influence shelf morphology through faulting and folding[37]. Despite the lack of plate tectonics on Mars, the erosion and deposition processes that are the primary controls on shelf morphology on Earth may also have been active on early Mars. The question arises as to whether Mars has a coastal shelf.

To address this question, we analysed the global topography of Earth to identify the characteristic signature of the modern ocean (see flowchart in Supplementary Fig. 1). First, we used fluvial landforms (major rivers and global deltas)[40,41] to establish the upper bound of elevation for the coastal plains. Next, we used a map of ocean geomorphic

[1]Division of Geological and Planetary Sciences, California Institute of Technology, Pasadena, CA, USA. [2]Department of Earth and Planetary Sciences, Jackson School of Geosciences, The University of Texas at Austin, Austin, TX, USA. ✉e-mail: Abdallah.zaki@jsg.utexas.edu

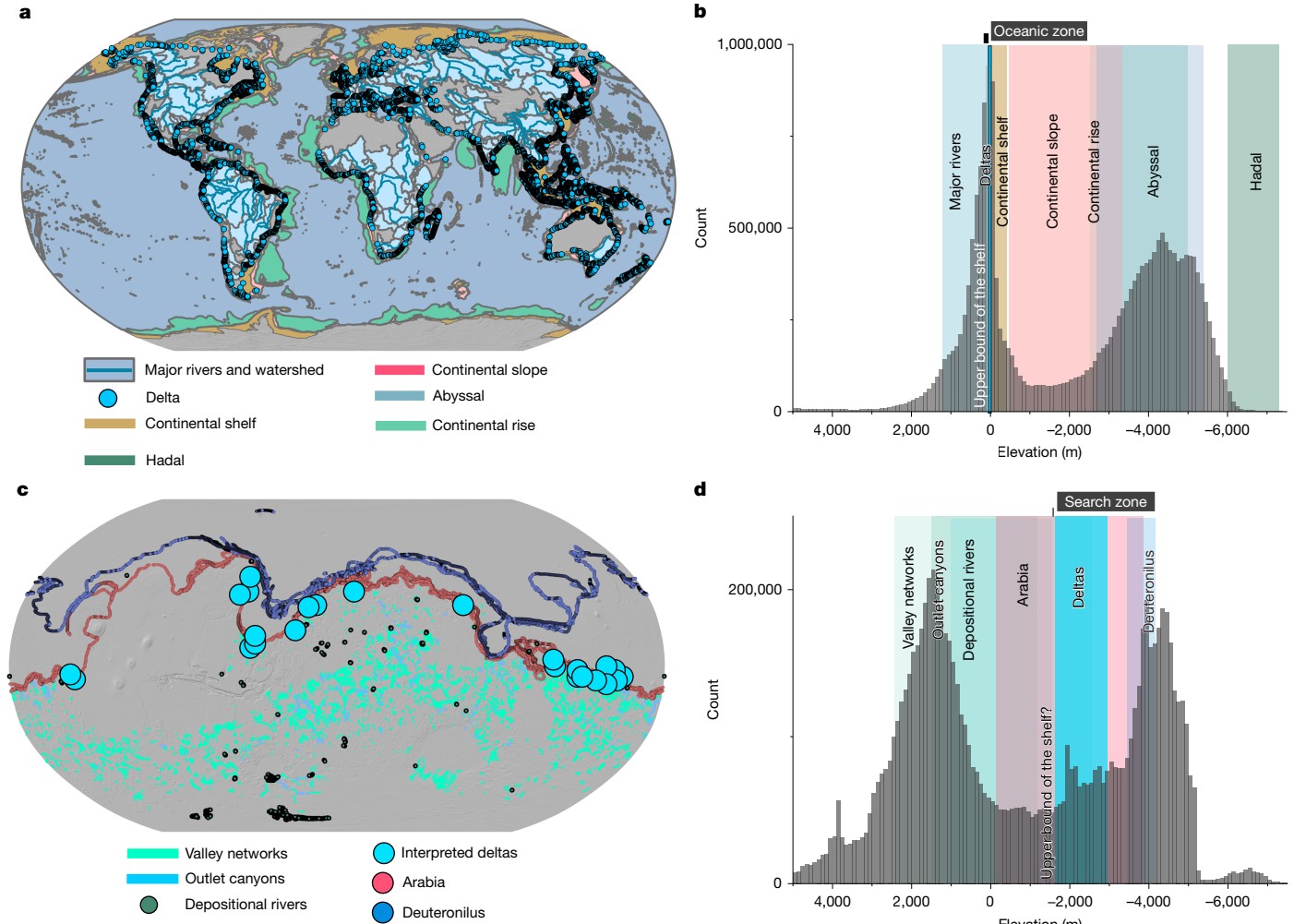

**Fig. 1 | Geographic distribution and elevation histograms of the surfaces of Earth and Mars, along with their water-formed landscapes, illustrating that deltas can serve as proxies for defining the upper boundary of the Martian shelf. a,b**, The distribution and elevation of land and sea landforms on Earth. **c,d**, A comparable sequence of water-formed landscapes on Mars. The water-formed landscapes on Earth include major rivers (195,022 data points) and deltas (10,848 data points)[40,41], followed by oceanic landforms[41], which consist of the continental shelf (14,820,634 data points), continental slope (7,606,463 data points), continental rise (12,144,045 data points), abyssal zone (116,749,407 data points) and hadal zone (1,238,491 data points). On Mars, a similar sequence includes valley networks (3,294,322 data points)[44], outlet canyons (248,865 data points)[44], depositional rivers (16,515 data points)[45], interpreted deltas (48 data points)[4,10,14,17]—single-lobate to stacked deltaic systems, which in some cases are connected to interpreted submarine-channel belts, as in Aeolis Dorsa[17]—and the proposed shorelines: Arabia (10,192 data points) and Deuteronilus (42,900 data

points)[10,22]. The elevation distributions of both landforms suggest that the transition from land to sea on Earth occurs across the continental shelf, which is characterized by shorter-wavelength topography compared with Mars. Furthermore, the data indicate that a similar transitional zone from terrestrial to oceanic environments on Earth can be identified within the first 2.5 km below sea level, representing the upper 90th percentile of the continental rise. Vertical bars in different colours highlight the 10–90% range of data points. These data points were calculated using the Zonal Statistics tool for polygons and by generating points along the polylines of these morphologies (Methods). The elevation was derived from both the ETOPO Global Relief Model for Earth and the global Mars Orbiter Laser Altimeter (MOLA) dataset for Mars (Methods). The histograms in **b** and **d** show the elevation distributions of the Earth and Mars surfaces, with 100-m binning, both exhibiting bimodal distributions. The map backgrounds are hillshade images derived from the respective topographic datasets.

features (including continental shelf, continental slope, continental rise, abyssal plain and hadal zone)[42] to set the lower limit of elevation for the shelf. We then calculated slope and curvature as topographic metrics to quantitatively define the coastal shelf. Next, we performed the same analysis on Mars, focusing on a search zone extending 2.5 km below the elevation of mapped river deltas, as river deltas build on coastal shelves. Our results support the presence of an ancient ocean on the northern plains of Mars that was bounded by a coastal shelf.

## Landforms that bound the coastal shelf

The transition between terrestrial and marine landforms is well-defined by river termini and delta deposits, which collectively establish the

upper-bound elevations of continental shelves (Fig. 1a,b). Mars also hosts abundant river and delta deposits (Fig. 1c), which could similarly mark the upper elevation of the Martian shelf. Although the lower boundary on Earth is defined by the shelf break and continental slope—landforms not present on Mars—we rely on other marine landforms from Earth to guide our search for the Martian shelf zone.

On Earth, there is a spike in the global elevation distribution of terrestrial and marine landforms near sea level (Fig. 1b). This clustering reflects erosion and deposition processes. Sediment is eroded from upland areas and transported to the coasts by rivers, with the 10th to 90th percentiles of global river elevations ranging from 19 m to 1,194 m (Fig. 1a,b). As river flow slows, much of the coarse sediment load is

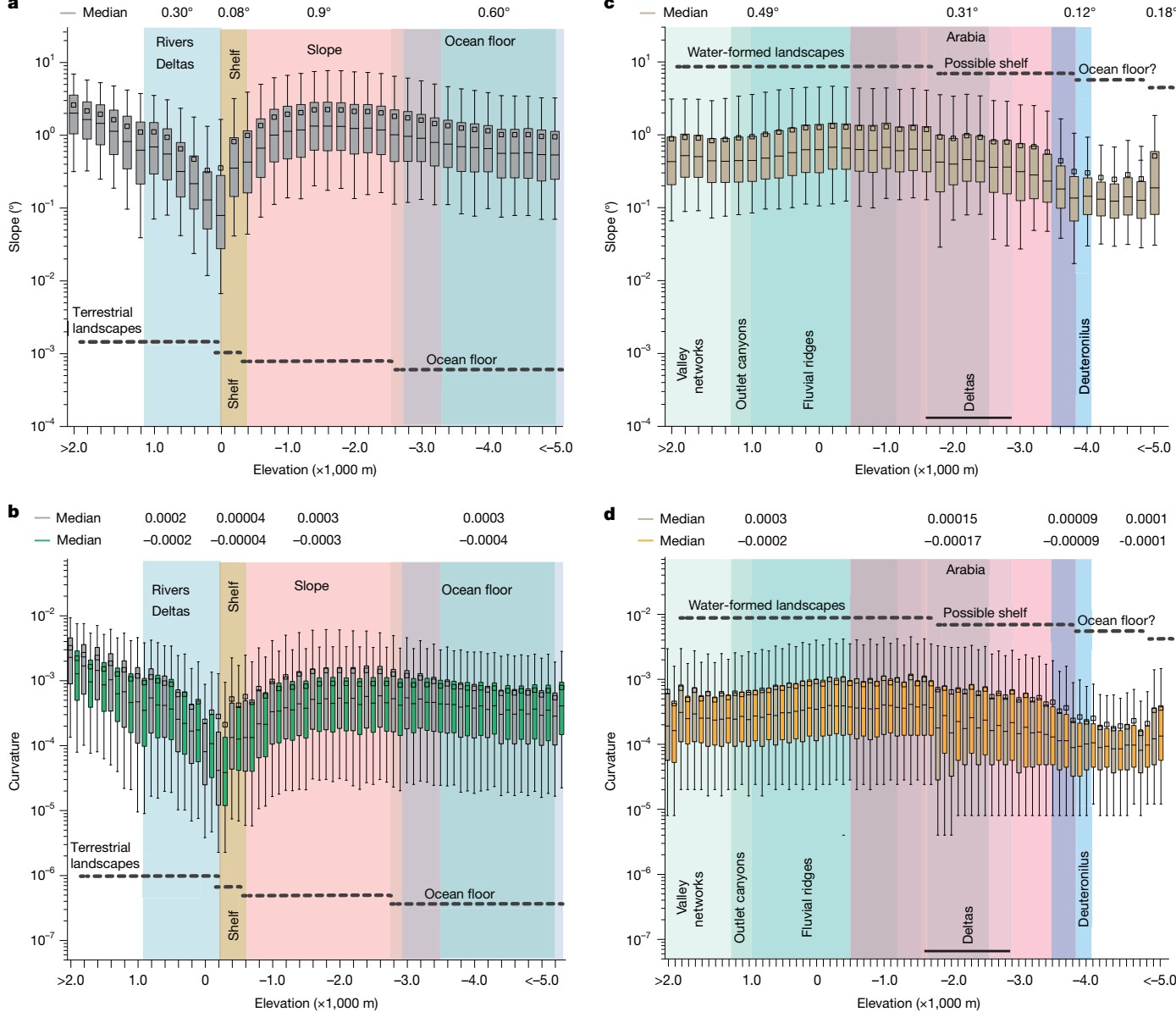

**Fig. 2 | Slope and curvature of the entire surfaces of Earth and Mars. a,b,** Slope (**a**) and curvature (**b**) of Earth. **c,d,** Slope (**c**) and curvature (**d**) of Mars. There is a clear decline in both topographic parameters towards the continental shelf on Earth, particularly within the first 200 m of elevation. By contrast, Mars exhibits a different pattern, with two distinct zones marked by declines in both slope and curvature: one between −1,800 m and −3,800 m, and another between −3,800 m and −5,000 m. The presence of fluvial ridges, deltas and two shorelines within the first zone (−1,800 m to −3,800 m) suggests that this is probably a transitional zone between landscape and seascape. The data were computed from the ETOPO

Global Relief Model for Earth and the MOLA data for Mars (see Methods for details). Uncertainties correspond to the 5–95% confidence interval for the median. The vertical bars in the background represent the 10th–90th percentile elevation range of the identified landforms, as shown in Fig. 1. Curvature values are reported in one hundredth (1/100) of the DEM $z$ unit (here, metres). As curvature includes both positive and negative values, we multiplied the negative values by −1 to improve interpretability, especially when comparing concave and convex surfaces (with negative values colour-coded in green for Earth and orange for Mars).

deposited, forming deltas with elevations between −11 m and 26 m (Fig. 1a,b). Modern deltas are built on the continental shelf, which has elevations that range from −410 m to −15 m (10th–90th percentile). Although the shelf spans a wide range of elevations, the median elevation is −87 m, aligning closely with the 120-m global sea-level drop during the last glacial cycle (the past 120,000 years)[43]. Beyond the shelf, the seabed transitions to the steeper continental slope (median = −1,292 m below sea level) and further to the ocean floor, including the continental rise (median = −4,010), abyssal plain (median = −4,273) and hadal zones (median = −6,149), in which elevations cluster around a median of −6,000 m. These analyses confirm that the elevation of deltas can define the upper bound of the shelf.

Although we do not have detailed mapping of potential oceanic landforms on Mars comparable to those on Earth, we extracted elevation data from a sequence of water-formed landforms (Fig. 1c,d) to define the upper bound of the search zone for the Martian shelf[4,10,14,17,22,44,45]. Similar to Earth, Mars hosts valley networks and outlet canyons—formed by lake-breach floods—that eroded and transported sediments from upstream sources (Fig. 1d). The 10th–90th percentiles of global elevation data for valley networks range from −1,177 m to 2,424 m, whereas the outlet canyons span from −1,530 m to 1,010 m (Fig. 1d). Depositional rivers are expressed as fluvial ridges (ridges preserve fluvial deposits)[45], which terminate abruptly and patchily at −2,577 m, marking the 90th percentile upper limit. Open-basin deltas along the

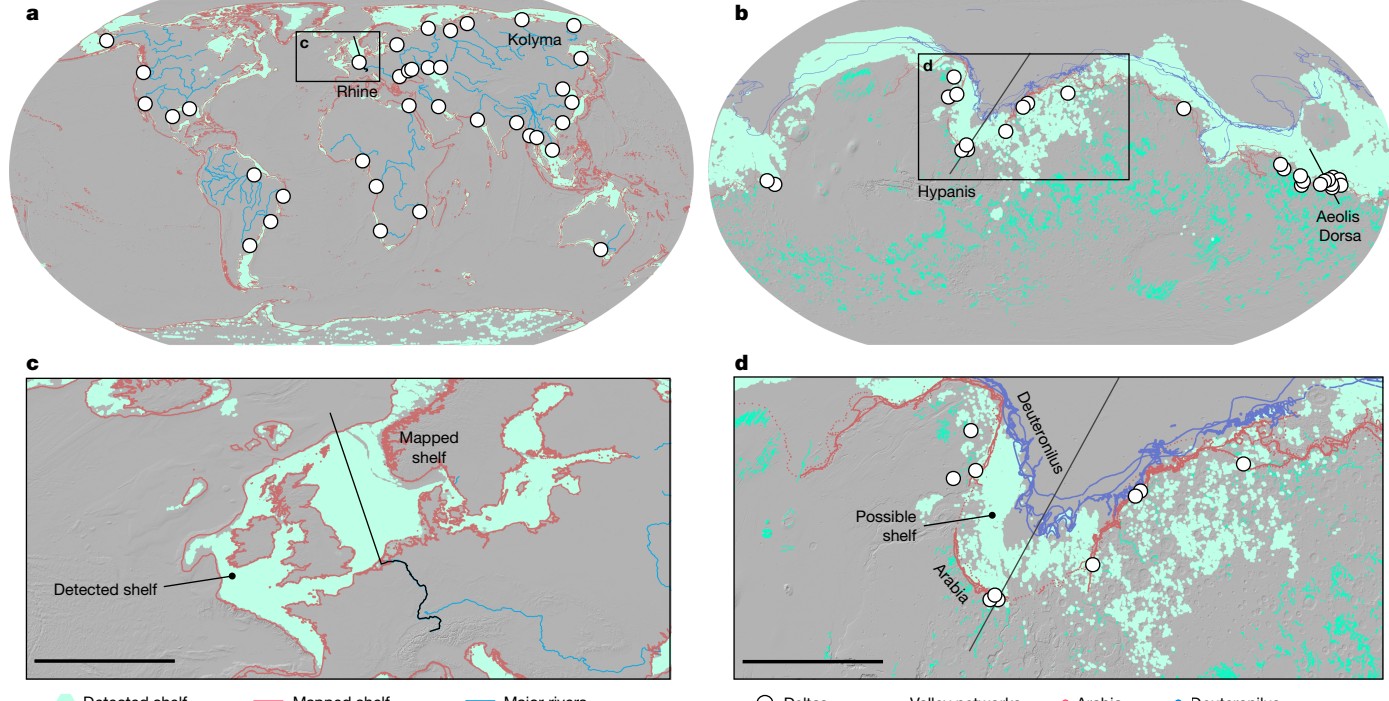

**Fig. 3 | Shelf detection using the Geomorphons function. a,b,** Shelf detection using the Geomorphons function on Earth (**a**) and Mars (**b**), alongside mapped shorelines on Mars (**b**) and mapped shelves on Earth (**a**). Flat surfaces on Mars and Earth were identified using a flat terrain angle threshold of 0.31° (Methods), selected as it represents the median slope of the Martian zone (−1,800 m to −3,800 m), which includes shorelines[10,22], interpreted deltas[4,10,14,17] and distinct flat surfaces. These panels show the comparison between the detected shelf on Earth and the mapped shelf[42], as well as the potential Martian shelf (−1,800 m to −3,800 m) and the proposed shorelines[10,22]. **c,d,** Magnified sections highlighting the mapped and detected shelves as well as the proposed shorelines. The black lines in both sections represent the Rhine River and the Hypanis Valles, each extending continuously into the deep ocean. The corresponding profiles are shown in Fig. 4. The pixels located above the putative Arabia shoreline share similar topographic characteristics with the classified shelf terrain. These areas may represent earlier shoreline positions formed during highstands or transgressions, or degraded or eroded surfaces that preserve similar slope signatures. Scale bars, 1,300 km (**c**); 1,500 km (**d**).

northern crustal dichotomy boundary range in elevation from −1,638 m to −2,880 m (refs. 4,10,14,17). These landforms record an open-system, ocean-margin depositional system of both single lobate and stacked deltas[4,10,14,17]. In Aeolis Dorsa specifically, the stacked deltaic deposits grade basinwards into the Aeolis Serpens submarine-channel belt[17] (Supplementary Table 1). The deltaic features are followed by two topographic contacts, Arabia and Deuteronilus, which have been interpreted as oceanic shorelines with median elevations ranging from −2,479 m to −3,798 m. From these analyses, we concluded that −1,638 m (the upper elevation of the deltas) can serve as a reference for setting the upper bound in the search for the Martian shelf.

## Topographic metrics for detecting a coastal shelf

Continental shelves on Earth have been defined regionally using diverse criteria and expert knowledge and compiled into a global map[42] (Fig. 1a). Here, we sought quantitative topographic metrics that can be used to distinguish the continental shelves on Earth, so that the same metrics might be applied to Mars. On Earth, we observed that all land above sea level globally has a median slope of approximately 0.3°, measured at 5-km pixel scale, which then declines by nearly fourfold to 0.08° within the first 200 m below sea level, representing the continental shelf (Fig. 2a). Below that, the continental slope is approximately 11 times steeper than the continental shelf (0.9°; Fig. 2a). The slope decreases to 0.6° when moving to the ocean floor (Fig. 2a). The results of topographic patterns are similar when slope is measured at scales of 2.5-km pixel scale and 10-km pixel scale (Methods and Supplementary Fig. 2). Similar to topographic slope, the continental shelf also is a minimum in topographic curvature measured at all three scales (Fig. 2b). At elevations higher than the shelf and lower than the shelf, both concavity (negative curvature) and convexity (positive curvature) values are larger (Fig. 2b). These analyses show that the dominant topographic signature of the global ocean is the continental shelf, and it is characterized by a distinct minimum in both global slope (0.08°) and global curvature (0.00004, −0.00004).

The same analysis of the topography of Mars does not reveal a single distinct minimum in global slope and curvature, as observed on Earth (Fig. 2c,d and Supplementary Fig. 3). Instead, two coupled minima in slope and curvature are observed on Mars (Extended Data Fig. 1), supported by a Kruskal–Wallis $H$ test ($H(2) = 27.50$; $P = 1.07 \times 10^{-6}$), indicating significant differences in median slope values among the observed elevation ranges. The first minima lie between approximately −1,800 m and −3,800 m (slope = 0.31°, curvature = 0.00015, −0.00017), and the second between −3,800 m and −5,000 m (slope = 0.12°, curvature = 0.00009, −0.00009). The minima are less distinct than on Earth, but this might be expected, given that the ocean of Mars, if it existed, has probably been dry for billions of years, with substantial topographic modification driven by wind erosion[46], meteor impacts[47], lava flows[13], Hesperian–Amazonian age outflow channels[48] and regolith creep[49]. These geomorphic processes can smooth[13,46–49], incise or infill the margin and thus broaden and mute the slope and curvature minima without eliminating an underlying shelf–slope break. Long-wavelength vertical motions associated with geophysical processes—such as isostatic or flexural adjustment to ocean loading and unloading, true polar wander or Tharsis-induced deformation[11,12]—would act over scales much larger than the mapped shelf width, primarily tilting or gently warping the margin and changing absolute slopes, but are unlikely to remove a relative slope and curvature minimum between shelf and slope.

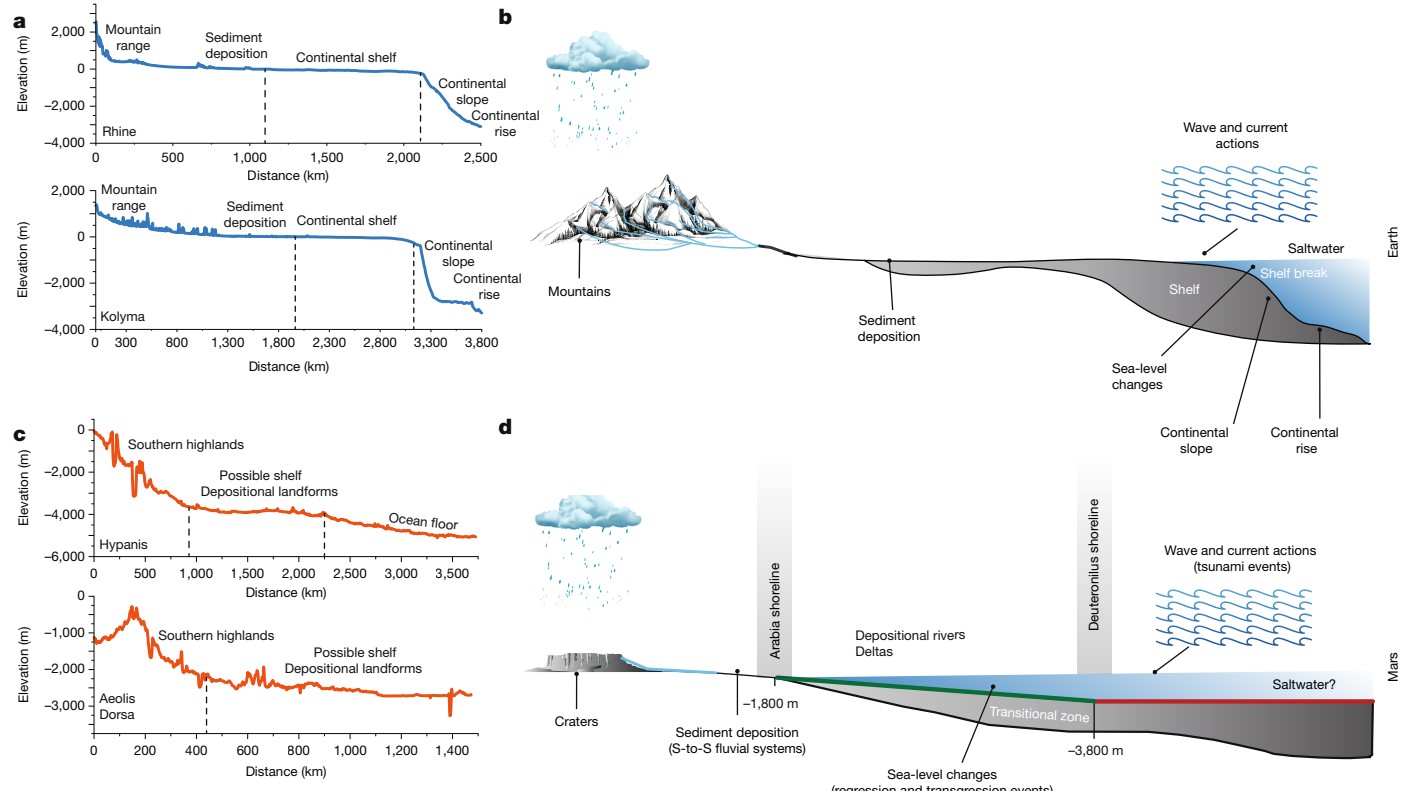

**Fig. 4 | Earth and Mars landscape-to-seascape transition zone and the mechanisms shaping this zone. a–d**, Data from four terrestrial and Martian fluvial systems of relatively comparable size (**a**,**c**) are coupled with schematics to show the morphologies and mechanisms associated with the land-to-sea transition shelf on Earth, compared with a similar possible zone on Mars (**b**,**d**). **a**,**b**, Elevation data from the Rhine River and the Kolyma River show how these systems descend from mountain ranges to their floodplains and eventually extend across up to about 1,000-km-wide continental shelves, which incline towards the continental slope and subsequently the continental rise. **c**,**d**, Elevation data from the southern highlands and their connection to the northern lowlands in the largest two deltaic systems, Hypanis and Aeolis Dorsa, which highlight the relatively flat proposed shelf zones transitioning into a flatter probable ocean floor. The schematics (**b**,**d**) show how fluvial systems evolve along mountains and crater rims before expanding across the shelves towards the ocean floor. The main mechanisms involved in forming shelves on Earth include wave and current action, sea-level changes, river deposition and saltwater influence. Similar mechanisms are suggested to have occurred on Mars, particularly sea-level changes, deposition during sea-level fluctuations, tsunami events and possible wave actions.

The processes creating basin relief also differ between the two planets. On Mars, an impact crater is probably the primary cause for the northern lowlands[50], whereas on Earth, plate tectonics allowed for chemical differentiation between oceanic and continental crust[51,52].

The question is whether these topographic features in Fig. 2 could represent the imprint of an ancient coastal shelf on Mars. In the context of available geologic evidence, the location of a coastal shelf between −1,800 m and −3,800 m is plausible. For example, geologic indicators of rivers—valley networks, outlet canyons and fluvial ridges (river deposits)—are mostly found at elevations above the putative coastal shelf, as expected (Figs. 1c,d and 2c,d). Likewise, mapped deltas, in some cases connected to interpreted submarine-channel belts and shorelines, lie within the elevation range of the proposed shelf (Figs. 1c,d and 2c,d).

To map the shelf using our topographic criteria, we used a land classification algorithm, 'Geomorphons'[53,54], and specifically its terrain class 'flats' that identifies contiguous areas of low gradient. We ran this algorithm at two grid resolutions (1.6 km and 5 km) to evaluate how resolution influences feature detection. Through 40 iterations of trial and error (Methods, Extended Data Figs. 2–4 and Supplementary Table 2), we found that using an upper limit of 0.31° at a grid resolution of 1.6 km—the median slope of the zone we propose to represent a coastal shelf on Mars—resulted in the correct identification of about 71% (around 1.6 km resolution) and about 69% (around 5 km resolution) of the continental shelf of Earth (Fig. 3a,c and Extended Data Fig. 4) when compared with previously compiled global mapping[42]. Applying this approach to Mars at a 5-km grid resolution, our automated landscape classification identified 10.2 million km$^2$—approximately 7% of the Martian surface—as a potential coastal shelf at elevations between −1,800 m and −3,800 m with slopes ≤0.31°, clustered along the crustal dichotomy boundary (Fig. 3b). This zone, which we interpret to be an approximate boundary of the coastal shelf of Mars, comprises most of the previously interpreted deltas, as well as the Arabia interpreted shoreline and a substantial portion of the Deuteronilus interpreted shoreline (Fig. 3b,d).

## The case for a coastal shelf on Mars

We investigated whether a global ocean would leave a topographic footprint at the planetary scale. Our global topographic analysis of elevation, slope and curvature shows that the transition from land to sea on Earth is not limited to a single distinct shoreline, but also includes a broader zone of low slope and curvature near the sea level. This morphometric transition complements, rather than replaces, the other landforms associated with coastlines, such as shorelines and deltas. We presented evidence for a similar topographic zone on Mars, which is bounded at high elevations by delta and river deposits, and encompasses previously proposed shorelines. Based on this evidence, we propose that Mars has a coastal shelf that once bounded a global ocean and is characterized by a band of low topographic curvature

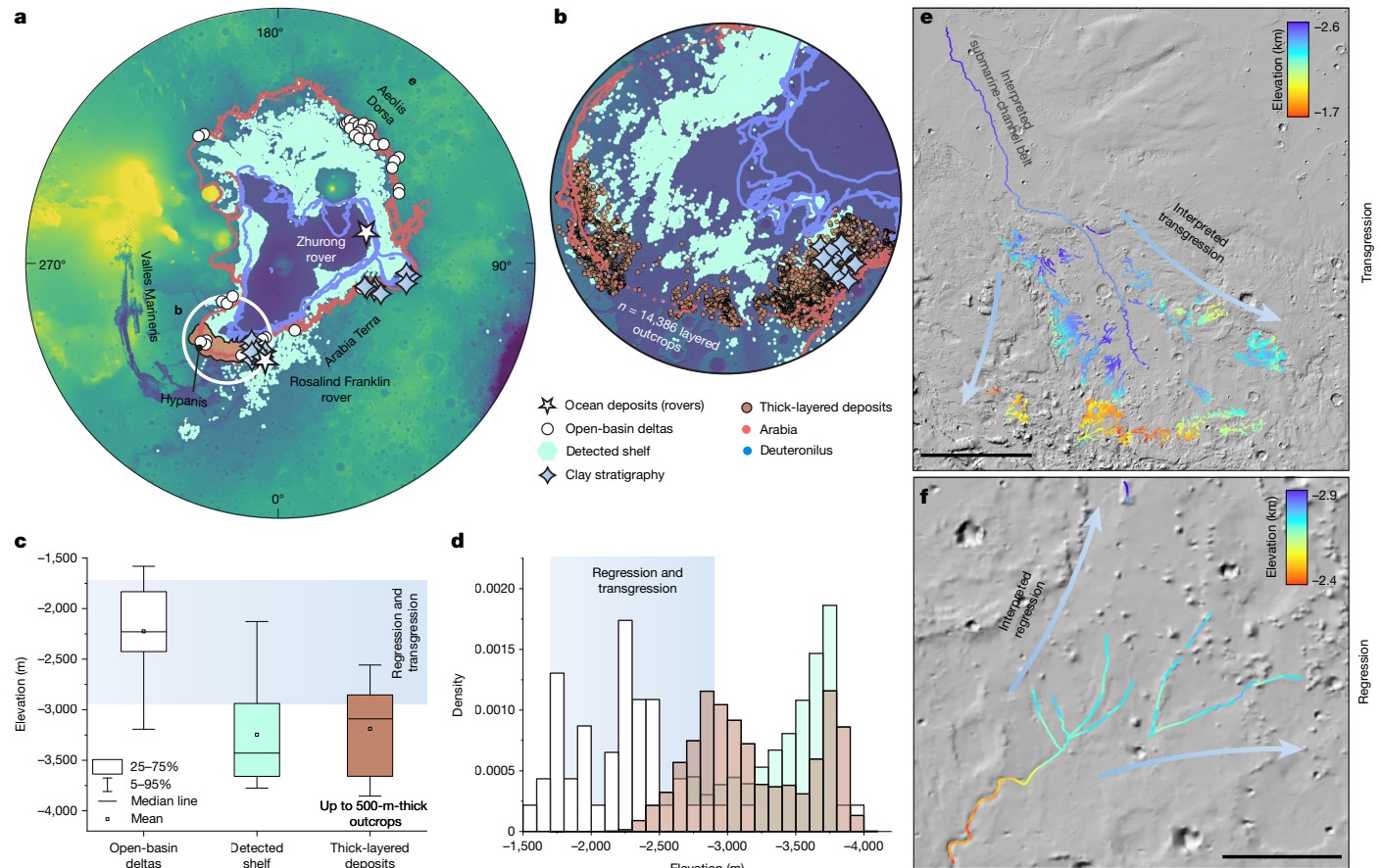

**Fig. 5 | Sedimentologic and mineralogic evidence for sedimentary rocks, clay stratigraphy and open-basin deltas along and within the detected shelf. a,b,** Polar stereographic projections of the detected shelf showing 14,386 mounds interpreted to have formed by dichotomy retreat, consisting of thick-layered deposits up to 500 m in southern Chryse Planitia, Mars, along with widespread clay-bearing stratigraphy and open-basin deltas[4,10,14,17,19–21]. **c,d,** Box plots (**c**) and histograms (**d**) of open-basin delta elevations show that most deltas are built atop the detected shelf and are consistent in elevation with the layered deposits. The blue gradient—horizontal in **c** and vertical in **d**—indicates the range of interpreted sea-level changes, based on evidence of transgression and regression in **e** and **f**. **e,f,** Evidence of transgression and regression from two large deltaic systems in Aeolis Dorsa[17] (**e**) and Hypanis[14] (**f**). The background map is adapted from MOLA. Scale bars, 300 km (**e**); 80 km (**f**).

and slope values within elevations between −1,800 m and −3,800 m, as shown in Fig. 3.

On Earth, the continental shelf typically occurs within the first few hundred metres below modern sea level, with 90% of its area located within 400 m depths. Deltas also tend to cluster within this zone. The range of shelf elevations reflects the processes that have shaped the shelf over time. Sea-level fluctuations, combined with river deposition during glacial cycles, have left a clear imprint on the continental shelf, producing shelf elevation ranges from tens of metres up to 134 m (refs. 29,35,43,55–57; Extended Data Fig. 5). Approximately 57% of the total shelf area falls within this water depth range (Extended Data Fig. 6). Similarly, storm reworking contributes to shelf morphology, bevelling shelves to depths of several tens of metres or even up to 100 m (ref. 36). These are the main mechanisms responsible for shaping much of the shelf. However, in some regions—such as the North Sea and the Antarctic—the shelf extends beyond 1,000 m below sea level because of glaciation and tectonic deformation[29] (Extended Data Fig. 6). These broader elevation ranges reflect the intersection of the shelf with incised canyons and fault systems associated with tectonic rifting and accretion[58], and deep glaciated zones such as those on the Antarctic and Arctic Ocean shelves[29].

Processes shaping terrestrial shelves could have been at work on Mars, including sediment deposition, sea-level fluctuation and wave bevelling (Fig. 4d, Extended Data Figs. 6–8 and Supplementary Movies 1 and 2). On Mars, open-basin deltas span a much wider range in elevation

(about 1.25 km) as compared with Earth, and the putative topographic shelf spans approximately 2 km in elevation. This wider range may arise because terrestrial deltas formed over the past approximately 10–100 kyr, whereas shelves are deformed and reformed over millions of years by plate tectonics. Without plate tectonics, Martian deltas and the coastal shelf may be influenced by more extreme sea level fluctuations over millions of years or more. Although the ocean of Mars dried completely at least once, evidence suggests that significant sea-level fluctuations occurred before this prolonged desiccation period. For instance, two large Martian deltaic systems in Aeolis Dorsa and Hypanis Valles record regressive and transgressive events in the range of 500–900 m (refs. 14,17; Extended Data Fig. 7), which is almost two to three times higher than the highest recorded sea level changes over the past 500 million years in the history of Earth (Extended Data Fig. 5) and four to eight times greater than the latest sea level fall that probably marks the recent continental shelf[43,55–57] (Extended Data Fig. 6). Apart from topographic deformation from Tharsis loading and true polar wander[11,12], the coastal shelf helps resolve the issue of elevation variations along shoreline traces (Fig. 3) by allowing for a surface that can amalgamate shoreline indicators over time.

Our shelf topographic reconstruction (Fig. 3) and the proposed mechanisms for its formation are supported by recent discoveries from the Zhurong rover of China[18], which is at present exploring Utopia Planitia about 150 km north of our mapped shelf. The rover detected 10–35 m of subsurface sediment with unidirectional dipping beds that

are inclined in the proposed seaward direction, similar to continental margin coastal deposits on Earth[35]. Moreover, more than 14,000 layered mounds of sedimentary rock up to 500 m thick (about 3.7 Ga and older), along with indicators of aqueous alteration and deltaic deposits recording marine regression and transgression[14,17,19–21], fall within the zone we identified as a shelf (Fig. 5). Further support comes from the recent identification of clay-bearing stratigraphy within the proposed coastal shelf zone[21] (Fig. 5a,b).

Continental margins are the main planetary sinks for sediment and carbon and account for much of the sedimentary record of Earth. This record has shaped our understanding of the evolution of climate, tectonics and life over geologic time[58,59]. By analogy, the proposed Martian coastal shelf represents an important target for future exploration, including the European Rosalind Franklin rover, scheduled to land in Oxia Planum in 2030. A rover investigating a coastal shelf might encounter deposits characteristic of sediment transport by ocean currents and waves, storm events and fallout from buoyant or plunging river plumes[60]. Some of these characteristics include prograding clinoforms that fine seaward, alternating beds of mudstone and sandstone event beds, combined flow ripple and hummocky cross stratification and turbidite sequences[61,62]. More broadly, our results indicate that long-lived ancient oceans on presently arid planets may be best identified not only through discrete shorelines but also through fluvio-deltaic landforms and broad bands of low-slope and low-curvature terrain that are part and parcel of a global coastal shelf.

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

## Methods

Our work relies on three datasets: (1) global digital elevation models and bathymetric data for both Earth and Mars; (2) maps of fluvial features on Earth (major global rivers and deltas) and Mars (valley networks, fluvial ridge systems, outlet canyons and interpreted deltas), along with maps of oceanic features on Earth (continental shelf, shelf break and ocean floor) and interpreted shorelines on Mars; and (3) results of elevation, slope, curvature and landscape classifications for each cell on both Earth and Mars. We describe the data and outline our methods for each dataset below and briefly explain the flowchart in Supplementary Fig. 1.

### Global digital elevation data and bathymetric data

**Earth.** Three global digital elevation models integrate both land and ocean surfaces at different resolutions and levels of consistency: (1) the ETOPO Global Relief Model with a general average resolution of about 1.85 km per pixel[63]; (2) the SRTM30_PLUS Estimated Topography with a resolution of around 1 km per pixel[64]; and (3) the General Bathymetric Chart of the Oceans (GEBCO) with a resolution of approximately 500 m (ref. 65). For our global-scale topographic analysis, we used the ETOPO1 Global Relief Model because of its consistent pixel resolution across both terrestrial and oceanic regions. ETOPO1 provides a uniform 1 arc-min resolution (about 1.85 km per pixel), integrating satellite altimetry, shipboard echo-sounding and terrestrial measurements into a cohesive dataset[63]. This consistency is important for studies requiring seamless data across different terrains, ensuring that both land and ocean topography are represented with the same level of detail. By contrast, the SRTM30_PLUS Global Bathymetry and Topography dataset, while offering higher resolution for land areas (30 arcsec), lacks uniformity as it focuses primarily on terrestrial regions and provides less detailed coverage for the oceans[64]. Moreover, the GEBCO dataset, although detailed for ocean bathymetry, does not offer the same consistent pixel resolution for land topography, leading to potential discrepancies when integrating land and ocean data[65]. Therefore, ETOPO1 was selected to ensure uniform resolution and comprehensive coverage across both terrestrial and oceanic environments, addressing the need for consistent pixel data in our analysis.

**Mars.** We used the global Mars Orbiter Laser Altimeter (MOLA) gridded topography, which offers a pixel resolution of 463 m per pixel[66]. This dataset is derived from more than 600 million measurements covering the entire Martian surface. These measurements were meticulously adjusted to ensure consistency, providing a uniform pixel resolution across the entire terrain of Mars[67,68].

### Data resampling

We resampled both digital elevation models to multiple resolutions—2.5 km, 5 km and 10 km—for several key reasons: (1) Resampling the topographic data of Earth and Mars to a uniform resolution is essential to apply uniform analytical methods and enable direct comparison of topographic features across both planetary surfaces. (2) These specific resolutions were selected to intentionally exclude fine-scale landforms on Mars, as these are generally younger in age[69]. Our study, however, focuses on older, broader-scale topographic features that provide insights into ancient surface processes. The resampling was conducted using the 'Resample' tool in ArcGIS[70,71], in which we applied the 'nearest neighbour' option to preserve the exact elevation values, minimizing significant interpolation and smoothing, and thereby ensuring the integrity of the original data[72,73].

### Maps of fluvial and oceanic features on both Earth and Mars

To identify a rough search zone on Mars for the transition from landscape to seascape, we used maps of the major world rivers and deltas on Earth[40,41]. These typically indicate where the terrestrial landscape ends and the oceanic zone begins. However, this assumption holds only if the rivers and deltas were active simultaneously. Changes in sea levels could alter this relationship, but the maps still provide a useful approximation of the extent of the transition zone. Moreover, we used an extensive dataset mapping seafloor geomorphic feature[42], which not only helps to define the zone but also offers insights into how oceanic geomorphic features evolve spatially. We focused on the key geomorphic features that define the transition: the continental shelf, the shelf-break slope, the continental slope, the continental rise and the key ocean floor landforms (abyssal and hadal zones).

We relied on global mapping of valley networks[44,74,75], outlet canyons[44], depositional rivers (fluvial ridges)[45] and interpreted deltas[4,10,14–17,76]. Furthermore, we used maps of topographic contacts, previously interpreted as shorelines[10,22,77], to assess how the water-formed landscape functioned.

Given the debate over whether Martian deltas formed in open or closed basin systems, we chose to compile the available delta datasets and then apply specific filtering criteria[4,10,14–17,76,78,79]. We selected deltas that (1) are open to downstream flow and located along the dichotomy boundary and/or (2) exhibit complex stacking patterns interpreted as evidence of formation within either regressive or transgressive depositional environments. This filtering resulted in a set of 48 deltas (Supplementary Table 1 and Supplementary Fig. 4). We further examined these deltas and classified them into two categories based on their morphology: single-lobate deltas and stacked deltaic systems. To further cross-validate our compilation, we calculated the elevations of all channels and lobes—including those preserved as ridges and interpreted as erosional remnants of ancient deltas—within the largest deltaic systems in Aeolis Dorsa and Hypanis[14–17], to capture elevation changes potentially associated with past sea-level fluctuations (Fig. 5e,f and Extended Data Fig. 7b,c).

### Setting a search zone for landscape-to-seascape transitions

We converted the shapefiles of terrestrial rivers, deltas and Martian rivers, deltas and proposed shorelines into points in ArcGIS Pro[70]. We then calculated the elevation of each point and plotted the results to examine where the continental zone transitions to the oceanic zone. For oceanic geomorphic features (polygons), we used the zonal statistics tool in ArcGIS Pro to calculate the number of pixels within each polygon, total area and the 10th–90th percentile elevation values for each zone.

We extracted elevation data for the major global terrestrial rivers (195,022 data points), global deltas (10,848 data points), the continental shelf (14,820,634 data points), the continental slope (7,606,463 data points) and the key ocean floor landforms, including the continental rise (12,144,045 data points), abyssal plain (116,749,407 data points) and hadal zone (1,238,491 data points) on Earth to identify the upper and lower bounds of the continental shelf. On Mars, the analysis included valley networks (3,294,322 data points), depositional rivers (16,515 data points), outlet canyons (248,865 data points), deltas (48 data points), and the Arabia (10,192 data points) and Deuteronilus (42,900 data points) shorelines, to establish the upper bound of the potential Martian shelf. On Earth, river deltas prograde across and rest atop continental shelves, and the transition from these deltaic deposits to the deep ocean typically takes place within the upper 2.5 km below sea level. We, therefore, use this depth interval to define the search window for a potential shelf on Mars.

### Raster to points of elevation, slope and curvature

To obtain elevation at each point of the resampled raster files, we used the ArcGIS 'Add Surface Information' tool to sample elevation values from grids at resolutions of 2.5 km, 5 km and 10 km (refs. 70,71; Supplementary Figs. 2 and 3). For each point, the $z$-value is derived from its $x$–$y$ coordinates on the underlying surface.

To calculate the steepness of each cell on both terrestrial and Martian surfaces, represented as raster grids, we used the 'Slope' tool in ArcMap[71]. Slope (degrees) was calculated in ArcMap (Spatial Analyst)

using the Slope tool (3 × 3 neighbourhood), which estimates $\partial z/\partial x$ and $\partial z/\partial y$ using a finite-difference gradient and computes $S° = \arctan(\sqrt{(\partial z/\partial x)^2 + (\partial z/\partial y)^2}) \times 57.29578$.

Curvature shows the shape of the slope, indicating whether it is convex (that is, ridges and plateaus) or concave-up surfaces (that is, valleys and depressions), which is particularly useful for identifying transitions between landscape and seascape. We calculated curvature using the 'Curvature' function in ArcMap[71], which fits a plane to the nine surrounding cells in a 3 × 3 window to determine surface curvature. The primary output provides cell-by-cell curvature values (second derivative of elevation). The positive values indicate upwardly convex surfaces, negative values indicate upwardly concave surfaces, and values near zero represent flat or nearly planar areas. In ArcGIS, curvature values are reported in units of one hundredth (1/100) of the DEM z-unit (here, metres; z-factor = 1). To treat concave and convex areas equally in terms of magnitude, negative values of both planets are multiplied by −1, allowing them to be plotted alongside positive values with different colours (Fig. 2).

### Landscape classification

To map the transition zone between continental and oceanic landscapes on both Earth and Mars, we used the 'Geomorphons' tool[53]. This algorithm uses the concept of Local Ternary Patterns (LTP) to analyse terrain features by comparing the elevation of each pixel with its neighbouring pixels[54]. Instead of a simple binary comparison, LTP classifies differences into three categories: values that are (1) similar to the centre pixel, (2) significantly higher or (3) significantly lower. This approach reduces the impact of noise and provides a more nuanced representation of terrain, capturing subtle variations in pixel elevation. The Geomorphons tool classifies each cell of an input raster into common landforms, including flat areas, ridges, shoulders, spurs, slopes, pits, footslopes, hollows and peaks.

On Earth, the transition typically occurs on a relatively flat surface, known as the continental shelf[29]. To detect these flat surfaces, we used the Geomorphons tool with a specific 'flat terrain angle threshold'. The tectonic system of Earth, driven by active plate tectonics, differs significantly from Mars, which lacks substantial tectonic activity. This absence of tectonism on Mars results in longer topographic wavelengths compared with Earth[52]. As a result, we applied different flat-terrain angle thresholds for the two planets.

For Earth, we conducted 40 Geomorphons experiments (Supplementary Table 2) to (1) classify the surface of Earth into shelf cells and non-shelf cells by comparing the detected flat cells to the mapped continental shelf; (2) determine the flat angle threshold that fully detects the terrestrial continental shelf; and (3) establish a range of shelf area detection at each angle threshold (Extended Data Figs. 2–4 and Supplementary Table 2), which will be used to assess the percentage and accuracy of shelf detection on Mars. In each experiment, we applied a different flat-terrain angle threshold and found that the continental shelf was fully detected at an angle of 1.22° (Supplementary Table 2). However, increasing the flat angle threshold led to the detection of both shelf and non-shelf areas. For instance, we detected 100% of the shelf (32,308,476 km²), but it also identified 238,719,226 km² of non-shelf areas, resulting in a precision of 12%. We, therefore, used the experiments to set an accuracy range for each flat angle threshold, which would be used to detect the Martian shelf (Extended Data Fig. 3 and Supplementary Table 2).

For Mars, no definitive maps of oceanic features exist, apart from two long-debated proposed shorelines[10,22,77]. To map potential shelf-oceanic zones, we focused on a region in the northern lowlands (Fig. 2c) characterized by a distinct flat zone compared with its surroundings. This region also preserves 48 deltaic systems, some of which are connected to interpreted submarine-channel belts that are thought to have formed along an ancient oceanic margin[4,10,14–17] (Supplementary Table 1). Moreover, it contains numerous valley network termini and

fluvial depositional ridges, both of which are believed to represent the endpoints of fluvial systems. The region also preserves the two debated shorelines. This zone is well-defined by elevation, ranging from −1,800 m to −3,800 m, with a distinct median slope of 0.31° at a grid resolution of 5 km. We ran the tool at this flat angle threshold and found that nearly the entire northern lowland was marked as a relatively flat surface. To refine the results, we combined our topographic analysis (elevation, slope and curvature) with key morphological features (valley network termini, fluvial ridges, deltas and the two proposed shorelines). This allowed us to spatially constrain the flat surface zone between −1,800 m and −3,800 m to areas coinciding with geomorphic indicators of a landscape-to-seascape transition, mostly located between about 30 °S and about 70 °N. On Earth, a flat angle threshold of 0.31° would detect nearly 69–71% of the continental shelf area, giving us confidence that the detected surface on Mars corresponds to a similar transition (Supplementary Table 2).

### Statistical analysis

To test whether surface steepness differs between elevation zones, we first computed median slope and curvature values at 200 m elevation intervals on both Earth and Mars. The resulting Martian profiles show an intermediate-elevation, low-slope, low-curvature interval between −1,800 m and −3,800 m, bounded by higher slopes and curvature at elevations >−1,800 m and lower slopes and curvature at elevations <−3,800 m (Fig. 2 and Extended Data Fig. 1). On this basis, we defined three elevation bands: >−1,800 m, −1,800 m to −3,800 m, and <−3,800 m. We then applied a Kruskal–Wallis $H$ test[80] to these three elevation bands to quantify whether their slope distributions differ, without assuming normality. The test showed a highly significant difference in median slopes ($H = 27.50$, $P = 1.07 × 10^{-6}$; Extended Data Fig. 1), indicating that the slope populations of the three elevation zones are statistically distinct. The Kruskal–Wallis test is used here to assess only the distinctness of elevation zones defined from the slope–curvature–elevation relationship, not to locate the breaks themselves.

### Limitations

Our results are subject to several limitations. One is the potential alteration of topography due to true polar wander and the emplacement of the Tharsis volcanic province, which probably caused uplift near Tharsis and subsidence farther away[11,12]. However, analysing the surface as geomorphic domains helps mitigate this, because this deformation would affect broad regions rather than the specific elevation ranges of landform mosaics. A second source of uncertainty is the isostatic response to ocean unloading, which on Earth can modify elevations by several hundred metres following ocean retreat[81]. However, recent estimates for Mars suggest that isostatic rebound probably ranged from several tens to just more than 100 m (ref. 12). Yet, the approximately 2 km elevation span of our detected shelf-like zone exceeds expected rebound estimates and remains consistent with depositional features. A third source of uncertainty is long-term burial, exhumation and erosion[46]. Although these processes may have introduced regional variability, they are unlikely to alter the broader topographic patterns we identify at the global scale. A fourth source of limitation is the erosion and sediment redistribution along the dichotomy boundary by Hesperian-aged outflow floods[13,48], which probably deposited substantial volumes of sediment along the northern dichotomy, particularly in Chryse Planitia. These outflow events probably contributed to locally flattening the surface there. However, similarly flat, low-slope surfaces are also present at other key sites, such as Aeolis Dorsa—rich in stacked deposits of different origins and interpretations, including fluvial and deltaic deposits, and possibly even submarine deposits[15–17,82–86]—and along the remaining segments of the proposed shelf. This broader distribution, together with independent evidence for sea-level changes recorded by deltaic deposits at Hypanis[14], suggests that although Hesperian-aged outflow floods helped flatten the surface in Chryse Planitia, they were

probably not the primary cause of surface flattening across the northern lowlands.

## Data availability

All topographic data analysed in this study—including raw elevation, slope and curvature for Earth and Mars at 2.5 km, 5 km and 10 km per pixel, and the shelf-cell point set interpreted as a putative Martian shelf—are archived on Zenodo (https://doi.org/10.5281/zenodo.18868496)[87]. The Zenodo archive also includes the extracted elevation values and spatial distributions used in our analysis of Martian water-formed landscapes (valley networks, outlet canyons, interpreted deltas and putative shorelines) (https://doi.org/10.5281/zenodo.18868496)[87]. Our analysis relied on two main topographic datasets for both Earth and Mars: the ETOPO Global Relief Model for Earth (60 arcsec resolution), available from the NOAA NCEI website (https://www.ncei.noaa.gov/products/etopo-global-relief-model) and the global MOLA gridded topography, available from the USGS Astrogeology website (https://astrogeology.usgs.gov/search/map/mars_mgs_mola_dem_463m). Maps of rivers (https://datacatalog.worldbank.org/search/dataset/0042032) and deltas (https://doi.org/10.17605/OSF.IO/S28QB) on Earth and terrestrial seafloor geomorphic features (https://bluehabitats.org/?page_id=58) were sourced from published datasets cited in the main text and reference list[40–42]. Martian geomorphic maps (valley networks (https://doi.org/10.18738/T8/STRFZH)[44], outlet canyons (https://doi.org/10.18738/T8/STRFZH)[44], depositional rivers[45], interpreted deltas[4,14–17] and putative shorelines (https://doi.org/10.5281/zenodo.4666803)[10,22]) were also sourced from the cited references.

## Code availability

The analyses presented here do not depend on specific code; the approach can be reproduced following the procedures described in the Methods.

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

**Acknowledgements** We acknowledge funding from the Swiss National Science Foundation Postdoc Mobility (grant no. P500PN_206718). We also acknowledge support from the California Institute of Technology. A.S.Z. acknowledges support from the Jackson School of Geosciences Distinguished Postdoctoral Fellowship, The University of Texas at Austin, during the revision of this manuscript.

**Author contributions** A.S.Z. and M.P.L. conceived and designed the study. A.S.Z. conducted the data analysis and drafted the manuscript, with inputs from M.P.L. A.S.Z. and M.P.L. revised the text.

**Competing interests** The authors declare no competing interests.

**Additional information**
**Correspondence and requests for materials** should be addressed to Abdallah S. Zaki.

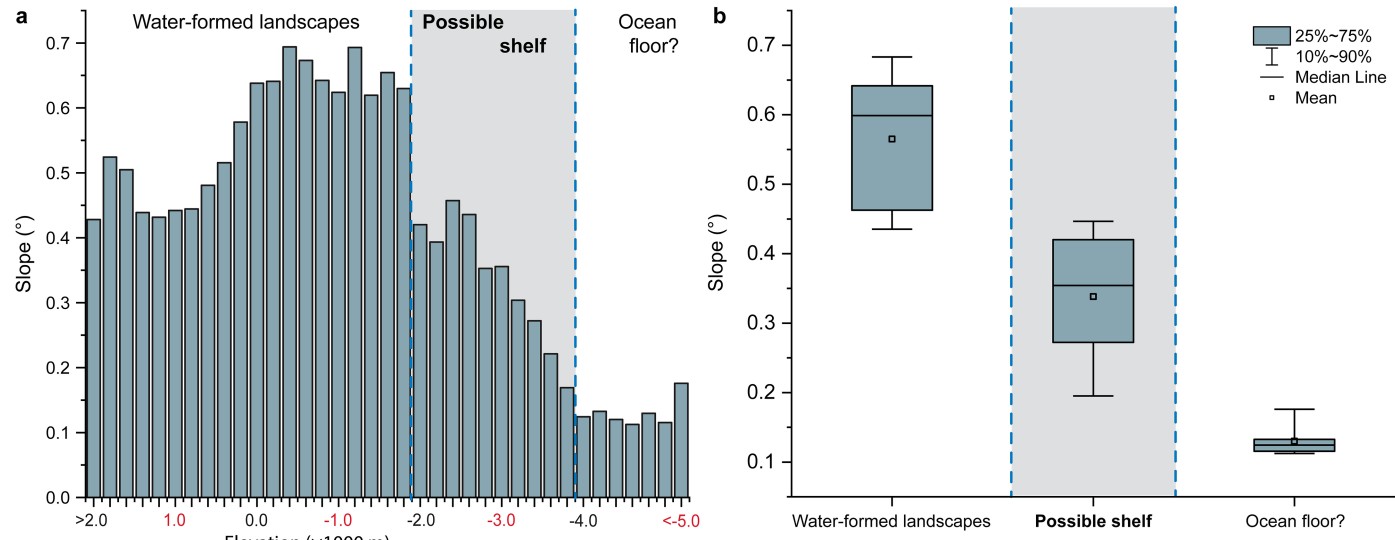

**Extended Data Fig. 1 | Median slope values across three elevation groups on Mars: water-formed landscapes, possible shelf, and ocean floor.** (a) and (b) Column and box plots show absolute median slope values for 200 m elevation bins across the three elevation groups shown in Fig. 2c, revealing two distinct breaks at −1800 m and −3800 m. These patterns, together with results from the Kruskal–Wallis test (see Methods), indicate statistically significant differences between the groups.

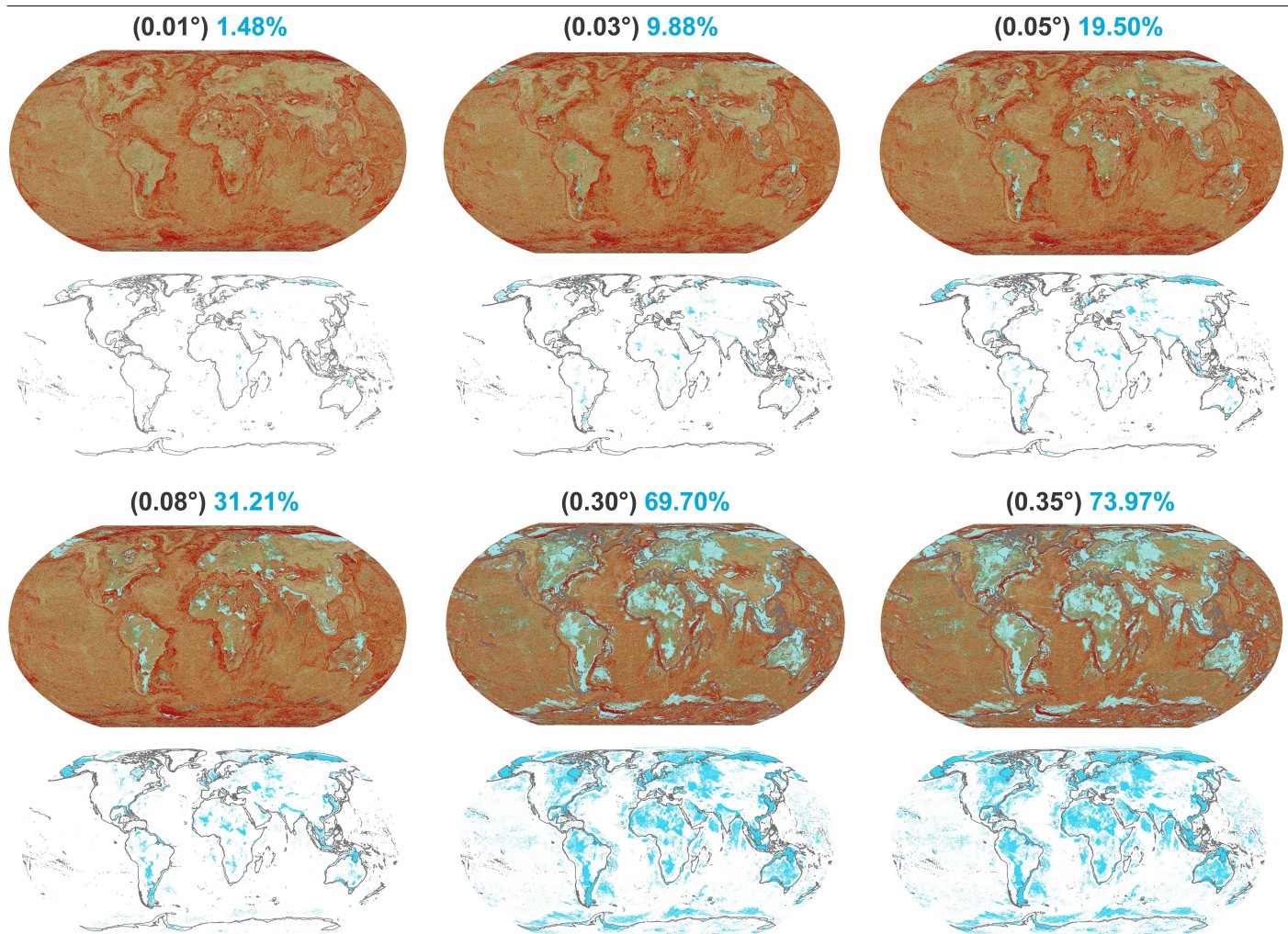

**Extended Data Fig. 2 | Examples of the Geomorphons algorithm output and its comparison to the mapped shelf on Earth[42].** The sky-blue color represents cells with a maximum slope equal to the specified value mentioned on top of each output. The figures below illustrate the correspondence between the sky-blue cells and the mapped shelf.

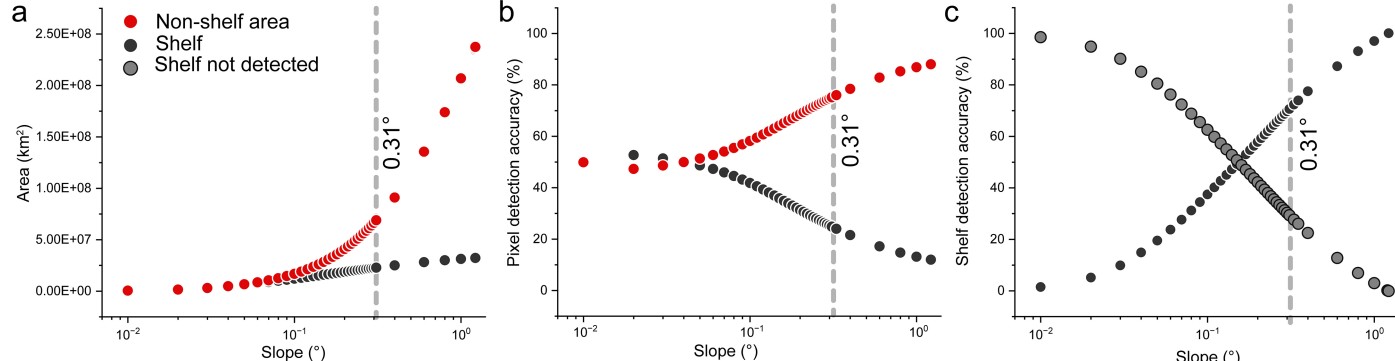

**Extended Data Fig. 3 | Results from 40 experiments aimed at constraining the accuracy of shelf versus non-shelf detection.** (a) The experiments show that increasing the flat terrain angle threshold leads to improved detection rates for both shelf and non-shelf areas. (b) Comparing the ratio of detected shelf to mapped shelf at various angle thresholds allows for an assessment of detection accuracy. The terrestrial shelf was fully detected (100%) at an angle of 1.22°. But, a flat angle threshold of 0.31° detected approximately 71% of the shelf. (c) Comparison of the percentage of the detected shelf with the percentage of the shelf that was not detected.

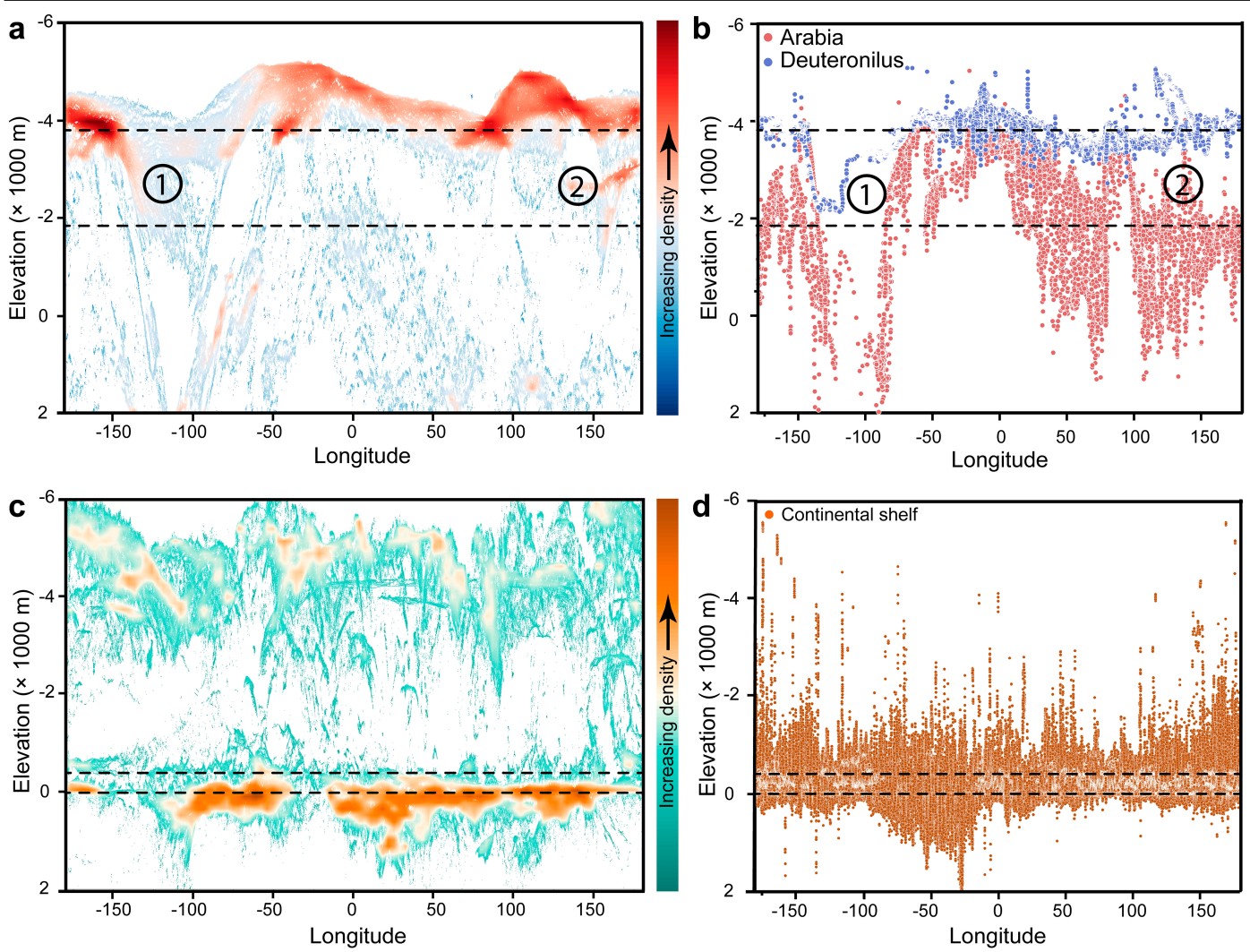

**Extended Data Fig. 4 | Detected flat surfaces at a flat angle of 0.31° on both Mars (a) and Earth (c).** Comparing these detected flat surfaces (a) with mapped shorelines on Mars (panel b) reveals that elevation deviations in the shorelines likely result from the presence of these flat surfaces (1, 2), which may preserve the shelf zone. Similarly, comparing the detected flat surfaces (c) with the mapped shelf (d) on Earth shows that the detected area, ranging from -410 to -15 meters in elevation, represents 71% of the mapped shelf.

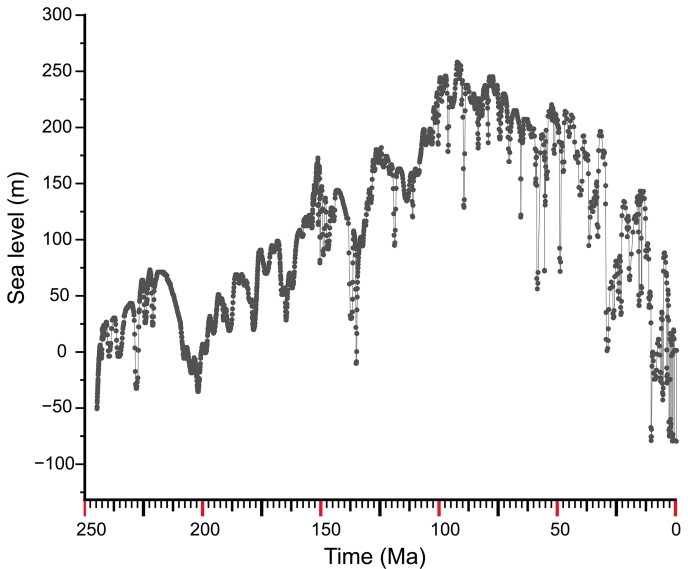

**Extended Data Fig. 5 | Sea level changes on Earth based on depositional models[88].** The graph illustrates that, over the past 250 million years, Earth has experienced sea level fluctuations, with regressions and transgressions occurring within a range of approximately 300 meters.

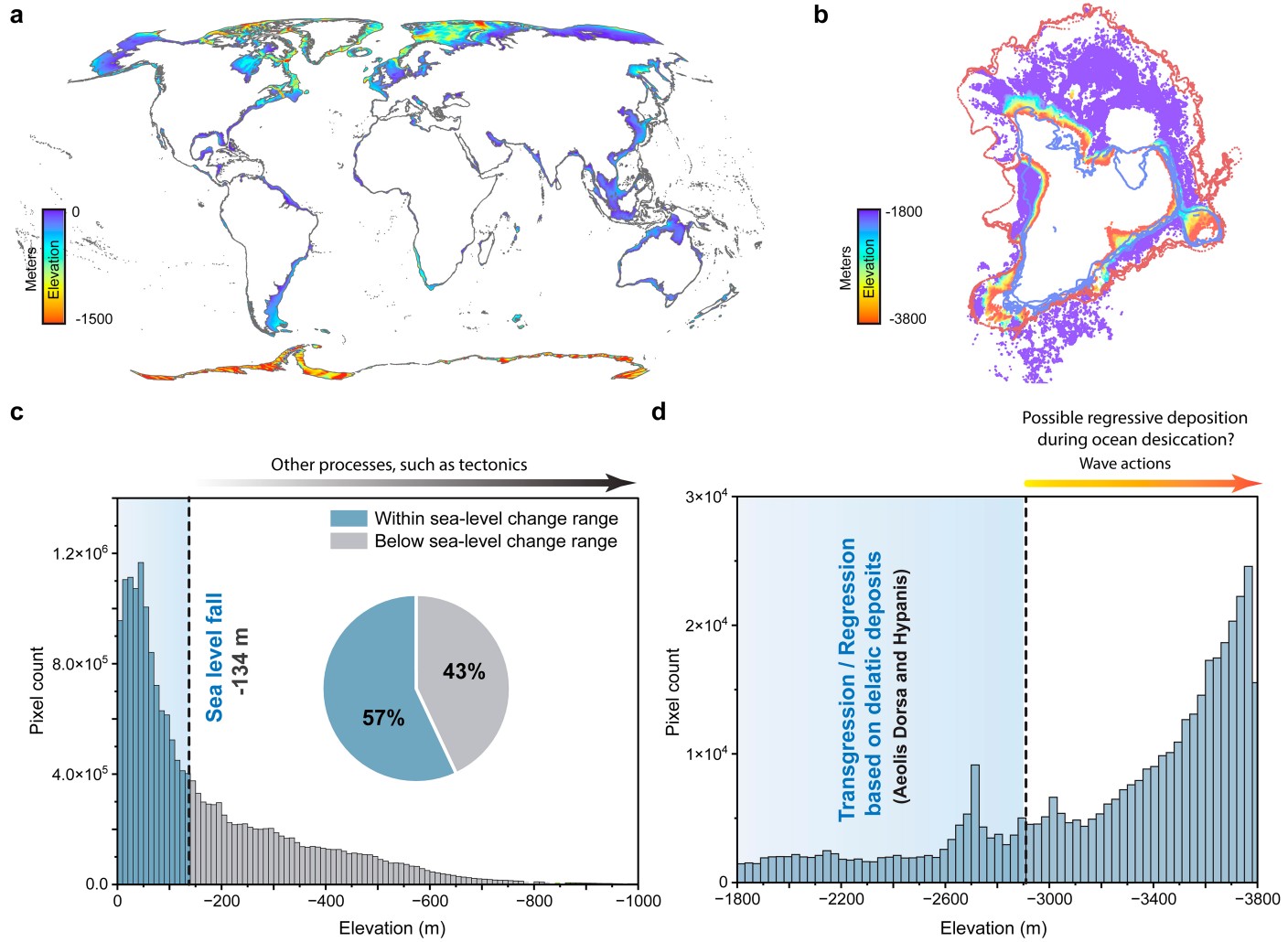

**Extended Data Fig. 6 | Elevation ranges across the terrestrial and Martian shelves.** (a,c) Elevation distribution across the mapped terrestrial shelf. These plots show how sea-level fall during the Last Glacial Maximum concentrated most shelf pixels into the glacial lowstand band; pixels at deeper elevations largely reflect additional processes such as tectonic subsidence. (b,d) Elevation distribution of the detected shelf-like pixels on Mars, compared with the elevations of the mapped Arabia shoreline (red) and Deuteronilus shoreline (blue). In (b), shelf-band pixels (−1800 to −3800 m) that satisfy the low-slope criterion are coloured by elevation using a single purple–to–yellow–to–red scale and shown in polar stereographic projection for visualization. Panel (d) shows the corresponding elevation histogram computed from an equal-area raster, so pixel counts are proportional to area and demonstrate that the deepest part of the band occupies the largest area. The shaded band marks the elevation range of relative sea-level change inferred from transgressive and regressive deltaic deposits at Aeolis Dorsa and Hypanis (Fig. 5; Extended Data Fig. 7). Pixels below this elevation range may represent regressive depositional environments associated with shoreline retreat, or surfaces modified by wave action during ocean desiccation, consistent with recent reports of coastal deposits up to -500 m below the Deuteronilus shoreline[18]. Together, these observations indicate that the transgressive–regressive phases captured by preserved deltas account for only a small fraction of the detected shelf-like pixels, and that prolonged shoreline retreat during ocean drying possibly produced additional, subtler depositional surfaces at lower elevations[18].

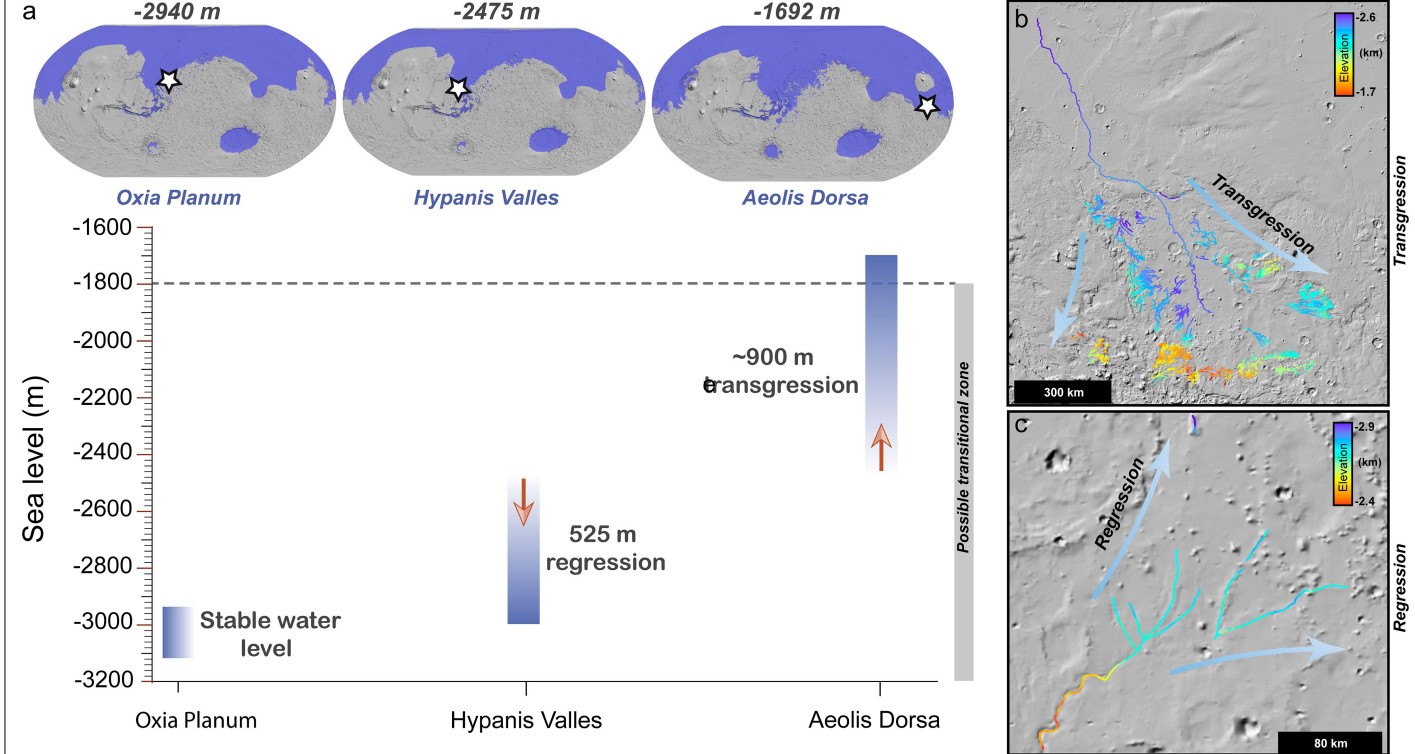

**Extended Data Fig. 7 | Sea level changes based on stratigraphic analyses from three major deltaic systems (Oxia Planum, Hypanis Valles, and Aeolis Dorsa)**[14,17,76]**. (a)** The figure demonstrates significant sea level fluctuations over geologic time, including a regression of approximately 500 meters and a transgression of around 900 meters. **(b)** and **(c)** show the elevation profiles distribution of the deltaic systems in Aeolis Dorsa and Hypanis. These level changes occur within the identified transitional zone (−1800 meters to −3800 meters).

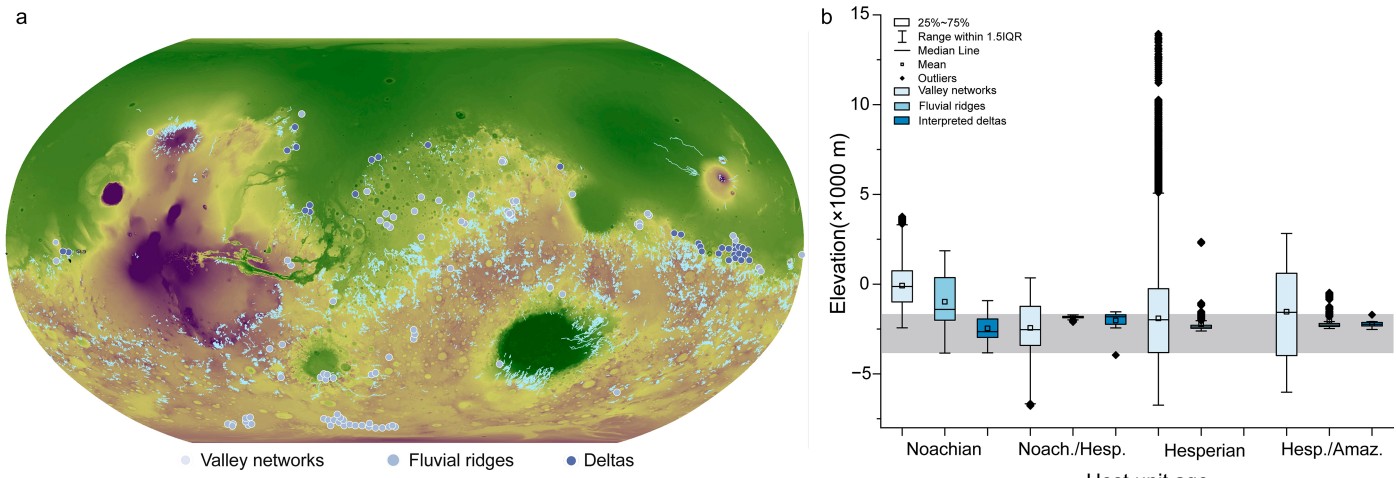

a

b

**Extended Data Fig. 8 | Source-to-sink fluvial deposition in the northern lowlands of Mars.** (a) Map showing the geomorphic units used to investigate the source-to-sink framework on Mars. These include valley networks, depositional ridges (currently preserved as inverted relief), and fan-shaped deposits interpreted as deltas that are not confined within craters. The distributions of these landforms are compiled from refs. 4,10,14–17,44,45,74,78,79. The background map is derived from NASA's Mars Orbiter Laser Altimeter (MOLA) data[66–68]. (b) Box plot illustrating how possible source-to-sink fluvial systems on Mars vary across elevation and geologic time. Elevation data were extracted from the MOLA topographic grid, and the relative ages were assigned based on the host geologic unit using a simplified global age map[89]. This approach reveals a potential global pattern in the spatial and temporal evolution of source-to-sink systems. The grey band marks a possible transitional zone between upstream landscapes and the termini of depositional systems (i.e., fluvial ridges and deltas). A strong association is observed between this transitional zone and the occurrence of depositional landforms.