## [Peer Review file · Nature]

Identifying the topographic signature of early martian oceans

Corresponding Author: Dr Abdallah Zaki

Version 0:

Reviewer comments:

Referee #1

(Remarks to the Author)

This paper shows that the continental shelf on Earth has a topographic signature (minima in slope and curvature), and used this criteria to search for evidence of a coastal shelf on Mars. The study seeks to investigate a very subtle topographic expression from oceans, first convincingly demonstrating the characteristics preserved on Earth and then exploring Mars for a similar signal. The work uses a novel approach and is a thought-provoking consistency test for a hypothesized martian ocean. The paper is well written and clearly organized. Clarification of the methods and terminology, as well as refinement of the figures would improve the manuscript for publication.

Methods.

1. Is there a statistical test that could evaluate the minima reported? Can the manuscript further describe how the -3800 m distinction between the two coupled minima was determined (L119-122)? By visual inspection of figure 3c/d?
2. The model did not independently identify the coast-to-ocean zone on Mars. L351-354 "Flat surfaces on Mars and Earth were identified using a flat terrain angle threshold of 0.31° , selected as it represents the median slope of the zone (-1800 m to -3800 m) that includes shorelines, interpreted deltas, and distinct flat surfaces." The manuscript contends that the automated landscape classification using this criteria matches the coastal landforms (depositional forms and shorelines), but this is by design. (See also next point).
3. There appears to be circular reasoning, as my understanding of the selection of data for analysis on Mars ensures the result.
 - a. Reviewer's summary of methods: At 5-km resolution, the martian northern lowlands is extremely flat ($<0.5^\circ$). Using the median slope of the first flat zone (identified in Fig2C) within the search area (e.g., less than -1638 m, the upper elevation of martian delta, and confined to the northern lowlands), the autonomous algorithm identified the entire northern lowlands as 'flat'. This result was then truncated to the terrain between erosional inlets (e.g., valley network) and putative shorelines (e.g., -1800 and -3800 meters). The authors selected the zone between "valley network termini, fluvial ridges, deltas, and the two proposed shorelines". Therefore, the detected shelf in this study was by design the zone between these landforms in the northern lowlands. Incidentally, this result would be identical if the study only adopted the final step (e.g., find the region between "valley network termini, fluvial ridges, deltas, and the two proposed shorelines" in the northern lowlands, omitting the autonomous component.) Step by step details are as follows:
 - b. The study examined the northern highlands. L506 "To identify potential shelf-oceanic zones, we focused on a region in the northern highlands (Fig. 2C) characterized by a distinct flatter zone compared to its surroundings."
 - c. The authors selected the region between two candidate shorelines, Arabia and Deuteronilus.
 - d. L512 [The region between two debated shorelines] "is well-defined with a distinct median slope of 0.31° at a grid resolution of 5 km. We ran the algorithm at this flat angle threshold at this grid resolution and found that the entire northern lowland was marked as a relatively flat surface. To refine the results, we combined our topographic analysis (elevation, slope, and curvature) with key morphological features (valley network termini, fluvial ridges, deltas, and the two proposed shorelines)."
 - e. L514 "To refine the results, we combined our topographic analysis (elevation, slope, and curvature) with key morphological features (valley network termini, fluvial ridges, deltas, and the two proposed shorelines). This allowed us to filter the flat surface zone between -1800 and -3800 meters, which likely preserves the transitional zone between landscapes and seascapes." It is unclear what is meant by 'filter'.
4. L143: How much of the martian surface in the northern lowlands is between -1800 and -3800 m? Is it 7%? "our automated

landscape classification identified 10.2 million km², or 7% of the martian surface, as a potential coastal shelf at elevations between -1800 m and -3800 m, clustered around the dichotomy boundary." Please clarify. Is this sentence reporting the surface area of Mars between -1800 and -3800 m elevation?

5. The authors discussed non-shelf cells detected in the terrestrial models (e.g., L493-504). Could they also discuss non-shelf detections on Mars. For example, Figure 3d shows significant terrain above the putative Arabia shoreline (red dots).

6. L491: "We applied different flat terrain angle thresholds for the two planets." What value for the flat terrain threshold angle was used for Mars? L 139-143 implies that 0.31° for 5 km grid resolution was used. However, this appears to be the same value used for Earth (L352)?

7. The second topographic minima (L121/122) is not mentioned again or discussed. Presumably, this is because the elevations between -3800 to -5000 m were filtered out (L514).

8. The author's could consider applying the same technique to other planetary bodies. Titan would offer another test of a topographic signature from an ocean. Oceanless worlds like the moon or Venus could be used to evaluate the robustness of this approach.

Clarifications for Discussion. The subtle topographic expression investigated is not uniquely tied to process, and alternative explanations should be discussed.

1. Can the authors discuss the impact of Tharsis on their data analysis?

a. Lowlands on several planetary surfaces are due to volcanism (e.g., moon, Venus). Furthermore, volcanic flows into the martian lowlands is well documented. In the 2005 USGS geologic map of the northern plains, "Apparently formed early in Mars' history, the northern lowlands served as a repository both for sediments shed from the adjacent ancient highlands and for volcanic flows and deposits from sources within and near the lowlands." <https://pubs.usgs.gov/sim/2005/2888/>

b. Furthermore, there are no mapped valley networks along a considerable stretch of the proposed shelf (e.g. roughly longitude 220-320° E; Fig. 1C).

2. L129 reports the elevation of many landforms associated with coasts are "at elevations above the putative coastal shelf, as expected." However, this result is by design. The study adopted a threshold elevation based on these same landforms (L89-101).

3. L190 Given the subtle signature of putative oceans, what potential evidence might rovers observe to validate or repudiate the hypothesis? Can you elaborate?

Terminology.

1. Please define outlet valleys (this terminology could be misconstrued with the well established martian term 'outlet channels').

2. The abstract states that the topographic feature on Earth is due to the coastal plains and the continental shelf (L16), yet reference to coastal plains is minimal in the text and absent in the figures.

3. It would be useful to definition of depositional rivers

Figures.

1. As they hypothesized martian ocean is exclusively in the northern hemisphere, the author's are encouraged to render there data in a polar stereographic projection to enhance the presentation (extended data figure). As an example, see Figure 1 in Sholes et al., 2021 (<https://doi.org/10.1029/2020JE006486>).

2. The full-page, detailed figures are difficult to read, and require enlargement (200%) on the screen to read the labels. There is concern that figures will be illegible if shrunk further for publication.

3. Figure 1a: What are the light blue polygons? Watersheds? Figure 1c: It is difficult to distinguish the green and blue lines (valley networks and outlet valleys) on the grey background. The small dots on the highlands are presumably depositional rivers, but the interior color of the circle is hard to discern.

4. Figure 2c: The area labeled possible shelf appears to extend beyond -1800 m. Is that intentional? Perhaps you mean search zone for shelf. At any rate the distinction between 'possible shelf' in this figure and 'probable shelf' in later figures may cause confusion if these are indeed meant to have different meanings. Note that Figure 4 adopts 'possible shelf'.

5. Figure 4c: How were the extents of the profiles determined? More specifically, why is the Hypanis profile so long and the Arabia Dorsa profile so short? Figure 4d: Consider moving or changing label ('shoreline') for Arabia and Deuteronilus, as these refer to both putative shorelines as well as geographic regions on Mars.

Minor comments:

L 353: To clarify, suggest adding the word martian before zone.

L507: Do you mean northern lowlands?

Almost the entire northern highlands is encompassed within the two zones identified in this study (-1800 to -5000 m; L119-122).

Fig.2 caption heading needs revised. "Slope and curvature of the entire surfaces of Earth and Mars." However, martian surface from +2 to + 8 km is excluded from the plots.

Referee #2

(Remarks to the Author)

The manuscript by Zaki and Lamb proposes a bold new hypothesis: that the topography surrounding the northern lowlands on Mars contains a feature analogous to the continental shelves on Earth, and that this shelf-like feature formed around the

margins of an ancient ocean. The idea that Mars once had an ocean is not new, but most previous studies that tested this idea have focused on possible shorelines or small coastal landforms (such as river deltas) that might have formed in the vicinity of shorelines, both of which are challenging to interpret because they are susceptible to degradation and alteration. The novelty of the Zaki and Lamb hypothesis is that it focuses on a broader-scale topographic feature that is more likely to have persisted over the billion+ years since the ocean would have dried up. The analysis does not demonstrate with complete certainty that Mars had an ocean, but it is the most substantial and creative contribution to the Mars ocean hypothesis in years, and it is bound to attract a lot of interest. It is also timely, given the recent discovery of possible coastal sedimentary deposits on Mars by China's Zhurong rover (Li et al. PNAS 2025).

I have two general comments, numbered below, followed by a set of comments related to specific passages in the paper. I think the authors should consider these comments before the paper is published.

General comments:

1. Null hypothesis.

On Earth, the bathymetry of the continental shelf presumably reflects the ocean-continent contrast in crustal density and thickness, combined with sedimentation and wave modification during dozens of glacial sea-level lowstands. In the likely absence of both major factors on Mars, should we expect a similar shelf to form? The authors acknowledge these differences and note (lines 125-127) that a huge impact probably created the dichotomy in crustal thickness on Mars (Andrews-Hanna et al. 2008). The null hypothesis, then, should be that any topographic characteristic of the northern lowlands is a product of the impact. The authors should explain why an ocean is a better explanation for the topographic feature they identified than an impact.

2. Post-ocean topographic change.

The authors assume implicitly that Mars' long-wavelength topography has not changed substantially since an ocean filled the northern basin. The mechanisms of subsequent erosion and sediment redistribution they mention (wind erosion, meteor impacts, and regolith creep; lines 124-125) are mostly local and would not have altered the large-scale topography much. However, geophysical mechanisms could have modified the large-scale topography of an ocean basin, including any shelf. True polar wander (Perron et al. 2007; Citron et al. 2018), dynamic topography, or deformation associated with the removal of the ocean water load could have generated long-wavelength topographic change. Even if the authors disagree with previous suggestions of true polar wander or dynamic topography, they should consider the isostatic and flexural response to removal of the ocean load, because it must have occurred if there was an ocean. The authors are comparing the bathymetry of Earth's oceans (at a sea-level highstand) with an empty basin on Mars. How would Earth's ocean bathymetry change if the water was all removed?

Comments on specific passages:

118-125: I'm having trouble seeing two minima in Figure 2b,d. The features at -3800m look more like a step down. Perhaps the box-and-whisker plot is not the most effective way to show this upper feature. I note that the contrast between the left and right columns in Fig. 2 makes the conceptual model in Figure 4 important: The evolution of a shelf-like feature on Mars may have differed from Earth, resulting in a more gradational transition from land to ocean basin (Fig. 4d).

127-128: This sentence should be the first sentence in the next paragraph.

130-132, 508-510: Chan et al. (JGR Planets 2018) considered valley networks as markers bounding the location of an ocean coastline, as the authors are suggesting. That paper would provide useful context for their arguments.

132-133: The possible Arabia shoreline switches from the inner to the outer edge of the possible shelf in the area of Arabia Terra (Fig. 3b). Perhaps deformation due to true polar wander, suggested by some previous studies (Perron et al. 2007; Citron et al. 2018) would reconcile this observation. More generally, it would be interesting to see if this proposed deformation would reduce the range of elevations of the hypothesized shelf (Fig. 1d and Fig. 2c,d).

151-152: "The transition from land to sea is not a distinct shoreline, but rather a zone of low slope and curvature near sea level on Earth." These are not mutually exclusive – Earth's coastline is a topographic and bathymetric feature, yes; but it is also marked by distinctive landforms, which is what previous authors have considered for Mars.

157-161: Does the continental shelf on Earth line up with glacial sea-level lowstands everywhere? Have the authors compared Earth's shelves with a lowstand sea-level model that includes solid-Earth deformation?

171-184: This is an important point that is sometimes neglected in studies of possible Mars coastal landforms. I agree with the authors that recessional coastal landforms could span a wide range of elevations on Mars.

192-194: I suggest rephrasing to "rather than only looking for distinct shorelines". The authors are not the first to suggest that fluvio-deltaic depositional landforms may mark shorelines. Indeed, depositional landforms are one class of features that have been used in multiple previous studies to identify possible shorelines (including by the second author: Cardenas & Lamb 2022; DiBiase et al. 2013).

Comments on figures:

Figure 1: In 1a, the delta points obscure the continental shelf and slope, the main subject of the paper! Also, what are the colors in 1b and 1d?

Figure 3: In 3c and 3d, the black transects appear to correspond to the text labels. I don't think that is what is meant. Also, the Rhine Valley and Hypanis Vallis are not perfectly straight. I think the authors mean the lines on their plots are transects in the orientations of these features.

Text labels in figures contain multiple typographical errors.

Referee #3

(Remarks to the Author)

This paper investigates whether a paleo-coastal shelf, formed by an ancient ocean(s), could exist in the northern lowlands of Mars. Whether an ancient ocean once filled Mars's northern hemisphere-spanning basin is a major open question in planetary science today. Many previous studies have focussed on whether the dichotomy boundary, which divides the northern lowlands of Mars from the southern highlands, is an erosional shoreline – this work proposes instead that we would not necessarily expect to see an erosional shoreline, but a depositional coastal shelf. This investigation primarily compares the modern topography of Mars's northern lowlands to coastal marginal topography on Earth. The investigation is framed both in the context of whether planets should be expected to leave discernible shorelines, and whether Mars has evidence for one. In general, the authors find strong similarities between the topography of Mars dichotomy and coastal shelves on Earth, such as slope and curvature.

The paper raises some interesting concepts, particularly what we should be looking for when investigating the signals of ancient oceans on Mars. I generally agree that a coastal shelf on Mars is possible – however, I do not believe that the analysis as it is currently presented makes a sufficient or plausible cause for it. In addition, there are many caveats that the paper does not sufficiently address. For these reasons, I am declining to recommend publication.

(1) Open vs basin deltas

A major point of the paper is that fluvial landforms, such as inferred deltas, along the dichotomy region of Mars, fall within the expected elevation range/position for a coastal shelf/ocean boundary. This interpretation builds on earlier work (Di Achille & Hynek, 2010), which interpreted such features as forming an equipotential surface and potentially formed into an open basin (an ancient ocean). The scale of the topographic analysis done in this new study using relatively coarse resolution, global MOLA (463 m/pix) would also support this interpretation. Indeed, in the case of the two examples presented (Hypanis and Aeolis Dorsa), this interpretation also holds true.

However, since the Di Achille & Hynek study, there have been multiple regional-scale studies, which re-analyzed the inferred open basin deltas, using higher resolution topographic datasets. All of these studies found that many or most of the inferred deltas formed into closed basins, along the dichotomy – likely forming smaller paleolakes, and that a large, ocean-sized body of water was not required. None of these papers are cited in this new manuscript. The papers below all demonstrate that careful assessment of the local or regional geology is necessary, in order to support global hypotheses. The authors need to present the caveats of the distribution of inferred deltas – multiple studies have now shown that a one size fits all approach does not work. This, in my opinion, is a major flaw of the manuscript.

<https://agupubs.onlinelibrary.wiley.com/doi/full/10.1029/2021GL094271>

<https://agupubs.onlinelibrary.wiley.com/doi/full/10.1029/2019GL083046>

<https://doi.org/10.1016/j.geomorph.2020.107129>

(2) Global scale approach vs regional geological complexities

A recent published paper inferred that the highland-lowland dichotomy has retreated by 100s of kilometres in the Mawrth Vallis and possibly across the wider Chryse Planitia region. The authors show that this retreat records a major episode of erosion and occurred between 4.0 – 3.7 Ga. One potential cause of this could be the retreat of a Noachian age ocean on Mars. This would seemingly disagree with the authors' hypothesis (for this new manuscript) that terrain north of the dichotomy is primarily a product of deposition.

<https://www.nature.com/articles/s41561-024-01634-8>

Similarly, Hesperian-Amazonian age outflow channels, likely caused by megaflooding events, in the Chryse Planitia region have had a major impact on the topography (i.e., Ares Valles, Kasei Valles). These channel systems have both eroded the terrain and deposited huge volumes of material in this area of Mars. While the paper does acknowledge that multiple oceans on Mars could have existed at different stages in time (I guess a late stage one filled by outflow channel debauching?), outflow channels are not directly mentioned. I would like to know how outflow channel formation plays into the authors' coastal shelf hypothesis.

(3) Lack of consideration for alternative hypotheses

Figure 3 illustrates the proposed zone for the coastal shelf on Mars, which has largely been inferred from the gentle sloping topography (taking into account point 1 above). Similarly, figure 4 schematically and graphically illustrates how the topography changes downslope for both Earth and Mars. A major issue I have with this work is that other hypotheses for what could produce this topographic distribution aren't really considered. And again, this relates to the point above about regional geological complexities. Why could this type of topography not be produced by outflow channel re-surfacing (point above) in the Chryse region? Why not volcanic re-surfacing – for example, could the gently sloping terrain around Tharsis be associated with the volcanoes here? In Isidis, the lavas from Syrtis Major extend into the basin here, making it unlikely to be a coastal shelf.

Line by line:

Line 96: "Open-basin deltas along the northern dichotomy range in elevation from -1,638 to -2,880 meters" – The Scholes et al 2022 paper which is referenced here notes: "Although, subsequent detailed regional mapping has shown that many of these deltas likely formed in closed localized paleolakes or seas rather than a northern ocean"

Line 192: Rosalind Franklin is scheduled to launch in 2028 and land in 2030.

Figure 4c: typo on left plot "Hyapnis"

Version 1:

Reviewer comments:

Referee #1

(Remarks to the Author)

Publication is recommended once problematic new portions are removed (see below).

Substantial weakness was introduced into the revised manuscript with the addition of content related to Hellas and layered deposits in the circum-Chryse basin (External Data 9 & 10). These components—primarily figures with captions—are under-developed, lacking description and key literature. As this is an appeal, my critique is brief and highlights the issues with two examples

1) There are no citations with respect to Hellas Planitia, yet the Hellas basin has been proposed as the location of a former sea or "standing body of water." https://pubs.usgs.gov/sim/3096/sim3096_pamphlet.pdf

<https://agupubs.onlinelibrary.wiley.com/doi/full/10.1029/2006JE002830>

Such prior work should at least be acknowledged,

2) Unclear location of area shown in External Data 10c, but it appears to be part of the chaotic terrain in and near Chryse Planitia. How are the layered mounds related to or distinguished from the plains units surrounding Chryse? This is a critical question to address in order to claim the layered mounds are remnant coastal shelf deposits. Alternate models for chaotic terrain are missing.

Referee #2

(Remarks to the Author)

Like the original manuscript, the revised manuscript by Zaki and Lamb advances the provocative new hypothesis that the northern lowlands of Mars bear a previously unrecognized signature of a past ocean: a shelf-like region analogous to the continental shelves on Earth. Although the paper does not demonstrate beyond any doubt that this topographic feature formed in an ancient Mars ocean, it adds a new dimension to the debate and will undoubtedly inspire further research on the topic. Perhaps most importantly, the paper makes a basic point that has not been widely discussed: if Earth's oceans were to dry up, the enduring surficial evidence would not be beaches, but rather the broader bathymetry of the ocean basins, including the shelves. This large-scale bathymetry should be a focus of searches for evidence of a past ocean on Mars (in addition to surface geology being explored by rovers).

I have organized my comments according to the main points raised in my original review and the authors' responses to those comments.

Original comment 1. Null hypothesis

1.1. Comparison with Hellas: I agree with the authors that the Hellas basin, while smaller than the northern lowlands, offers the best available comparison. I also agree that the slopes in Hellas (Fig. ED10b) look more uniform than those in and around the northern lowlands (Fig. 2c). This shows that Hellas does not have as prominent a bench as the northern lowlands. However, eyeballing the plots in the paper suggests that the slopes in Hellas do form two distinct regions that span roughly a factor of 2 (Fig. ED10c) (compared with roughly a factor of 3 or 4 in the north; 0.015 degrees to 0.05 degrees, Fig. 2c), so Hellas does appear to have a flatter region near the edge of the analyzed area. Both slope ranges are small relative to the range of slopes on Earth, which is more like a factor of 20 (Fig. 2a). So my question for the authors is why – in terms of quantitative criteria – does Hellas not qualify as having a shelf, whereas the northern lowlands do? A related minor point: Fig. ED10 only shows slope, not slope and curvature, yet the authors cite this figure in their response to support the

statement that Hellas does not have coupled slope and curvature minima. I also find it difficult to see the blue lakes and yellow canyons in ED10a due to the blue and yellow colors in the elevation basemap.

1.2, 1.3: Sedimentary and stratigraphic features: These datasets are informative additions that specifically address the point that a shelf could be an ocean-generated overprint on an impact basin rather than an original feature of the impact basin.

Original comment 2. Post-ocean topographic change

The authors' response to this comment – that vertical motions associated with geophysical processes would be too small in amplitude to affect the topographic features they detect on Mars – is plausible, but it does not quantitatively demonstrate that the isostatic response to ocean unloading would not alter the slope and curvature patterns. Since the authors' argument is largely based on slope and curvature measurements, the possibility that deformation could tilt or warp topography seems important. However, I don't think this point should stand in the way of publication, in part because modeling the response to ocean unloading on Mars would require estimates of lithospheric properties that are currently unknown.

I would also find it helpful if the authors could address an apparent contradiction related to this point. As noted above, they suggest that geophysical mechanisms would not generate enough vertical deformation to substantially alter the topography of the hypothesized shelf on Mars. Elsewhere in the paper, however, they argue that more localized geological processes COULD alter the topography of the shelf: "The [slope and curvature] minima are less distinct than on Earth, but this might be expected, given that Mars' ocean, if it existed, has likely been dry for billions of years, with substantial topographic modification driven by wind erosion(36), meteor impacts(37), and regolith creep(38)."

Original comment: Are there two minima in slope and curvature on Mars?

In their response, the authors cite the K-W test in Fig. ED1 (they write Fig. ED3, but I think they mean ED1). I agree that the histogram in ED1a appears to show at least three (perhaps four) elevation zones with distinct slope distributions. However, I'm not convinced that the K-W test used in ED1 is the right choice for detecting breaks in slope. My understanding is that K-W tests for significantly different medians between samples that are not normally distributed, but it doesn't say where the boundaries between the samples should be drawn. The fact that the three highlighted regions have significantly different medians (Fig. ED1b) does not necessarily mean that there are breaks in slope between those regions. For example, wouldn't an inclined plane divided into three non-overlapping elevation regions yield the same result, even though the slope is the same everywhere?

Original comment: Do shelves on Earth correspond to sea-level lowstands?

I appreciate the authors' new analysis. Earth's continental shelf depths are a bit of a puzzle! But I must be missing something about Fig. ED6d. In ED6b, there are many purple dots (elevations around -2000 m), whereas points at lower elevations (yellow to red) are less abundant. But ED6d indicates the opposite: that pixels with lower elevations are much more abundant. What does the label "Transgression Regression" mean? (Those are opposites, and what are they labeling?) Are both maps equal-area, such that pixel counts are proportional to area? Polar stereographic projections are not equal-area.

Minor points:

629-632: Duplicate sentence but with different citations.

Fig. ED1: Caption should state that this is for Mars.

Referee #3

(Remarks to the Author)

Comments to authors.

This is the second time that I have reviewed this manuscript, which investigates whether a paleo-coastal shelf, formed by an ancient ocean(s), could exist in the northern lowlands of Mars, and what topographic signature that it would leave behind.

I really appreciate all the efforts that these authors have made to address the comments and issues that the other two reviewers and I raised. One previous concern I had was that the authors have not considered alternative hypotheses to the coastal shelf one and that there was regional complexities to the interpretation. While the authors have added significant new material, for example, concerning the remnant mounds in Chryse Planitia (<https://www.nature.com/articles/s41561-024-01634-8>), and the comparative topographic study of Hellas (as the non-ocean, null hypothesis), I still don't consider this point to be fully addressed. One outstanding point I have concerns outflow channels. From my original review:

"Similarly, Hesperian-Amazonian age outflow channels, likely caused by megaflooding events, in the Chryse Planitia region have had a major impact on the topography (i.e., Ares Valles, Kasei Valles). These channel systems have both eroded the terrain and deposited huge volumes of material in this area of Mars."

From the response doc:

"We also acknowledge the role of outflow channels in shaping the northern lowlands during the Hesperian and Amazonian. While these channels contributed both erosion and deposition, the key shelf-related deposits we highlight are older, with many stratigraphically and chronologically constrained to the Noachian–early Hesperian (e.g., 4.1–3.7 Ga), preceding the major outflow events. We now discuss the role of burial and erosion that may have further modified the shelf signal. These points are also reflected in the updated limitations section, where we explicitly address the potential overprinting of the original surface by later erosion and resurfacing events."

"A third source of uncertainty is long-term burial, exhumation, and erosion³⁹. While these processes may have introduced regional variability, they are unlikely to alter the broader topographic patterns we identify at the global scale."

The last highlighted sentence is a good point, but Chryse Planitia is one the two key regions making the case that the northern lowlands is a coastal shelf. There are several major outflow channel systems surrounding Chryse Planitia, which deposited huge volumes of material here via megaflooding. Indeed, the USGS geologic map of the northern lowlands attributes much of the material here to the outflow channels (<https://pubs.usgs.gov/sim/2005/2888/>). There is significant overlap between these outflow channel deposits and the regions that the authors are defining as coastal shelf. The authors have still not explained why the topographic signature they are seeing here is not because of the outflow channels.

Author Rebuttals to Initial Comments:

Referee comments are in **bold black text**

Our replies to comments are in **purple text (Times New Roman)**

Text that we revised upon reflecting on one or more Referee comments is in **light blue text (Arial)**

Reviewer #1 (Formal Review for Authors (shown to authors)):

This paper shows that the continental shelf on Earth has a topographic signature (minima in slope and curvature), and used this criteria to search for evidence of a coastal shelf on Mars. The study seeks to investigate a very subtle topographic expression from oceans, first convincingly demonstrating the characteristics preserved on Earth and then exploring Mars for a similar signal. The work uses a novel approach and is a thought-provoking consistency test for a hypothesized martian ocean. The paper is well written and clearly organized. Clarification of the methods and terminology, as well as refinement of the figures would improve the manuscript for publication.

We thank Reviewer #1 for their positive feedback and insightful comments, which have greatly improved the paper. Please find our responses to each specific comment below.

Methods.

1. Is there a statistical test that could evaluate the minima reported? Can the manuscript further describe how the -3800 m distinction between the two coupled minima was determined (L119-122)? By visual inspection figure 3c/d?

We thank the reviewer for this valuable comment. In response, we conducted a Kruskal–Wallis H-test—a non-parametric statistical test suitable for comparing more than two independent samples without assuming normality—to quantitatively assess the differences identified by visual inspection of the reported minima.

We grouped the data into three elevation intervals: (1) above -1800 meters, (2) from -1800 to -3800 meters, and (3) below -3800 meters. The Kruskal–Wallis test yielded a p-value of 1.07×10^{-6} , indicating that the likelihood of the observed differences in slope and curvature medians across these elevation bands arising by chance is less than 0.00011%. This result strongly supports the presence of statistically significant and abrupt shifts in slope characteristics between these zones.

Accordingly, we have incorporated this statistical assessment into the revised manuscript, updating the main text, methods section, and Extended Data Fig. 3 to reflect these changes.

Main text

“The same analysis of Mars' topography does not reveal a single distinct minimum in global slope and curvature, as observed on Earth (Fig. 2c, d; Extended Data Fig. 2). Instead, two coupled minima in slope and curvature are observed on Mars (Extended Data Fig. 3), supported by a Kruskal–Wallis H-test ($p = 1.07 \times 10^{-6}$), indicating less than a 0.00011% probability that the differences in median slope values among the observed elevation ranges occurred by chance. The first minimum lies between approximately –1800 m and –3800 m (slope = 0.31° , curvature =

0.00015, -0.00017), and the second between -3800 m and -5000 m (slope = 0.12° , curvature = 0.0009 , -0.0009). The minima are less distinct than on Earth, but this might be expected, given that Mars' ocean, if it existed, has likely been dry for billions of years, with substantial topographic modification driven by wind erosion³⁶, meteor impacts³⁷, and regolith creep³⁸. Moreover, the processes creating basin relief that allow for oceans differ. On Mars, an impact crater is likely the primary driver for the northern lowlands³⁹, whereas on Earth, plate tectonics allowed for chemical differentiation between oceanic and continental crust^{40,41}. Could these features represent the imprint of an ancient coastal shelf on Mars?"

Methods

Statistical analysis

To test whether surface steepness differs across geomorphic surfaces, we computed the median slope and curvature values at 200-meter elevation intervals. Visual inspection of the resulting profiles revealed distinct topographic breaks. To assess whether these breaks were statistically significant, we applied a Kruskal–Wallis H-test⁶⁶, a non-parametric method suitable for comparing more than two independent groups without assuming a normal distribution. The test revealed a highly significant difference in medians ($H = 27.50$, $p = 1.07 \times 10^{-6}$), indicating that the observed differences are unlikely to have occurred by chance.

Extended data

Extended Data Fig. 1 | Slope values across three elevation groups: water-formed landscapes, possible shelf, and ocean floor. (a) and (b) Column and box plots show absolute median slope values across the three elevation groups, revealing two distinct breaks at $-1,800$ m and $-3,800$ m. These patterns, together with results from the Kruskal–Wallis test (see Methods), indicate statistically significant differences between the groups.

2. The model did not independently identify the coast-to-ocean zone on Mars. L351-354
 “Flat surfaces on Mars and Earth were identified using a flat terrain angle threshold of

0.31°, selected as it represents the median slope of the zone (-1800 m to -3800 m) that includes shorelines, interpreted deltas, and distinct flat surfaces.” The manuscript contends that the automated landscape classification using this criteria matches the coastal landforms (depositional forms and shorelines), but this is by design. (See also next point).

We thank the reviewer for this important observation and the opportunity to clarify the logic of our method. We clarify that the model does identify the coast-to-ocean zone independently, using a structured two-step approach:

- The first step involves an autonomous classification of flat terrain based purely on slope (0.31° threshold), applied globally at 5-km resolution.
- The second step applies spatial refinement using independently mapped geomorphic indicators (putative shorelines) to constrain regions that plausibly preserve shelf characteristics.

The 0.31° threshold was not arbitrarily assigned based on specific landform positions but calculated as the median slope across a broad elevation zone (-1800 m to -3800 m), which includes a diversity of terrain. This threshold was then applied uniformly to the topography to classify flat areas without predefining their location relative to valleys, deltas, or shorelines.

3. There appears to be circular reasoning, as my understanding of the selection of data for analysis on Mars ensures the result.

We thank the reviewer for the detailed breakdown of our approach and carefully address each point below. We clarify that our method is not circular, but was specifically designed to separate a data-driven classification (based on slope and curvature) from the subsequent morphologic refinement. To clarify this distinction, we have created a flowchart (Fig. S1) that summarizes our two-step approach (on Earth and Mars), and we have revised the manuscript text accordingly to enhance clarity.

a. Reviewer’s summary of methods: At 5-km resolution, the martian northern lowlands is extremely flat (<0.5°). Using the median slope of the first flat zone (identified in Fig2C) within the search area (e.g., less than -1638 m, the upper elevation of martian delta, and confined to the northern lowlands), the autonomous algorithm identified the entire northern lowlands as ‘flat’. This result was then truncated to the terrain between erosional inlets (e.g., valley network) and putative shorelines (e.g., -1800 and -3800 meters). The authors selected the zone between “valley network termini, fluvial ridges, deltas, and the two proposed shorelines”. Therefore, the detected shelf in this study was by design the zone between these landforms in the northern lowlands. Incidentally, this result would be identical if the study only adopted the final step (e.g., find the region between “valley network termini, fluvial ridges, deltas, and the two proposed shorelines” in the northern lowlands, omitting the autonomous component.) Step by step details are as follows:

This is true and reflects the general flatness of the northern lowlands. The autonomous classification intentionally casts a wide net to detect flat areas. However, many of these areas—such as basin centers or degraded plains—are not consistent with shelf morphology. Hence, the

second step spatially filters the autonomous result to isolate areas coinciding with known fluvial or marine landform indicators, such as the valley networks, fluvial ridges, deltas, and the proposed shorelines.

b. The study examined the northern highlands. L506 “To identify potential shelf-oceanic zones, we focused on a region in the northern highlands (Fig. 2C) characterized by a distinct flatter zone compared to its surroundings.”

We thank the reviewer for noticing this typographical error. We confirm that the analysis was conducted in the northern lowlands, and we have corrected the manuscript accordingly. Please see below.

Methods

“For Mars, no definitive maps of oceanic features exist, aside from two long-debated proposed shorelines^{10,13}. To identify potential shelf-oceanic zones, we focused on a region in the northern lowlands (Fig. 2C) characterized by a distinct flatter zone compared to its surroundings.”

c. The authors selected the region between two candidate shorelines, Arabia and Deuteronilus.

We thank the reviewer for this comment. The Arabia and Deuteronilus shorelines were only used during the refinement phase to spatially limit the flat surface result, not to drive the classification. The autonomous step was applied to the entire northern lowlands regardless of shoreline placement.

d. L512 [The region between two debated shorelines] “is well-defined with a distinct median slope of 0.31° at a grid resolution of 5 km. We ran the algorithm at this flat angle threshold at this grid resolution and found that the entire northern lowland was marked as a relatively flat surface. To refine the results, we combined our topographic analysis (elevation, slope, and curvature) with key morphological features (valley network termini, fluvial ridges, deltas, and the two proposed shorelines).”

Thank you for raising this comment. This threshold was derived from slope statistics across the elevation band from –1800 m to –3800 m and was not tailored to match the position of any specific landforms (see the statistical analysis of the proposed zone (–1800 to –3800 m) in Extended Data Fig. 1). It reflects the characteristic slope of the broad transitional zone hypothesized to represent a shelf rather than any specific landform.

Extended Data Fig. 1 | Slope values across three elevation groups: water-formed landscapes, possible shelf, and ocean floor. (a) and (b) Column and box plots show absolute median slope values across the three elevation groups, revealing two distinct breaks at $-1,800$ m and $-3,800$ m. These patterns, together with results from the Kruskal–Wallis test (see Methods), indicate statistically significant differences between the groups.

e. L514 “To refine the results, we combined our topographic analysis (elevation, slope, and curvature) with key morphological features (valley network termini, fluvial ridges, deltas, and the two proposed shorelines). This allowed us to filter the flat surface zone between -1800 and -3800 meters, which likely preserves the transitional zone between landscapes and seascapes.” It is unclear what is meant by ‘filter’.

Thank you for this comment. We agree that “filter” was ambiguous. In the revised manuscript, we now refer to this step as a “spatial constraint informed by morphologic context” and explicitly describe its function: refining the slope-based classification to areas where geomorphic evidence (e.g., deltas, ridges, shorelines) supports a transitional setting. We have updated this section accordingly in the revised version.

Method

“For Mars, no definitive maps of oceanic features exist, aside from two long-debated proposed shorelines^{10,13}. To identify potential shelf-oceanic zones, we focused on a region in the northern lowlands (Fig. 2C) characterized by a distinct flatter zone compared to its surroundings. This region also preserves 48 deltaic systems, which are thought to have formed along an ancient ocean shoreline. Additionally, it contains numerous valley network termini and fluvial depositional ridges, both of which are believed to represent the endpoints of fluvial systems. The region also preserves the two debated shorelines. This zone is well-defined by elevation, ranging from -1800 to -3800 meters, with a distinct median slope of 0.31° at a grid resolution of 5 km. We ran the algorithm at this flat angle threshold at this grid resolution and found that the entire northern lowland was marked as a relatively flat surface. To refine the results, we combined our topographic analysis (elevation, slope, and curvature) with key morphological features (valley network termini, fluvial ridges, deltas, and the two proposed shorelines). This allowed us to

spatially constrain the flat surface zone between -1800 and -3800 meters to areas coinciding with geomorphic indicators of a landscape-to-seascape transition. On Earth, a flat angle threshold of 0.31° would detect nearly 69% to 71% of the continental shelf area, giving us a high level of confidence that the detected surface on Mars corresponds to a similar transition.”

In summary, we have revised the text for clarity and include Fig. S1, which visually separates the autonomous slope-based classification from the morphologic refinement, helping to demonstrate that our workflow does not involve circular reasoning.

Fig S1 | Conceptual workflow for identifying the topographic signature of the shelf on Earth and Mars. The flowchart outlines the stepwise approach used to define and detect shelf-like topography. On Earth, we first used terrestrial landforms (major rivers and deltas) to establish the upper bound of the shelf, followed by mapping the extent of oceanic landforms (continental shelf, slope, rise, abyssal plain, and hadal zone). We then described these landforms topographically to characterize the shelf and applied this understanding to detect and refine its expression. For Mars, we followed a similar approach using valley networks, outlet canyons, fluvial ridges, deltas, and the Arabia and Deuteronilus margins to set the upper bound of the search zone. We then analyzed the surface topography within the first 2.5 km below the delta elevations to determine whether distinct topographic metrics were present. A median slope value of 0.31° —characteristic of the $-1,800$ m to $-3,800$ m zone—was used to identify cells with similar topographic signatures. These cells were filtered to fall between the known delta elevations and the Deuteronilus boundary. Finally, we compiled sedimentologic observations of layered deposits that could potentially represent preserved shelf deposits.

4. L143: How much of the martian surface in the northern lowlands is between -1800 and -3800 m? Is it 7%? “our automated landscape classification identified 10.2 million km², or 7% of the martian surface, as a potential coastal shelf at elevations between -1800 m and -

3800 m, clustered around the dichotomy boundary.” Please clarify. Is this sentence reporting the surface area of Mars between -1800 and -3800 m elevation?

We thank the reviewer for raising this important point. To clarify, the 7% figure does not refer to the total surface area of Mars between -1800 and -3800 m. Rather, it refers specifically to the area classified by our automated method as potential coastal shelf—i.e., pixels that fall within that elevation range *and* exhibit low slopes (between 0.0° and 0.31°), consistent with shelf-like topography. The total martian surface area within the -1800 to -3800 m elevation range is indeed larger. However, our classification further filters this zone based on slope criteria derived from our topographic analysis. We have revised the sentence in the main text to clarify this distinction. Please see below.

Main

“Applying this approach to Mars at a 5-km grid resolution, our automated landscape classification identified 10.2 million km²—approximately 7% of the Martian surface—as a potential coastal shelf at elevations between -1,800 m and -3,800 m with slopes $\leq 0.31^\circ$, clustered along the dichotomy boundary (Fig. 3b).”

5. The authors discussed non-shelf cells detected in the terrestrial models (e.g., L493-504). Could they also discuss non-shelf detections on Mars. For example, Figure 3d shows significant terrain above the putative Arabia shoreline (red dots).

We thank the reviewer for this helpful observation. Some shelf-like cells detected above the putative Arabia shoreline likely reflect a combination of possibilities. In some cases, they fall within regions that also contain depositional rivers and deltas, suggesting that these elevated shelf-like detections could represent relict coastal zones formed during earlier highstands or sea-level transgressions, as observed in Aeolis Dorsa (Cardenas and Lamb, 2022). In other areas, however, these detections may simply result from degraded or eroded surfaces that retain a slope signature similar to shelf terrain but lack geomorphic coherence. It is difficult to distinguish between these possibilities given the similarity in elevation range. To address this, we have now briefly discussed these potential interpretations in the caption of Figure 3 highlighting that not all detected shelf-like cells necessarily reflect coastal shelves—they may also arise from unrelated erosional processes.

References

Cardenas, B. T., & Lamb, M. P. (2022). Paleogeographic reconstructions of an ocean margin on Mars based on deltaic sedimentology at Aeolis Dorsa. *Journal of Geophysical Research: Planets*, 127(10). <https://doi.org/10.1029/2022je007390>

Main Text

Fig. 3 | Shelf detection using the Geomorphons function on both Earth (a) and Mars (b), alongside mapped shorelines on Mars (b) and mapped shelves on Earth (a). Flat surfaces on Mars and Earth were identified using a flat terrain angle threshold of 0.31° (see Methods), selected as it represents the median slope of the zone (-1800 m to -3800 m) that includes shorelines^{10,13}, interpreted deltas¹⁰, and distinct flat surfaces. These panels illustrate the comparison between the detected shelf on Earth and the mapped shelf², as well as the potential

martian shelf (-1800 to -3800 meters) and the proposed shorelines^{10,13}. (c) and (d) show zoomed-in sections highlighting the mapped and detected shelves as well as the proposed shorelines. The black lines in both sections represent the Rhine River and Hypanis Valles, extending continuously to the deep ocean. Profiles are shown in Fig. 4. Pixels above the putative Arabia shoreline share the same topographic characteristics as classified shelf terrain. These may represent earlier shoreline positions formed during highstands or transgressions, or they may simply be degraded or eroded surfaces that retain similar slope signatures.

6. L491: “We applied different flat terrain angle thresholds for the two planets.” What value for the flat terrain threshold angle was used for Mars? L 139-143 implies that 0.31° for 5 km grid resolution was used. However, this appears to be the same value used for Earth (L352)?

Thank you for this question. Yes, the slope threshold used to detect shelf-like cells on Mars was 0.31°. On Earth, we conducted 40 experiments using a range of slope thresholds from 0.01° to 1.22°, which allowed us to fully capture the extent of the terrestrial shelf. Below, we include a table summarizing the results of these experiments, along with plots showing the relationship between slope threshold, detected area, and shelf detection accuracy. We also provide examples from selected experiments to illustrate the approach.

Extended Data Fig. 2 | Examples of the Geomorphons algorithm output and its comparison to the mapped shelf⁶². The sky-blue color represents cells with a maximum slope equal to the specified value mentioned on top of each output. The figures below illustrate the correspondence between the sky-blue cells and the mapped shelf.

Extended Data Fig. 3 | Results from 40 experiments aimed at constraining the accuracy of shelf versus non-shelf detection. (a) The experiments show that increasing the flat terrain angle threshold leads to improved detection rates for both shelf and non-shelf areas. (b) Comparing the ratio of detected shelf to mapped shelf at various angle thresholds allows for an assessment of detection accuracy. The terrestrial shelf was fully detected (100%) at an angle of 1.22°. But, a flat angle threshold of 0.31° detected approximately 71% of the shelf. (c) Comparison of the percentage of the detected shelf with the percentage of the shelf that was not detected.

Extended Data Table 1 | Trial and error using Geomorphons terrain classification for different maximum terrain angles. Results show the terrain area (at ~1.6 x ~1.6 km grid cell resolution) that was correctly and incorrectly identified as continental shelf, as defined by previous mapping.

Run No.	Maximum terrain angle (°)	Detected area at specific terrain angle (km ²)	Total area of detected cells that correspond to the mapped shelf (km ²)	Percent of areas correctly identified	Area of detected cells that do not correspond to the mapped shelf (km ²)	Percent of areas that do not correspond to the shelf	Areas of the shelf that were not detected (km ²)	Areas of the shelf that were not detected
1	0.01	953693	477853	1.5	475840	49.9	31764692	98.5
2	0.02	3141193	1655086	5.1	1486107	47.3	30587459	94.9
3	0.03	6204915	3187174	9.9	3017741	48.6	29055371	90.1
4	0.04	9603489	4806557	14.9	4796931	49.9	27435988	85.1
5	0.05	12925097	6288288	19.5	6636809	51.3	25954257	80.5
6	0.06	16193424	7660595	23.8	8532829	52.7	24581950	76.2
7	0.07	19398629	8909908	27.6	10488721	54.1	23332637	72.4
8	0.08	22586358	10063033	31.2	12523325	55.4	22179512	68.8
9	0.09	25741303	11111961	34.5	14629342	56.8	21130584	65.5
10	0.1	28866211	12072680	37.4	16793531	58.2	20169865	62.6
11	0.11	31979441	12960441	40.2	19019000	59.5	19282104	59.8
12	0.12	35078577	13779876	42.7	21298702	60.7	18462669	57.3
13	0.13	38163289	14532814	45.1	23630475	61.9	17709731	54.9
14	0.14	41232690	15233392	47.2	25999299	63.1	17009153	52.8

15	0.15	44304544	15891198	49.3	28413346	64.1	16351347	50.7
16	0.16	47369985	16508641	51.2	30861344	65.1	15733904	48.8
17	0.17	50430628	17091810	53.0	33338818	66.1	15150735	47.0
18	0.18	53474889	17641712	54.7	35833177	67.0	14600833	45.3
19	0.19	56515464	18162593	56.3	38352871	67.9	14079952	43.7
20	0.2	59524231	18650898	57.8	40873334	68.7	13591647	42.2
21	0.21	62535445	19118544	59.3	43416901	69.4	13124001	40.7
22	0.22	65532194	19564504	60.7	45967690	70.1	12678041	39.3
23	0.23	68505830	19987388	62.0	48518442	70.8	12255157	38.0
24	0.24	71475981	20394120	63.3	51081862	71.5	11848425	36.7
25	0.25	74414043	20780235	64.4	53633808	72.1	11462310	35.6
26	0.26	77341749	21149241	65.6	56192507	72.7	11093304	34.4
27	0.27	80245679	21500839	66.7	58744840	73.2	10741706	33.3
28	0.28	83131205	21838059	67.7	61293145	73.7	10404486	32.3
29	0.29	85991447	22161227	68.7	63830220	74.2	10081318	31.3
30	0.3	88836570	22473059	69.7	66363511	74.7	9769486	30.3
31	0.31	91649737	22771879	70.6	68877858	75.2	9470666	29.4
32	0.33	97210900	23332144	72.4	73878783	76.0	8910401	27.6
33	0.35	102691427	23852540	74.0	78838887	76.8	8390005	26.0
34	0.4	115990743	24999725	77.5	90991018	78.4	7242820	22.5
35	0.6	163671127	28127836	87.2	135543291	82.8	4114709	12.8
36	0.8	203990151	30013010	93.1	173977141	85.3	2229535	6.9
37	1	238151117	31279267	97.0	206871850	86.9	963278	3.0
38	1.2	267078221	32191561	99.8	234886660	87.9	50984	0.2
39	1.21	268406667	32231259	100.0	236175408	88.0	11286	0.0
40	1.22	269721337	32270206	100.1	237451131	88.0	-27661	-0.1

7. The second topographic minima (L121/122) is not mentioned again or discussed. Presumably, this is because the elevations between -3800 to -5000 m were filtered out (L514).

We thank the reviewer for this careful observation. Yes, the second minima—identified between -3800 m and -5000 m—was not the focus of further discussion because it falls outside the elevation range used for shelf classification. As noted in Line 514, our method filters out elevations below -3800 m to avoid including deep-basin floors that may represent unrelated geomorphic domains (e.g., crater interiors, volcanic plains, or depositional sinks) with similarly low slopes but no shoreline relevance.

8. The author's could consider applying the same technique to other planetary bodies. Titan would offer another test of a topographic signature from an ocean. Oceanless worlds like the moon or Venus could be used to evaluate the robustness of this approach.

We thank the reviewer for this excellent suggestion to test our method on other planetary bodies. As an internal comparison within Mars, we applied the same topographic classification to Hellas Basin—the second-largest basin on the planet after the northern lowlands. Hellas is an ideal test case: it is a large, well-preserved impact structure that has been extensively modified by processes unrelated to a standing ocean, and it benefits from the same uniform, high-resolution MOLA digital elevation model as the northern lowlands.

Importantly, Hellas and the northern lowlands have relatively similar formation ages, which implies broadly comparable surface histories and climate conditions. This makes Hellas an appropriate null-hypothesis test for evaluating whether the topographic signature we observe is unique to the northern lowlands. Unlike the lowlands, Hellas does not exhibit a similar pattern of coupled slope and curvature minima, supporting our interpretation that the northern lowlands host a distinctive morphologic imprint potentially linked to a former ocean.

We agree that extending this method to other bodies such as Titan, Venus, or the Moon is a promising avenue for future work. However, these bodies present key limitations. Titan’s global digital elevation model has a resolution of ~45 km per pixel—about 90 times coarser than MOLA—and only ~9% of the elevation values come from direct measurements (Corlies et al., 2017); the rest are interpolated. Venus faces similar constraints, with ~13% of data from direct measurements and the remainder derived from interpolation (Ford and Pettengill, 1992). Such low-resolution, sparsely measured datasets would introduce significant noise and limit the reliability of a slope- and curvature-based classification.

Given these constraints and the absence of hypothesized global oceans on the Moon or Venus, we chose to focus our analysis on Mars. We have now clarified this reasoning in the Methods section and highlight the Hellas Basin test as a null-hypothesis evaluation of the northern lowlands' distinctiveness. In response, we also created a new figure and added a dedicated subsection in the Methods describing this control analysis.

In addition, we created a new figure (Fig. 5) showing more than 14 000 occurrences of layered deposits—up to 500 m thick—that are spatially consistent with both our detected shelf-like cells and the clay stratigraphy. These deposits provide potential sedimentologic evidence supporting the presence of past water and depositional environments within the interpreted shelf zone. The figure also highlights clay mineral-bearing stratigraphies within the same elevation range, further reinforcing the interpretation of a watery and sediment-rich setting. Finally, we show that the elevations of these layered deposits and open-basin deltas align well with our proposed regression–transgression pattern, strengthening the case for shoreline fluctuations within this zone.

References

Ford, P. G. & Pettengill, G. H. Venus topography and kilometer-scale slopes. *Journal of Geophysical Research: Planets* **97**, 13103–13114 (1992).

Corlies, P. *et al.* Titan’s topography and shape at the end of the cassini mission. *Geophysical Research Letters* **44**, (2017).

Methods

Testing the null hypothesis: impact basin control

To assess whether the stepped slope morphology observed in the northern lowlands is a unique topographic signature of that region or a generic consequence of large impact basin formation modified by processes such as fluvial activity⁶², volcanic resurfacing⁶⁷, or prolonged burial and erosion³⁶, we conducted a parallel topographic analysis of Hellas Basin—the second largest basin on Mars after the northern lowlands. We selected Hellas for its large scale and state of preservation, though it lacks clear evidence for a standing ocean and has been extensively modified by burial, exhumation, volcanic, and fluvial processes. We applied the same analytical framework as in the northern lowlands: a uniform grid at 5 km per pixel resolution, with elevation values binned every 200 m. For each bin, we calculated the median slope and tested for statistically significant slope breaks using the Kruskal–Wallis H-test. This consistent methodology enables direct comparison of elevation-dependent morphologic transitions between basins. If shelf-like slope signatures were solely the result of post-impact erosional and depositional processes, similar features would be expected in both basins. However, no comparable slope break was observed in the control basin (Extended Data Fig. 10), suggesting that the features in the northern lowlands reflect modification by additional processes—such as long-lived oceans^{1,2,3,4,5,6,7,8,9,10,11,12,13,14,15} and deposition during sea-level fluctuations—rather than impact alone^{45,47}.

Extended Data Fig. 10 | Null hypothesis test using Hellas Basin. (a) Topography of Hellas Basin, the second-largest impact basin on Mars, overlaid with valley networks (black)^{34,62}, open-

basin lakes (blue)³⁴, and outlet canyons (orange)³⁴. (b) and (c) Median slope values per 200 m elevation bin show no evidence of abrupt, shelf-like transitions similar to those identified in the northern lowlands. The distribution also lacks the coupled minima observed in the ocean-bearing hypothesis, which would be shelf and ocean or large sea floor. These results reinforce that the morphometric signal detected in the northern lowlands is not a generic outcome of large-basin formation or surface degradation but may instead reflect a distinct process, such as ocean-related resurfacing.

Fig. 5 | Sedimentologic and mineralogic evidence for sedimentary rocks, clay stratigraphy, and open-basin deltas along and within the detected shelf. (a) and (b) Polar stereographic projections of the detected shelf showing 14,386 mounds interpreted to have formed by dichotomy retreat, consisting of thick layered deposits up to 500 m in southern Chryse Planitia, Mars, along with widespread clay-bearing stratigraphy and open-basin deltas. (c) and (d) Box plots and histograms of open-basin delta elevations show that most deltas are built atop the detected shelf and are consistent in elevation with the layered deposits. The blue gradient—horizontal in c and vertical in d—indicates the range of interpreted sea-level changes, based on evidence of transgression and regression from two large deltaic systems in Aeolis Dorsa (e), particularly Hypanis (f). Background map is from MOLA.

Clarifications for Discussion. The subtle topographic expression investigated is not uniquely tied to process, and alternative explanations should be discussed.

We have addressed the alternative hypothesis in detail, as explained in the revised manuscript. For the remaining points, we provide a point-by-point response below. Thank you.

1. Can the authors discuss the impact of Tharsis on their data analysis?

- a. Lowlands on several planetary surfaces are due to volcanism (e.g., moon, Venus). Furthermore, volcanic flows into the martian lowlands is well documented. In the 2005 USGS geologic map of the northern plains, “Apparently formed early in Mars' history, the northern lowlands served as a repository both for sediments shed from the adjacent ancient highlands and for volcanic flows and deposits from sources within and near the lowlands.” <https://pubs.usgs.gov/sim/2005/2888/>
- b. Furthermore, there are no mapped valley networks along a considerable stretch of the proposed shelf (e.g. roughly longitude 220-320° E; Fig. 1C).

Thank you for raising this important concern. In response, we have added a dedicated section discussing the limitations and uncertainties of our analysis. In this section, we specifically address the potential influence of the Tharsis rise on regional topography and how it may affect our results. We also discuss additional sources of uncertainty, including the effects of burial, exhumation, and erosion, which could obscure or modify the original shelf-like signals. These additions can be found in the revised manuscript and are also summarized below.

Methods

Limitations and uncertainties

“Our results are subject to several uncertainties. One is the potential alteration of topography due to true polar wander and the emplacement of the Tharsis volcanic province, which likely caused uplift near Tharsis and subsidence farther away^{11,12}. However, analyzing the surface as geomorphic domains helps mitigate this, since such deformation would affect broad regions rather than the specific elevation ranges of landform mosaics. A second source of uncertainty is the isostatic response to ocean unloading, which on Earth can modify elevations by several hundred meters following ocean retreat. However, recent estimates for Mars suggest that isostatic rebound likely ranged from several tens to just over one hundred meters⁷⁸. However, recent estimates for Mars suggest that isostatic rebound likely ranged from several tens to just over one hundred meters¹². Yet, the ~2 km elevation span of our detected shelf-like zone exceeds expected rebound estimates and remains consistent with depositional features. A third source of uncertainty is long-term burial, exhumation, and erosion³⁹. While these processes may have introduced regional variability, they are unlikely to alter the broader topographic patterns we identify at the global scale.

2. L129 reports the elevation of many landforms associated with coasts are “at elevations above the putative coastal shelf, as expected.” However, this result is by design. The study adopted a threshold elevation based on these same landforms (L89-101).

Thank you for this comment. We would like to clarify that the slope threshold used to detect shelf-like terrain was not based on any landform data. It was derived solely from topographic parameters, specifically the identification of slope and curvature minima across the elevation range. Only after detecting these topographic zones did we compare their elevation range with the distribution of coastal landforms (e.g., deltas and ridges) to assess consistency.

To clarify our methodology, we have now added a flowchart illustrating the workflow—from topographic classification to morphological validation. This figure is included below and in the supplementary file.

Fig S1 | Conceptual workflow for identifying the topographic signature of the shelf on Earth and Mars. The flowchart outlines the stepwise approach used to define and detect shelf-like topography. On Earth, we first used terrestrial landforms (major rivers and deltas) to establish the upper bound of the shelf, followed by mapping the extent of oceanic landforms (continental shelf, slope, rise, abyssal plain, and hadal zone). We then described these landforms topographically to characterize the shelf and applied this understanding to detect and refine its expression. For Mars, we followed a similar approach using valley networks, outlet canyons, fluvial ridges, deltas, and the Arabia and Deuteronilus margins to set the upper bound of the search zone. We then analyzed the surface topography within the first 2.5 km below the delta elevations to determine whether distinct topographic metrics were present. A median slope value of 0.31°—characteristic of the -1,800 m to -3,800 m zone—was used to identify cells with similar topographic signatures. These cells were filtered to fall between the known delta elevations and the Deuteronilus boundary. Finally, we compiled sedimentologic observations of layered deposits that could potentially represent preserved shelf deposits.

3. L190 Given the subtle signature of putative oceans, what potential evidence might rovers observe to validate or repudiate the hypothesis? Can you elaborate?

We thank the reviewer for raising this comment. We have expanded the relevant section to clarify what a rover could observe from a sedimentological perspective. Please see the revised text in the section below.

Main text

“Continental margins are major planetary sinks for sediment and carbon and account for much of Earth’s sedimentary record. This record has shaped our understanding of the evolution of climate, tectonics, and life over geologic time⁵⁸. By analogy, the proposed martian coastal shelf represents a critical target for future exploration. The deposits within this shelf may preserve a record of Mars’ geologic, oceanic, and environmental history. The coastal shelf hypothesis is supported by recent in-situ findings of coastal-like deposits⁵³, massive layered sequences along the detected shelf^{55,56}, and widespread clay-bearing stratigraphy along the dichotomy⁵⁷. Additional support may come from future missions, including the European Rosalind Franklin rover, scheduled to land in Oxia Planum in 2030. A rover investigating a coastal shelf might encounter centimeter-scale planar laminations and beds formed by oscillatory wave processes, cross-stratification from unidirectional (storm) and oscillatory (wave) flows, wave ripples, well-sorted fine sand to silt, large clinforms, and laterally continuous layering^{59,54}. More broadly, our results suggest that ancient oceans on presently arid planets may be best identified not only through discrete shoreline features but also through fluvio-deltaic depositional systems and low-slope, low-curvature topographic zones.

Terminology.

1. Please define outlet valleys (this terminology could be misconstrued with the well established martian term ‘outlet channels’).

Thank you for the suggestion. In response, we have revised the terminology to “outlet canyons” and provided a definition in the manuscript to ensure clarity and consistency.

Main text

“Similar to Earth, Mars hosts valley networks and outlet canyons—formed by lake-breach floods—that eroded and transported sediments from upstream sources (Fig. 1d).”

2. The abstract states that the topographic feature on Earth is due to the coastal plains and the continental shelf (L16), yet reference to coastal plains is minimal in the text and absent in the figures.

Thank you for pointing this out. We agree with the reviewer that coastal plains are not explicitly shown in the figures and are only briefly mentioned in the text. This is because there is no consistent global dataset for coastal plains, and it is topographically challenging to distinguish coastal plains from continental shelves using slope and curvature metrics alone. In most cases, the transition from coastal plain to shelf is gradual and often masked by sedimentation or sea-level changes.

3. It would be useful to definition of depositional rivers

Thank you for this helpful suggestion. We have now clarified the terminology by defining *depositional ridges* (also referred to as *fluvial ridge systems*) in the main text. These features represent inverted channels or river deposits that have been preserved in positive relief due to differential erosion, and they serve as geomorphic evidence of past sediment-laden flows and deposition.

Main

“Deposition zones are expressed as fluvial ridges (ridges preserve fluvial deposits)³⁵, which terminate abruptly and patchily at -2,577 meters, marking the 90th percentile upper limit. Open-basin deltas along the northern dichotomy range in elevation from -1,638 to -2,880 meters^{4,10}.”

Figures.

1. As they hypothesized martian ocean is exclusively in the northern hemisphere, the author’s are encouraged to render there data in a polar stereographic projection to enhance the presentation (extended data figure). As an example, see Figure 1 in Sholes et al., 2021 (<https://doi.org/10.1029/2020JE006486>).

Thank you for suggesting this excellent point. In the revised version, we included examples of the detected shelf rendered in polar stereographic projection, particularly in cases where no direct comparison with Earth is made, as such projection would compromise the visual clarity of Earth–Mars comparisons. Additionally, for some examples, we aimed to highlight terrestrial water-formed landscapes, which would be difficult to effectively present using a polar stereographic projection. Figure 5 is one such example, where we used the polar stereographic projection to focus on the Martian shelf without comparison to Earth.

Fig. 5 | Sedimentologic and mineralogic evidence for sedimentary rocks, clay stratigraphy, and open-basin deltas along and within the detected shelf. (a) and (b) Polar stereographic projections of the detected shelf showing 14,386 mounds interpreted to have formed by dichotomy retreat, consisting of thick layered deposits up to 500 m in southern Chryse Planitia, Mars, along with widespread clay-bearing stratigraphy and open-basin deltas. (c) and (d) Box plots and histograms of open-basin delta elevations show that most deltas are built atop the detected shelf and are consistent in elevation with the layered deposits. The blue gradient—horizontal in c and vertical in d—indicates the range of interpreted sea-level changes, based on evidence of transgression and regression from two large deltaic systems in Aeolis Dorsa (e), particularly Hypanis (f). Background map is from MOLA.

3. Figure 1a: What are the light blue polygons? Watersheds? Figure 1c: It is difficult to distinguish the green and blue lines (valley networks and outlet valleys) on the grey background. The small dots on the highlands are presumably depositional rivers, but the interior color of the circle is hard to discern.

Thank you. Revised. Please see the version below.

4. Figure 2c: The area labeled possible shelf appears to extend beyond -1800 m. Is that intentional? Perhaps you mean search zone for shelf. At any rate the distinction between ‘possible shelf’ in this figure and ‘probable shelf’ in later figures may cause confusion if these are indeed meant to have different meanings. Note that Figure 4 adopts ‘possible shelf’.

Thank you for this careful and thoughtful observation. This was done intentionally, as the vertical bars represent the 10th to 90th percentile elevation range of the identified landforms. We have clarified this point in the revised figure caption—please see below. For the confusion between possible and probable, we adopted possible (please see below). Thanks again.

Main

Fig. 2 | Slope and curvature of the entire surfaces of Earth (a and b) and Mars (c and d). The figure shows a clear decline in both topographic parameters toward the continental shelf on Earth, particularly within the first 200 meters. In contrast, Mars exhibits a different pattern, with two distinct zones marked by declines in both slope and curvature: one between -1800 m and -3800 m, and another between -3800 m and -5000 m. The presence of fluvial ridges, deltas, and two shorelines within the first zone (-1800 m to -3800 m) suggests that this is likely a transitional

zone between landscape and seascape. The data were computed from the ETOPO Global Relief Model for Earth and the MOLA data for Mars (see Methods for details). Uncertainties correspond to the 5–95% confidence interval for the median. The vertical bars in the background represent the 10th to 90th percentile elevation range of the identified landforms, as shown in Fig. 1. Since curvature includes both positive and negative values, we multiplied the negative values by (-1) to improve interpretability, especially when comparing concave and convex surfaces (with negative values color-coded in green for Earth and orange for Mars).

Fig. 3

5. Figure 4c: How were the extents of the profiles determined? More specifically, why is the Hypanis profile so long and the Arabia Dorsa profile so short? Figure 4d: Consider moving or changing label ('shoreline') for Arabia and Deuteronilus, as these refer to both putative shorelines as well as geographic regions on Mars.

Thank you for this comment. The purpose of including the profiles was to illustrate elevation changes across different erosional and depositional environments and to highlight variations in topography. We selected profiles that are spatially close on both Earth and Mars and that capture a range of erosional and depositional landforms. The Arabia profile is shorter than the Hypanis profile because, unlike Hypanis, which extends continuously toward the ocean floor, the Arabia profile is interrupted by the presence of Elysium Mons, which prevents it from reaching the ocean floor.

We also changed the position of the putative shorelines. Please see the revised version below.

Main

Minor comments:

L 353: To clarify, suggest adding the word martian before zone.

Fixed. Thank you.

L507: Do you mean northern lowlands?

Yes, thank you. Addressed.

Almost the entire northern highlands is encompassed within the two zones identified in this study (-1800 to -5000 m; L119-122).

Yes, that is correct. The elevation range from -1800 to -5000 m does encompass most of the northern lowlands.

Fig.2 caption heading needs revised. “Slope and curvature of the entire surfaces of Earth and Mars.” However, martian surface from +2 to + 8 km is excluded from the plots.

Thank you for raising this point. We confirm that we analyzed the entire surfaces of both Earth and Mars. However, in the plots shown in Fig. 2, we focused our binning and detailed presentation on the elevation ranges most relevant to our shelf analysis. The Martian elevations between +2 km and +8 km were not binned in detail, as they do not contribute to the objectives of this study.

Referee #2 (Remarks to the Author):

The manuscript by Zaki and Lamb proposes a bold new hypothesis: that the topography surrounding the northern lowlands on Mars contains a feature analogous to the continental shelves on Earth, and that this shelf-like feature formed around the margins of an ancient ocean. The idea that Mars once had an ocean is not new, but most previous studies that tested this idea have focused on possible shorelines or small coastal landforms (such as river deltas) that might have formed in the vicinity of shorelines, both of which are challenging to interpret because they are susceptible to degradation and alteration. The novelty of the Zaki and Lamb hypothesis is that it focuses on a broader-scale topographic feature that is more likely to have persisted over the billion+ years since the ocean would have dried up. The analysis does not demonstrate with complete certainty that Mars had an ocean, but it is the most substantial and creative contribution to the Mars ocean hypothesis in years, and it is bound to attract a lot of interest. It is also timely, given the recent discovery of possible coastal sedimentary deposits on Mars by China's Zhurong rover (Li et al. PNAS 2025).

I have two general comments, numbered below, followed by a set of comments related to specific passages in the paper. I think the authors should consider these comments before the paper is published.

We thank the reviewer for their thoughtful and encouraging summary of our manuscript. We greatly appreciate the recognition of the novelty and broader significance of our approach to evaluating the Mars ocean hypothesis through persistent topographic signatures. Our goal is to offer a complementary perspective to prior shoreline-focused studies by leveraging large-scale topographic metrics that are less susceptible to post-depositional modification. We are also pleased that the relevance of this work in the context of recent findings from the Zhurong rover was noted.

Below, we respond point by point to the reviewer's general and specific comments, and we have made revisions throughout the manuscript to address the concerns raised.

General comments:

1. Null hypothesis.

On Earth, the bathymetry of the continental shelf presumably reflects the ocean-continent contrast in crustal density and thickness, combined with sedimentation and wave modification during dozens of glacial sea-level lowstands. In the likely absence of both major factors on Mars, should we expect a similar shelf to form? The authors acknowledge these differences and note (lines 125-127) that a huge impact probably created the dichotomy in crustal thickness on Mars (Andrews-Hanna et al. 2008). The null hypothesis, then, should be that any topographic characteristic of the northern lowlands is a product of the impact. The authors should explain why an ocean is a better explanation for the topographic feature they identified than an impact.

We thank the reviewer for this insightful comment, which encouraged us to evaluate the null hypothesis more thoroughly. In response, we addressed this question at multiple levels:

1. **Comparison with another large impact basin (Hellas):** We applied the same slope and curvature analysis to the Hellas Basin—the second-largest impact structure on Mars after the northern lowlands. Hellas is similar in scale and has undergone modification by a range of processes including fluvial activity, volcanism, burial, and erosion. Its elevation range also overlaps with the northern lowlands (see Extended Data Fig. 10). However, Hellas does not exhibit the coupled slope and curvature minima that we observe in the northern lowlands. This absence suggests that the unique morphometric signal we identify is not a general consequence of large impact basins or post-impact surface evolution.
2. **Sedimentologic and mineralogic evidence consistent with shelf deposition:** We compiled a dataset of over 14,000 occurrences of layered deposits up to 500 m thick across the northern lowlands. Many of these deposits are spatially consistent with the classified shelf zone. Furthermore, multiple sites within this zone also host clay-bearing stratigraphies, which point to persistent aqueous conditions and sediment accumulation.
3. **Delta and shoreline stratigraphic relationships:** We show that major deltas—such as those in Aeolis Dorsa and Hypanis—were emplaced at or near the elevation range of the detected shelf and were often constructed atop the thick layered deposits. The morphostratigraphic relationships in these systems are consistent with shoreline dynamics involving transgression and regression, which may have contributed to the buildup of a coastal shelf.

Taken together, these lines of evidence suggest that the topographic signal we detect is not solely a relic of the impact that formed the northern lowlands. Rather, it likely records a later overprint from sedimentary and hydrologic processes consistent with the presence of a long-lived ocean. We have clarified this point in the revised manuscript and included supporting data and figures to strengthen this argument.

Methods

Testing the null hypothesis: impact basin control

To assess whether the stepped slope morphology observed in the northern lowlands is a unique topographic signature of that region or a generic consequence of large impact basin formation modified by processes such as fluvial activity⁶², volcanic resurfacing⁶⁷, or prolonged burial and erosion³⁶, we conducted a parallel topographic analysis of Hellas Basin—the second largest basin on Mars after the northern lowlands. We selected Hellas for its large scale and state of preservation, though it lacks clear evidence for a standing ocean and has been extensively modified by burial, exhumation, volcanic, and fluvial processes. We applied the same analytical framework as in the northern lowlands: a uniform grid at 5 km per pixel resolution, with elevation values binned every 200 m. For each bin, we calculated the median slope and tested for statistically significant slope breaks using the Kruskal–Wallis H-test. This consistent methodology enables direct comparison of elevation-dependent morphologic transitions between basins. If shelf-like slope signatures were solely the result of post-impact erosional and depositional processes, similar features would be expected in both basins. However, no comparable slope break was observed in the control basin (Extended Data Fig. 10), suggesting that the features in the northern lowlands reflect modification by additional processes—such as long-lived

oceans^{1,2,3,4,5,6,7,8,9,10,11,12,13,14,15} and deposition during sea-level fluctuations—rather than impact alone^{45,47}.

Extended Data Fig. 10 | Null hypothesis test using Hellas Basin. (a) Topography of Hellas Basin, the second-largest impact basin on Mars, overlaid with valley networks (black)^{34,62}, open-basin lakes (blue)³⁴, and outlet canyons (orange)³⁴. (b) and (c) Median slope values per 200 m elevation bin show no evidence of abrupt, shelf-like transitions similar to those identified in the northern lowlands. The distribution also lacks the coupled minima observed in the ocean-bearing hypothesis, which would be shelf and ocean or large sea floor. These results reinforce that the morphometric signal detected in the northern lowlands is not a generic outcome of large-basin formation or surface degradation but may instead reflect a distinct process, such as ocean-related resurfacing.

Main

“We investigated whether a global ocean drying up would leave a topographic footprint at planetary scale. Our global topographic analysis of elevation, slope, and curvature shows that the transition from land to sea on Earth is not limited to a single distinct shoreline, but also includes a broader zone of low slope and curvature near sea level. This morphometric transition complements, rather than replaces, the geomorphic features typically associated with coastlines, such as shorelines and deltas. We presented evidence for a similar topographic zone on Mars, which is bounded at high elevations by deltaic landforms and depositional rivers, and encompasses previously proposed shorelines. Based on this evidence, we propose that Mars has an ancient coastal shelf from -1800 to -3800 m elevation that once bounded a global ocean (Figs. 2, 3, and 4). Alternative explanations for the observed slope–curvature break—such as post-impact relaxation, volcanic resurfacing, or fluvial erosion and deposition—are evaluated (see Methods; Extended Data Fig. 10), but none reproduce the coherent stepped morphology and spatial consistency of the shelf-like transition as plausibly as long-term modification by a standing body of water.”

Fig. 5 | Sedimentologic and mineralogic evidence for sedimentary rocks, clay stratigraphy, and open-basin deltas along and within the detected shelf. (a) and (b) Polar stereographic projections of the detected shelf showing 14,386 mounds interpreted to have formed by dichotomy retreat, consisting of thick layered deposits up to 500 m in southern Chryse Planitia, Mars, along

with widespread clay-bearing stratigraphy and open-basin deltas. (c) and (d) Box plots and histograms of open-basin delta elevations show that most deltas are built atop the detected shelf and are consistent in elevation with the layered deposits. The blue gradient—horizontal in c and vertical in d—indicates the range of interpreted sea-level changes, based on evidence of transgression and regression from two large deltaic systems in Aeolis Dorsa (e), particularly Hypanis (f). Background map is from MOLA.

2. Post-ocean topographic change.

The authors assume implicitly that Mars' long-wavelength topography has not changed substantially since an ocean filled the northern basin. The mechanisms of subsequent erosion and sediment redistribution they mention (wind erosion, meteor impacts, and regolith creep; lines 124-125) are mostly local and would not have altered the large-scale topography much. However, geophysical mechanisms could have modified the large-scale topography of an ocean basin, including any shelf. True polar wander (Perron et al. 2007; Citron et al. 2018), dynamic topography, or deformation associated with the removal of the ocean water load could have generated long-wavelength topographic change. Even if the authors disagree with previous suggestions of true polar wander or dynamic topography, they should consider the isostatic and flexural response to removal of the ocean load, because it must have occurred if there was an ocean. The authors are comparing the bathymetry of Earth's oceans (at a sea-level highstand) with an empty basin on Mars. How would Earth's ocean bathymetry change if the water was all removed?

We thank the reviewer for this excellent and important point. We agree that it is important to address how geophysical mechanisms might have altered the topography. In response, we have added a dedicated section to the manuscript discussing the limitations and uncertainties of our analysis, including the potential for large-scale geophysical modifications that may have occurred after the ocean dried up.

Specifically, we now discuss the potential influence of true polar wander, dynamic topography, and the isostatic or flexural response to ocean unloading. While the magnitude and direction of these effects on Mars remain debated, we agree that any long-lived ocean would have imposed a significant load, and its removal could have resulted in vertical deformation.

However, several factors suggest that the shelf-like signature we identify may have persisted despite such post-ocean adjustments:

1. **Expected magnitude of deformation:** Studies such as Citron et al. (2018) and Perron et al. (2007) suggest that post-ocean loading effects would produce vertical shifts on the order of hundreds of meters. Given that the shelf zone spans ~2,000 m of elevation (from -1800 to -3800 m), the observed signal is broader than the expected vertical deformation alone. We have plotted the putative shoreline of Arabia (both modern and ancient topography, which shows that the vertical changes are in the range of 100s meters). It is based on data retrieved from Sholes and and Rivera-Hernandez, 2021.

2. **Comparative resilience:** If a similar topographic signal had existed solely due to the formation of the basin or geophysical relaxation, we would expect to observe analogous features in other large basins such as Hellas. However, our null test in Hellas Basin did not reveal comparable shelf-like slope or curvature minima, suggesting the northern lowlands host a unique and possibly depositional signature.
3. **Dry basin analogs on Earth:** We acknowledge that our analysis compares the present-day dry basin of Mars to bathymetric shelves on Earth formed under a standing sea. However, even if Earth's oceans were removed, the isostatic rebound would not erase the lateral extent or slope break of continental shelves—they would remain recognizable due to the inherited topographic gradient, thick sedimentary sink, and underlying lithospheric structure.

We now include these considerations in the revised discussion and limitations section, and we have clarified that while the shelf signal may have been modified over time, its scale and consistency with depositional features support the interpretation of a long-lived ocean margin.

Methods

Limitations and uncertainties

Our results are subject to several uncertainties. One is the potential alteration of topography due to true polar wander and the emplacement of the Tharsis volcanic province, which likely caused uplift near Tharsis and subsidence farther away^{11,12}. However, analyzing the surface as geomorphic domains helps mitigate this, since such deformation would affect broad regions rather than the specific elevation ranges of landform mosaics. A second source of uncertainty is the isostatic response to ocean unloading, which on Earth can modify elevations by several hundred meters following ocean retreat. However, recent estimates for Mars suggest that isostatic rebound likely ranged from several tens to just over one hundred meters⁷⁸. However, recent estimates for Mars suggest that isostatic rebound likely ranged from several tens to just

over one hundred meters¹². Yet, the ~2 km elevation span of our detected shelf-like zone exceeds expected rebound estimates and remains consistent with depositional features. A third source of uncertainty is long-term burial, exhumation, and erosion³⁹. While these processes may have introduced regional variability, they are unlikely to alter the broader topographic patterns we identify at the global scale.”

Comments on specific passages:

118-125: I’m having trouble seeing two minima in Figure 2b,d. The features at -3800m look more like a step down. Perhaps the box-and-whisker plot is not the most effective way to show this upper feature. I note that the contrast between the left and right columns in Fig. 2 makes the conceptual model in Figure 4 important: The evolution of a shelf-like feature on Mars may have differed from Earth, resulting in a more gradational transition from land to ocean basin (Fig. 4d).

Thank you for raising this excellent comment. To improve clarity, we have now plotted the median slope and curvature values on a linear (rather than logarithmic) scale in a supplementary figure, which more clearly shows the stepped-down pattern near -3800 m.

Additionally, to test the significance of the observed breaks, we conducted a Kruskal–Wallis H-test—a non-parametric statistical test suitable for comparing more than two independent groups without assuming normality. We grouped the data into three elevation intervals: (1) above -1800 meters, (2) from -1800 to -3800 meters, and (3) below -3800 meters. The test yielded a p-value of 1.07×10^{-6} , indicating a highly significant difference in median slope and curvature values between these zones. This result strongly supports the presence of statistically distinct topographic regimes, reinforcing our interpretation of a stepped transition.

We have incorporated this statistical analysis into the revised manuscript and updated the main text, Methods section, and Extended Data Figure 3 accordingly.

Main text

“The same analysis of Mars’ topography does not reveal a single distinct minimum in global slope and curvature, as observed on Earth (Fig. 2c, d; Extended Data Fig. 2). Instead, two coupled minima in slope and curvature are observed on Mars (Extended Data Fig. 3), supported by a Kruskal–Wallis H-test ($p = 1.07 \times 10^{-6}$), indicating less than a 0.00011% probability that the differences in median slope values among the observed elevation ranges occurred by chance. The first minimum lies between approximately -1800 m and -3800 m (slope = 0.31° , curvature = 0.00015, -0.00017), and the second between -3800 m and -5000 m (slope = 0.12° , curvature = 0.0009, -0.0009). The minima are less distinct than on Earth, but this might be expected, given that Mars’ ocean, if it existed, has likely been dry for billions of years, with substantial topographic modification driven by wind erosion³⁶, meteor impacts³⁷, and regolith creep³⁸. Moreover, the processes creating basin relief that allow for oceans differ. On Mars, an impact crater is likely the primary driver for the northern lowlands³⁹, whereas on Earth, plate tectonics allowed for chemical differentiation between oceanic and continental crust^{40,41}. Could these features represent the imprint of an ancient coastal shelf on Mars?”

Methods

Statistical analysis

To test whether surface steepness differs across geomorphic surfaces, we computed the median slope and curvature values at 200-meter elevation intervals. Visual inspection of the resulting profiles revealed distinct topographic breaks. To assess whether these breaks were statistically significant, we applied a Kruskal–Wallis H -test⁶⁶, a non-parametric method suitable for comparing more than two independent groups without assuming a normal distribution. The test revealed a highly significant difference in medians ($H = 27.50$, $p = 1.07 \times 10^{-6}$), indicating that the observed differences are unlikely to have occurred by chance.

Extended data

Extended Data Fig. 1 | Slope values across three elevation groups: water-formed landscapes, possible shelf, and ocean floor. (a) and (b) Column and box plots show absolute median slope values across the three elevation groups, revealing two distinct breaks at $-1,800$ m and $-3,800$ m. These patterns, together with results from the Kruskal–Wallis test (see Methods), indicate statistically significant differences between the groups.

127-128: This sentence should be the first sentence in the next paragraph.

Thank you for this great suggestion. Addressed.

Main

Could these features represent the imprint of an ancient coastal shelf on Mars? In the context of available geologic evidence, the presence of a coastal shelf between $-1,800$ m and $-3,800$ m is plausible. On Earth, coastal shelves typically occupy elevations several hundred meters below sea level, in part due to the effects of transgressive and regressive sea-level fluctuations and associated sediment deposition. Similar processes seemed to have operated on Mars. For example, interpreted stacked and multi-lobed deltas in two major deltaic systems record significant sea-level changes—up to 900 meters of transgression in Aolis Dorsa and over 500

meters of regression in Hypanis^{36,39}. Additional evidence includes river avulsions that formed multiple deltaic lobes³⁷, narrowing of river channel belts near the coast due to backwater morphodynamics³⁸, and valley network systems terminating near the proposed coastal margin⁴⁶. Moreover, the proposed elevation range of the shelf is consistent with the upper limit of the putative shorelines^{10,13}—a pattern also commonly observed on Earth. Together, these lines of evidence support the possibility of a coastal shelf on Mars.”

130-132, 508-510: Chan et al. (JGR Planets 2018) considered valley networks as markers bounding the location of an ocean coastline, as the authors are suggesting. That paper would provide useful context for their arguments.

Thank you for mentioning this paper. We agree, and we incorporated it in our revised manuscript.

132-133: The possible Arabia shoreline switches from the inner to the outer edge of the possible shelf in the area of Arabia Terra (Fig. 3b). Perhaps deformation due to true polar wander, suggested by some previous studies (Perron et al. 2007; Citron et al. 2018) would reconcile this observation. More generally, it would be interesting to see if this proposed deformation would reduce the range of elevations of the hypothesized shelf (Fig. 1d and Fig. 2c,d).

Thank you for raising this excellent comment. When comparing the modern topography to the paleotopography corrected for true polar wander (based on previous models), we find that the Arabia shoreline shifts to higher elevations in some regions, suggesting that post-ocean deformation likely modified the elevation of both the shoreline and the associated shelf zone (see figure below).

However, our approach of analyzing the surface as geomorphic domains (valley networks connected to outlet canyons, and depositional rivers and deltas) helps mitigate this uncertainty, as such long-wavelength deformation would affect broad areas rather than selectively altering the elevation ranges of globally distributed landform mosaics. As a result, we expect the overall signal of the shelf-like zone to remain coherent despite these large-scale geophysical adjustments.

We now acknowledge this potential source of uncertainty more explicitly in the revised limitations section.

Methods

“Limitations and uncertainties

Our results are subject to several uncertainties. One is the potential alteration of topography due to true polar wander and the emplacement of the Tharsis volcanic province, which likely caused uplift near Tharsis and subsidence farther away^{11,12}. However, analyzing the surface as geomorphic domains helps mitigate this, since such deformation would affect broad regions rather than the specific elevation ranges of landform mosaics. A second source of uncertainty is the isostatic response to ocean unloading, which on Earth can modify elevations by several hundred meters following ocean retreat. However, recent estimates for Mars suggest that isostatic rebound likely ranged from several tens to just over one hundred meters⁷⁸. However, recent estimates for Mars suggest that isostatic rebound likely ranged from several tens to just over one hundred meters¹². Yet, the ~2 km elevation span of our detected shelf-like zone exceeds expected rebound estimates and remains consistent with depositional features. A third source of uncertainty is long-term burial, exhumation, and erosion³⁹. While these processes may have introduced regional variability, they are unlikely to alter the broader topographic patterns we identify at the global scale.”

151-152: "The transition from land to sea is not a distinct shoreline, but rather a zone of low slope and curvature near sea level on Earth." These are not mutually exclusive – Earth's coastline is a topographic and bathymetric feature, yes; but it is also marked by distinctive landforms, which is what previous authors have considered for Mars.

We thank the reviewer for this important clarification. We agree that the transition from land to sea on Earth is marked both by a topographic signature—such as a zone of low slope and

curvature—and by a suite of geomorphic features, including beach ridges, wave-cut platforms, and deltas. Our intention was not to separate these aspects, but rather to emphasize that, beyond individual shoreline landforms, Earth's coastal transition also manifests as a broad and persistent topographic signal, particularly in continental shelf zones.

We have revised the sentence to clarify that these two perspectives are complementary rather than mutually exclusive and to better reflect how previous studies on Mars have primarily focused on geomorphic shoreline features, while our approach emphasizes the broader morphometric context.

Main

“We investigated whether a global ocean drying up would leave a topographic footprint at planetary scale. Our global topographic analysis of elevation, slope, and curvature shows that the transition from land to sea on Earth is not limited to a single distinct shoreline, but also includes a broader zone of low slope and curvature near sea level. This morphometric transition complements, rather than replaces, the geomorphic features typically associated with coastlines, such as shorelines and deltas. We presented evidence for a similar topographic zone on Mars, which is bounded at high elevations by deltaic landforms and depositional rivers, and encompasses previously proposed shorelines. Based on this evidence, we propose that Mars has an ancient coastal shelf from -1800 to -3800 m elevation that once bounded a global ocean (Figs. 2, 3, and 4). Alternative explanations for the observed slope–curvature break—such as post-impact relaxation, volcanic resurfacing, or fluvial erosion and deposition—are evaluated (see Methods; Extended Data Fig. 10), but none reproduce the coherent stepped morphology and spatial consistency of the shelf-like transition as plausibly as long-term modification by a standing body of water. “

157-161: Does the continental shelf on Earth line up with glacial sea-level lowstands everywhere? Have the authors compared Earth’s shelves with a lowstand sea-level model that includes solid-Earth deformation?

Thank you for raising this important comment. It motivated us to conduct an additional analysis to evaluate how much of Earth’s continental shelf falls within the elevation range associated with glacial sea-level lowstands. We now include maps showing the distribution of elevations within the proposed shelf zones on both Earth and Mars.

On Earth, we found that approximately 57% of the continental shelf lies within the elevation zone affected by glacial sea-level fall (around -130 ± 10 meters), while the remaining 43% lies below that zone. This deeper portion likely reflects sedimentation and subsidence accumulated over multiple sea-level cycles. While we did not directly compare our results with a glacial isostatic adjustment (GIA) model that includes solid-Earth deformation, we acknowledge that incorporating such models (e.g., ICE-6G or ICE-7G) could refine this comparison in future work. We now mention this as a limitation in the revised manuscript.

We also applied a similar elevation-distribution analysis to the detected shelf zone on Mars. There, we found that a smaller proportion of pixels occur in the lower part of the elevation range, which is consistent with the retreat of the crustal dichotomy and the spatial arrangement of deltas. The broader and deeper portions of the shelf zone on Mars may have formed through regressive

deposition during ocean desiccation, combined with sediment redistribution by waves, currents, and other depositional processes.

We have revised the manuscript text to include these findings and added an extended data figure to support this new analysis.

Main

“On Earth, the continental shelf typically occurs within the first few hundred meters below sea level, with 90% of its area located within the upper 400 meters. Deltas also tend to cluster within this zone. The range of shelf elevations reflects the processes that have shaped the shelf over time. Sea-level fluctuations, combined with river deposition during late glacial cycles, have left a clear imprint on the continental shelf, producing elevation ranges from tens of meters up to 134 meters (Extended Data Figs. 6, 7)^{19, 25, 33}. Similarly, storm reworking contributes to shelf morphology, generating elevation variations from a few meters to several tens or even up to 100 meters^{26, 44}. These appear to be major mechanisms responsible for shaping much of the shelf. For example, when calculating the area influenced by sea-level change down to -134 meters, we find that approximately 57% of the total shelf area falls within this range (Fig. x). However, in some regions—such as the North Sea shelf and the Antarctic shelf—the shelf extends beyond 1,000 meters below sea level due to glaciation and tectonic deformation (Extended Data Fig. 5)¹⁹. These broader elevation ranges reflect: (1) the intersection of the shelf with incised canyons and fault systems associated with tectonic rifting and accretion⁴⁴, and (2) deep glaciated zones such as those on the Antarctic and Arctic Ocean shelves¹⁹.”

Extended Data Fig. 6 | Elevation ranges across the terrestrial and martian shelves. (a) and (c) Elevation distribution across the mapped terrestrial shelf. These plots show how the sea-level fall during the Last Glacial Maximum significantly influenced the distribution of pixels on the continental shelf. Elevations below the glacial lowstand may reflect additional processes such as tectonic subsidence. (b) and (d) Elevation distribution of the detected shelf-like pixels on Mars, compared with the elevations of the mapped Arabia shoreline (red) and Deuteronilus shoreline (blue). The histograms illustrate how the detected shelf pixels span the elevation range between interpreted transgressive and regressive phases. Pixels below this range may represent regressive depositional environments associated with ocean retreat or modification by wave action during desiccation.

171-184: This is an important point that is sometimes neglected in studies of possible Mars coastal landforms. I agree with the authors that recessional coastal landforms could span a wide range of elevations on Mars.

Thank you for this supportive comment. We agree that the wide elevation range of recessional coastal landforms is a key consideration and have clarified this point further in the revised text. Extended Data Figure 7 and figure S4 now highlight this mechanism more. Thanks.

Extended Data Fig. 7 | Sea level changes based on stratigraphic analyses from three major deltaic systems (Oxia Planum, Hypanis Valles, and Aeolis Dorsa)^{37,38,39}. (a) The figure demonstrates significant sea level fluctuations over geologic time, including a regression of approximately 500 meters and a transgression of around 900 meters. (b) and (c) show the

elevation profiles distribution of the deltaic systems in Aeolis Dorsa and Hypanis. Interestingly, these level changes occur within the identified transitional zone (-1800 meters to -3800 meters).

Fig. S4 | Examined deltas along the proposed shorelines, detected shelf cells, and layered deposits. (a) Global distribution of examined deltas—both open-basin and closed-basin—used in this study, along with the proposed shorelines, detected shelf cells, and thick layered deposits. Background map is based on MOLA topography. (b) Histogram showing the elevation ranges of the two delta categories.

192-194: I suggest rephrasing to "rather than only looking for distinct shorelines". The authors are not the first to suggest that fluvio-deltaic depositional landforms may mark shorelines. Indeed, depositional landforms are one class of features that have been used in multiple previous studies to identify possible shorelines (including by the second author: Cardenas & Lamb 2022; DiBiase et al. 2013).

Thank you. We agree, and we adjusted the text.

Comments on figures:

Figure 1: In 1a, the delta points obscure the continental shelf and slope, the main subject of the paper! Also, what are the colors in 1b and 1d?

Thank you for this helpful comment. We included the deltas in Fig. 1a to show that their elevations generally align with the continental shelf, but we agree that they may obscure some shelf details. To address this, we have added a supplementary figure that shows the shelf extent more clearly across different regions. Regarding the colors in Fig. 1b and 1d, they correspond to the same features mapped in panels 1a and 1c, respectively. We have clarified this in the figure caption to avoid confusion.

Figure 3: In 3c and 3d, the black transects appear to correspond to the text labels. I don't think that is what is meant. Also, the Rhine Valley and Hypanis Vallis are not perfectly straight. I think the authors mean the lines on their plots are transects in the orientations of these features.

Thank you for this comment. To clarify: the transect across Hypanis Vallis is straight because it follows the broader landscape context, not just the channel itself. In contrast, the transect across the Rhine Valley follows the active river channel, which is sinuous. This reflects the difference in how the features were traced—Hypanis represents an inactive depositional system, while the Rhine is an actively evolving fluvial system. We have revised the figure in response to your comment. Please see below.

Text labels in figures contain multiple typographical errors.

Thank you for pointing this out. We have carefully reviewed all figure labels and corrected the typographical errors. Updated figures have been included in the revised manuscript. One example is Figure 4.

Referee #3 (Remarks to the Author):

This paper investigates whether a paleo-coastal shelf, formed by an ancient ocean(s), could exist in the northern lowlands of Mars. Whether an ancient ocean once filled Mars's northern hemisphere-spanning basin is a major open question in planetary science today. Many previous studies have focussed on whether the dichotomy boundary, which divides the northern lowlands of Mars from the southern highlands, is an erosional shoreline – this work proposes instead that we would not necessarily expect to see an erosional shoreline, but a depositional coastal shelf. This investigation primarily compares the modern topography of Mars's northern lowlands to coastal marginal topography on Earth. The investigation is framed both in the context of whether planets should be expected to leave discernible shorelines, and whether Mars has evidence for one. In general, the authors find strong similarities between the topography of Mars dichotomy and coastal shelves on Earth, such as slope and curvature.

The paper raises some interesting concepts, particularly what we should be looking for when investigating the signals of ancient oceans on Mars. I generally agree that a coastal shelf on Mars is possible – however, I do not believe that the analysis as it is currently presented makes a sufficient or plausible cause for it. In addition, there are many caveats that the paper does not sufficiently address. For these reasons, I am declining to recommend publication.

We thank the reviewer for their thoughtful summary and for recognizing the relevance of exploring shelf-like signals as evidence for ancient oceans on Mars. While we acknowledge the reviewer's concerns, we have substantially revised the manuscript to address key limitations, clarify our interpretations, and strengthen our analysis. Specifically, we clarified that we did not use any closed-basin deltas, provided and tested alternative explanations for the formation of the shelf, and clarified the distinctions between global and local investigation constraints. Detailed responses and new supporting evidence are provided point-by-point below.

We hope these additions address the reviewer's concerns and present a more compelling and plausible case for our interpretation.

(1) Open vs basin deltas

A major point of the paper is that fluvial landforms, such as inferred deltas, along the dichotomy region of Mars, fall within the expected elevation range/position for a coastal shelf/ocean boundary. This interpretation builds on earlier work (Di Achille & Hynes, 2010), which interpreted such features as forming an equipotential surface and potentially formed into an open basin (an ancient ocean). The scale of the topographic analysis done in this new study using relatively coarse resolution, global MOLA (463 m/pix) would also support this interpretation. Indeed, in the case of the two examples presented (Hypanis and Aeolis Dorsa), this interpretation also holds true.

However, since the Di Achille & Hynes study, there have been multiple regional-scale studies, which re-analyzed the inferred open basin deltas, using higher resolution topographic datasets. All of these studies found that many or most of the inferred deltas

formed into closed basins, along the dichotomy – likely forming smaller paleolakes, and that a large, ocean-sized body of water was not required. None of these papers are cited in this new manuscript. The papers below all demonstrate that careful assessment of the local or regional geology is necessary, in order to support global hypotheses. The authors need to present the caveats of the distribution of inferred deltas – multiple studies have now shown that a one size fits all approach does not work. This, in my opinion, is a major flaw of the manuscript.

<https://agupubs.onlinelibrary.wiley.com/doi/full/10.1029/2021GL094271>

<https://agupubs.onlinelibrary.wiley.com/doi/full/10.1029/2019GL083046>

<https://doi.org/10.1016/j.geomorph.2020.107129>

We thank the reviewer for this important comment. We agree that not all deltas along the dichotomy boundary formed into open basins and that local and regional geological context is essential. We would like to clarify that our analysis did not rely on all previously mapped deltas. Instead, we applied specific criteria to identify a subset of deltas that most likely flowed into the northern lowlands. These include deltas that: (1) are open to downstream flow, (2) are located along the dichotomy boundary rather than within crater interiors, and (3) exhibit complex deltaic morphologies—such as multiple lobes—interpreted to reflect transgressive or regressive processes.

In response to the reviewer’s comment, we have now cited the relevant studies and addressed this debate explicitly in the Methods section. We also include coordinates and classification maps of the deltas used in our analysis. Additionally, we calculated the elevation of delta channels and lobes to quantify the range of regression and transgression recorded in their stratigraphic and geomorphic expression. Please see the revised text and accompanying data below.

Methods

“Given the ongoing debate over whether martian deltas formed in open or closed basin systems, we chose to compile all available delta datasets and then apply specific filtering criteria^{4,10,36,37,38,39,72,73,74,75}. We selected deltas that (1) are open to downstream flow and located along the dichotomy boundary, and (2) exhibit complex stacking patterns interpreted as evidence of formation within either regressive or transgressive depositional environments. This filtering resulted in a set of 48 deltas (Table S1; Fig. S4). We further examined these deltas and classified them into two categories based on their morphology: single-lobate deltas and stacked deltaic systems. To further cross-validate our compilation, we calculated the elevations of all channels and lobes (excluding those preserved as ridges) within the largest deltaic systems in Aeolis Dorsa and Hypanis, in order to capture elevation changes potentially associated with past sea-level fluctuations.”

Fig. 5 | Sedimentologic and mineralogic evidence for sedimentary rocks, clay stratigraphy, and open-basin deltas along and within the detected shelf. (a) and (b) Polar stereographic projections of the detected shelf showing 14,386 mounds interpreted to have formed by dichotomy retreat, consisting of thick layered deposits up to 500 m in southern Chryse Planitia, Mars, along with widespread clay-bearing stratigraphy and open-basin deltas. (c) and (d) Box plots and histograms of open-basin delta elevations show that most deltas are built atop the detected shelf and are consistent in elevation with the layered deposits. The blue gradient—horizontal in c and vertical in d—indicates the range of interpreted sea-level changes, based on evidence of transgression and regression from two large deltaic systems in Aolis Dorsa (e), particularly Hypanis (f). Background map is from MOLA.

Extended Data Fig. 7 | Sea level changes based on stratigraphic analyses from three major deltaic systems (Oxia Planum, Hypanis Valles, and Aeolis Dorsa)^{37,38,39}. (a) The figure demonstrates significant sea level fluctuations over geologic time, including a regression of approximately 500 meters and a transgression of around 900 meters. (b) and (c) show the elevation profiles distribution of the deltaic systems in Aeolis Dorsa and Hypanis. Interestingly, these level changes occur within the identified transitional zone (-1800 meters to -3800 meters).

Fig. S4 | Examined deltas along the proposed shorelines, detected shelf cells, and layered deposits. (a) Global distribution of examined deltas—both open-basin and closed-basin—used in this study, along with the proposed shorelines, detected shelf cells, and thick layered deposits. Background map is based on MOLA topography. (b) Histogram showing the elevation ranges of the two delta categories.

Table S1. The examined open-basin deltaic features along the dichotomy boundary. This dataset was filtered based on the following studies 1,2,3,4,5,6,7,8.

Longitude	Latitude	Elevation (m)	Classification
150.3957	-2.57724	-2467	Stacked deltaic systems
151.5326	-5.34104	-2491	Stacked deltaic systems
150.9137	-4.96192	-2405	Stacked deltaic systems
150.558	-4.69387	-2293	Stacked deltaic systems
149.1995	-2.7041	-2268	Stacked deltaic systems
150.0536	-3.35775	-2391	Stacked deltaic systems
153.7412	-4.08644	-2108	Stacked deltaic systems
154.6857	-4.59259	-2299	Stacked deltaic systems
154.909	-4.98344	-2341	Stacked deltaic systems
149.3736	-2.52552	-2400	Stacked deltaic systems
155.0388	-4.69236	-2245	Stacked deltaic systems
154.8564	-4.53841	-2218	Stacked deltaic systems
153.679	-6.01441	-2323	Stacked deltaic systems
154.5498	-6.66616	-2137	Stacked deltaic systems
153.1269	-6.59133	-2075	Stacked deltaic systems
149.1735	-6.28996	-1982	Stacked deltaic systems
150.8376	-6.85914	-1983	Stacked deltaic systems
149.3147	-5.96477	-1802	Stacked deltaic systems
148.7687	-5.74833	-1706	Stacked deltaic systems
-45.4158	11.35919	-2604	Stacked deltaic systems
-44.5966	11.58354	-2665	Stacked deltaic systems
-44.9182	13.10371	-2880	Stacked deltaic systems
-147.21	-5	-2531	Single-lobate deltas
148.47	-7.75	-1586	Single-lobate deltas
148.7	-7.2	-1993	Single-lobate deltas
148.29	-7.44	-1806	Single-lobate deltas
-23.8943	17.83914	-2978	Single-lobate deltas
149.76	-7.71	-1699	Single-lobate deltas
149.94	-7.93	-1756	Single-lobate deltas
-15.168	31.603	-3941	Single-lobate deltas
132.832	-5.059	-2359	Single-lobate deltas
132.694	-3.615	-2237	Single-lobate deltas

131.193	-1.654	-1815	Single-lobate deltas
121.636	2.167	-1916	Single-lobate deltas
147.827	-7.462	-1638	Single-lobate deltas
140.492	-5.63	-2270	Single-lobate deltas
142.634	-3.725	-2452	Single-lobate deltas
144.791	-4.05	-2211	Single-lobate deltas
122.409	1.704	-933	Single-lobate deltas
-52.881	36.517	-3193	Single-lobate deltas
-55.944	44.194	-2203	Single-lobate deltas
9.7	37.603	-2709	Single-lobate deltas
-146.354	-5.28035	-2200	Single-lobate deltas
71.944	29.462	-1702	Single-lobate deltas
148.003	-7.625	-1592	Single-lobate deltas
147.411	-6.649	-1827	Single-lobate deltas
-14.404	32.19	-3823	Single-lobate deltas
-57.254	34.931	-1240	Single-lobate deltas

(2) Global scale approach vs regional geological complexities

A recent published paper inferred that the highland-lowland dichotomy has retreated by 100s of kilometres in the Mawrth Vallis and possibly across the wider Chryse Planitia region. The authors show that this retreat records a major episode of erosion and occurred between 4.0 – 3.7 Ga. One potential cause of this could be the retreat of a Noachian age ocean on Mars. This would seemingly disagree with the authors’ hypothesis (for this new manuscript) that terrain north of the dichotomy is primarily a product of deposition.

<https://www.nature.com/articles/s41561-024-01634-8>

Similarly, Hesperian-Amazonian age outflow channels, likely caused by megaflooding events, in the Chryse Planitia region have had a major impact on the topography (i.e., Ares Valles, Kasei Valles). These channel systems have both eroded the terrain and deposited huge volumes of material in this area of Mars. While the paper does acknowledge that multiple oceans on Mars could have existed at different stages in time (I guess a late stage one filled by outflow channel debauching?), outflow channels are not directly mentioned. I would like to know how outflow channel formation plays into the authors’ coastal shelf hypothesis.

Thank you for raising this excellent comment, which motivated us to expand and clarify our interpretation and further consider regional complexity versus global patterns. We agree that recent work by McNeil et al., 2025 provides compelling evidence for large-scale erosion and retreat of the dichotomy boundary in parts of Chryse Planitia and Mawrth Vallis. However, this retreat is associated with the removal of previously deposited layered materials—some up to 500

m thick. These same layered deposits fall within our mapped shelf zone, suggesting that what was eroded may have originally been part of a depositional coastal system. In response, we now include a new figure (Extended Data Fig. 9) illustrating how depositional and erosional phases may have modified the shelf over time.

We also acknowledge the role of outflow channels in shaping the northern lowlands during the Hesperian and Amazonian. While these channels contributed both erosion and deposition, the key shelf-related deposits we highlight are older, with many stratigraphically and chronologically constrained to the Noachian–early Hesperian (e.g., 4.1–3.7 Ga), preceding the major outflow events. We now discuss the role of burial and erosion that may have further modified the shelf signal.

These points are also reflected in the updated limitations section, where we explicitly address the potential overprinting of the original surface by later erosion and resurfacing events.

Fig. 5 | Sedimentologic and mineralogic evidence for sedimentary rocks, clay stratigraphy, and open-basin deltas along and within the detected shelf. (a) and (b) Polar stereographic projections of the detected shelf showing 14,386 mounds interpreted to have formed by dichotomy retreat, consisting of thick layered deposits up to 500 m in southern Chryse Planitia, Mars, along with widespread clay-bearing stratigraphy and open-basin deltas. (c) and (d) Box plots and histograms of open-basin delta elevations show that most deltas are built atop the detected shelf

and are consistent in elevation with the layered deposits. The blue gradient—horizontal in c and vertical in d—indicates the range of interpreted sea-level changes, based on evidence of transgression and regression from two large deltaic systems in Aeolis Dorsa (e), particularly Hypanis (f). Background map is from MOLA.

Extended Data Fig. 9 | Detected shelf, layered deposits, and clay-bearing stratigraphies, and their possible evolution over time. (a) Polar stereographic projections of the detected shelf in southern Chryse Planitia, Mars, showing 14,386 mounds interpreted as products of dichotomy retreat and composed of thick layered deposits up to 500 m, along with widespread clay-bearing stratigraphy. Background map is based on MOLA topography. (b) Schematic diagram illustrating a possible evolutionary scenario in which the sedimentary layered deposits formed as part of a shelf system that was later eroded. (c) Image from Context Camera (CTX) aboard NASA's Mars Reconnaissance Orbiter showing the connection between the layered mounds and the dichotomy boundary. Together, these observations provide sedimentologic and mineralogic evidence for a potential coastal shelf in southern Chryse Planitia.

Limitations and uncertainties

“Our results are subject to several uncertainties. One is the potential alteration of topography due to true polar wander and the emplacement of the Tharsis volcanic province, which likely caused uplift near Tharsis and subsidence farther away^{11,12}. However, analyzing the surface as geomorphic domains helps mitigate this, since such deformation would affect broad regions rather than the specific elevation ranges of landform mosaics. A second source of uncertainty is the isostatic response to ocean unloading, which on Earth can modify elevations by several hundred meters following ocean retreat. However, recent estimates for Mars suggest that isostatic rebound likely ranged from several tens to just over one hundred meters⁷⁸. However, recent estimates for Mars suggest that isostatic rebound likely ranged from several tens to just over one hundred meters¹². Yet, the ~2 km elevation span of our detected shelf-like zone exceeds expected rebound estimates and remains consistent with depositional features. A third source of uncertainty is long-term burial, exhumation, and erosion³⁹. While these processes may have introduced regional variability, they are unlikely to alter the broader topographic patterns we identify at the global scale.”

(3) Lack of consideration for alternative hypotheses

Figure 3 illustrates the proposed zone for the coastal shelf on Mars, which has largely been inferred from the gentle sloping topography (taking into account point 1 above). Similarly, figure 4 schematically and graphically illustrates how the topography changes downslope for both Earth and Mars. A major issue I have with this work is that other hypotheses for what could produce this topographic distribution aren't really considered. And again, this relates to the point above about regional geological complexities. Why could this type of topography not be produced by outflow channel re-surfacing (point above) in the Chryse region? Why not volcanic re-surfacing – for example, could the gently sloping terrain around Tharsis be associated with the volcanoes here? In Isidis, the lavas from Syrtis Major extend into the basin here, making it unlikely to be a coastal shelf.

Thank you very much for this important comment. We agree that alternative resurfacing processes—including volcanic flows, outflow channel activity, and fluvial erosion—must be considered. To address this, we conducted a comparative analysis using the Hellas Basin, the second-largest impact basin on Mars. Like the northern lowlands, Hellas has experienced fluvial erosion, volcanic resurfacing, and the presence of large channels, but it lacks any proposed oceanic history.

We applied the same topographic classification to Hellas at the same scale, and we did not observe the distinct slope and curvature signal that characterizes the northern lowlands. This contrast suggests that the northern lowland morphometric pattern is not simply a result of generic resurfacing or basin evolution, but may instead reflect a different process—potentially linked to a past ocean.

In response, we added a new section to the manuscript explicitly discussing the null hypothesis, and we included a new extended data figure showing the Hellas results. We also revised the main text to consider these alternative explanations and to clarify why our observations support a marine interpretation.

Main

“We investigated whether a global ocean drying up would leave a topographic footprint at planetary scale. Our global topographic analysis of elevation, slope, and curvature shows that the transition from land to sea on Earth is not limited to a single distinct shoreline, but also includes a broader zone of low slope and curvature near sea level. This morphometric transition complements, rather than replaces, the geomorphic features typically associated with coastlines, such as shorelines and deltas. We presented evidence for a similar topographic zone on Mars, which is bounded at high elevations by deltaic landforms and depositional rivers, and encompasses previously proposed shorelines. Based on this evidence, we propose that Mars has an ancient coastal shelf from -1800 to -3800 m elevation that once bounded a global ocean (Figs. 2, 3, and 4). Alternative explanations for the observed slope–curvature break—such as post-impact relaxation, volcanic resurfacing, or fluvial erosion and deposition—are evaluated (see Methods; Extended Data Fig. 10), but none reproduce the coherent stepped morphology and spatial consistency of the shelf-like transition as plausibly as long-term modification by a standing body of water. “

Testing the null hypothesis: impact basin control

To assess whether the stepped slope morphology observed in the northern lowlands is a unique topographic signature of that region or a generic consequence of large impact basin formation modified by processes such as fluvial activity^{34,71}, volcanic resurfacing⁷⁷, or prolonged burial and erosion⁴⁰, we conducted a parallel topographic analysis of Hellas Basin—the second largest basin on Mars after the northern lowlands. We selected Hellas for its large scale and state of preservation, though it lacks clear evidence for a standing ocean and has been extensively modified by burial, exhumation, volcanic, and fluvial processes. We applied the same analytical framework as in the northern lowlands: a uniform grid at 5 km per pixel resolution, with elevation values binned every 200 m (Extended Data Fig. 10). For each bin, we calculated the median slope and tested for statistically significant slope breaks using the Kruskal–Wallis H-test. This consistent methodology enables direct comparison of elevation-dependent morphologic transitions between basins. If shelf-like slope signatures were solely the result of post-impact erosional and depositional processes, similar features would be expected in both basins. However, no comparable slope break was observed in the control basin (Extended Data Fig. 10), suggesting that the features in the northern lowlands reflect modification by additional processes—such as long-lived oceans^{1,2,3,4,5,6,7,8,9,10,11,12,13,14,15} and deposition during sea-level fluctuations—rather than impact alone^{36,37,38,39}.

Extended Data Fig. 10 | Null hypothesis test using Hellas Basin. (a) Topography of Hellas Basin, the second-largest impact basin on Mars, overlaid with valley networks (black)^{34,68}, open-basin lakes (blue)³⁴, and outlet canyons (orange)³⁴. (b) and (c) Median slope values per 200 m elevation bin show no evidence of abrupt, shelf-like transitions similar to those identified in the northern lowlands. The distribution also lacks the coupled minima observed in the ocean-bearing hypothesis, which would be shelf and ocean or large sea floor. These results reinforce that the morphometric signal detected in the northern lowlands is not a generic outcome of large-basin formation or surface degradation but may instead reflect a distinct process, such as ocean-related resurfacing.

Line by line:

Line 96: “Open-basin deltas along the northern dichotomy range in elevation from -1,638 to -2,880 meters” – The Scholes et al 2022 paper which is referenced here notes: “Although, subsequent detailed regional mapping has shown that many of these deltas likely formed in closed localized paleolakes or seas rather than a northern ocean”

Thank you for raising this comment. We did not rely solely on Di Achille et al. (2010) or Scholes et al. (2022). Instead, we filtered the deltas presented in both studies to include only those in open-basin systems, and we supplemented them with major deltaic systems from Aeolis Dorsa (Cardenas et al., 2022) and Hypanis Valles (Fawdon et al., 2018). This resulted in a refined dataset of 48 deltaic systems, now classified in Table S1 as either single-lobed or multi-lobed stacked systems. Our main interpretation is not based on all previously mapped deltas, but specifically on the most robust examples—especially the large deltaic systems at Aeolis Dorsa and Hypanis—where we calculated elevation profiles to analyze the vertical distribution of lobes and assess regression and transgression dynamics. In response, we have clarified this in the revised text and refer to our detailed reply to Comment 1 above.

Methods

“Given the ongoing debate over whether martian deltas formed in open or closed basin systems, we chose to compile all available delta datasets and then apply specific filtering criteria^{4,10,36,37,38,39,72,73,74,75}. We selected deltas that (1) are open to downstream flow and located along the dichotomy boundary, and (2) exhibit complex stacking patterns interpreted as evidence of formation within either regressive or transgressive depositional environments. This filtering resulted in a set of 48 deltas (Table S1; Fig. S4). We further examined these deltas and classified them into two categories based on their morphology: single-lobate deltas and stacked deltaic systems. To further cross-validate our compilation, we calculated the elevations of all channels and lobes (excluding those preserved as ridges) within the largest deltaic systems in Aeolis Dorsa and Hypanis, in order to capture elevation changes potentially associated with past sea-level fluctuations.”

Fig. 5 | Sedimentologic and mineralogic evidence for sedimentary rocks, clay stratigraphy, and open-basin deltas along and within the detected shelf. (a) and (b) Polar stereographic projections of the detected shelf showing 14,386 mounds interpreted to have formed by dichotomy retreat, consisting of thick layered deposits up to 500 m in southern Chryse Planitia, Mars, along with widespread clay-bearing stratigraphy and open-basin deltas. (c) and (d) Box plots and histograms of open-basin delta elevations show that most deltas are built atop the detected shelf and are consistent in elevation with the layered deposits. The blue gradient—horizontal in c and vertical in d—indicates the range of interpreted sea-level changes, based on evidence of transgression and regression from two large deltaic systems in Aeolis Dorsa (e), particularly Hypanis (f). Background map is from MOLA.

Extended Data Fig. 7 | Sea level changes based on stratigraphic analyses from three major deltaic systems (Oxia Planum, Hypanis Valles, and Aeolis Dorsa)^{37,38,39}. (a) The figure demonstrates significant sea level fluctuations over geologic time, including a regression of approximately 500 meters and a transgression of around 900 meters. (b) and (c) show the elevation profiles distribution of the deltaic systems in Aeolis Dorsa and Hypanis. Interestingly, these level changes occur within the identified transitional zone (-1800 meters to -3800 meters).

Fig. S4 | Examined deltas along the proposed shorelines, detected shelf cells, and layered deposits. (a) Global distribution of examined deltas—both open-basin and closed-basin—used in this study, along with the proposed shorelines, detected shelf cells, and thick layered deposits. Background map is based on MOLA topography. (b) Histogram showing the elevation ranges of the two delta categories.

Line 192: Rosalind Franklin is scheduled to launch in 2028 and land in 2030.

Thank you. Addressed.

Figure 4c: typo on left plot “Hyapnis”

Thank you. Fixed.

Author Rebuttals to Initial Comments:

Referee comments are in **bold black text**

Our replies to comments are in **purple text (Times New Roman)**

Text that we revised upon reflecting on one or more Referee comments is in **light blue text (Arial)**

Reviewer #1 (Formal Review for Authors (shown to authors)):

Publication is recommended once problematic new portions are removed (see below).

Substantial weakness was introduced into the revised manuscript with the addition of content related to Hellas and layered deposits in the circum-Chryse basin (External Data 9 & 10). These components—primarily figures with captions—are under-developed, lacking description and key literature. As this is an appeal, my critique is brief and highlights the issues with two examples

We thank the referee for this feedback and agree that the newly added material related to Hellas and layered deposits in the circum-Chryse basin (Extended Data Figs. 9 & 10) was not sufficiently developed for inclusion in an appeal-stage revision. To address this concern directly, we have removed Extended Data Figs. 9 and 10 in their entirety, along with the associated captions and the related section about the analysis of Hellas Basin. Please see our detailed responses to the individual comments below.

**1) There are no citations with respect to Hellas Planitia, yet the Hellas basin has been proposed as the location of a former sea or “standing body of water.” https://pubs.usgs.gov/sim/3096/sim3096_pamphlet.pdf
<https://agupubs.onlinelibrary.wiley.com/doi/full/10.1029/2006JE002830>**

Such prior work should at least be acknowledged,

We thank the reviewer for this great point. We agree that any discussion of Hellas should acknowledge prior work proposing potentially standing bodies of water. Because we removed the Hellas-related analysis and figure, we also removed the associated text, and therefore this issue no longer applies to the revised manuscript.

2) Unclear location of area shown in External Data 10c, but it appears to be part of the chaotic terrain in and near Chryse Planitia. How are the layered mounds related to or distinguished from the plains units surrounding Chryse? This is a critical question to address in order to claim the layered mounds are remnant coastal shelf deposits. Alternate models for chaotic terrain are missing.

We thank the reviewer for this valuable comment. We note that the location issue raised here pertains to Extended Data Fig. 9c (rather than Fig. 10c). We agree that interpreting layered mounds within or near chaotic terrain depicted in Extended Data Fig. 9c as remnant coastal shelf deposits would require clear location context, explicit distinction from surrounding plains units,

and engagement with alternate models for chaotic terrain. Accordingly, we have removed Extended Data Fig. 9 (including panel c).

We note that layer deposits in Fig. 5a,b and the associated plot are not based on the chaotic-terrain mounds highlighted in Extended Data Fig. 9. Instead, Fig. 5 uses previously mapped layered deposits/mounds along the dichotomy margin that have been interpreted as records of dichotomy retreat (i.e., erosional remnants of a once more continuous Noachian unit) rather than collapse-related chaos (please see McNeil et al., 2021). These mounds differ from chaotic terrain in that they are erosional remnant mounds, not collapse blocks; they show a stratigraphic expression consistent with dichotomy-margin stratigraphy, suggesting they are remnants of a formerly more continuous layer; and crater-count constraints place their age in the Early to Middle Noachian. Recent work investigating the mineralogy of these mounds further indicates that widespread aqueous alteration affected the mounds and surrounding highlands, consistent with extensive water–rock interactions that could have been facilitated by an early northern ocean (McNeil et al., 2025).

1. McNeil, J. D., Fawdon, P., Balme, M. R. & Coe, A. L. Morphology, morphometry and distribution of isolated landforms in southern Chryse Planitia, Mars. *Journal of Geophysical Research: Planets* 126, (2021).

2. McNeil, J. D. et al. Dichotomy retreat and aqueous alteration on Noachian Mars recorded in Highland remnants. *Nature Geoscience* 18, 124–132 (2025).

Referee #2 (Remarks to the Author):

Like the original manuscript, the revised manuscript by Zaki and Lamb advances the provocative new hypothesis that the northern lowlands of Mars bear a previously unrecognized signature of a past ocean: a shelf-like region analogous to the continental shelves on Earth. Although the paper does not demonstrate beyond any doubt that this topographic feature formed in an ancient Mars ocean, it adds a new dimension to the debate and will undoubtedly inspire further research on the topic. Perhaps most importantly, the paper makes a basic point that has not been widely discussed: if Earth’s oceans were to dry up, the enduring surficial evidence would not be beaches, but rather the broader bathymetry of the ocean basins, including the shelves. This large-scale bathymetry should be a focus of searches for evidence of a past ocean on Mars (in addition to surface geology being explored by rovers).

I have organized my comments according to the main points raised in my original review and the authors’ responses to those comments.

We thank the reviewer for their thoughtful and constructive assessment of our work. We are pleased that the revised manuscript is seen as adding a new dimension to the debate on a past northern ocean on Mars. We appreciate the reviewer’s summary of our contribution, particularly the emphasis that our goal is not to demonstrate the ocean hypothesis beyond any doubt, but to

highlight a robust, shelf-like topographic signal that motivates further testing. Below, we address each of the reviewer's specific comments in turn.

Original comment 1. Null hypothesis

1.1. Comparison with Hellas: I agree with the authors that the Hellas basin, while smaller than the northern lowlands, offers the best available comparison. I also agree that the slopes in Hellas (Fig. ED10b) look more uniform than those in and around the northern lowlands (Fig. 2c). This shows that Hellas does not have as prominent a bench as the northern lowlands. However, eyeballing the plots in the paper suggests that the slopes in Hellas do form two distinct regions that span roughly a factor of 2 (Fig. ED10c) (compared with roughly a factor of 3 or 4 in the north; 0.015 degrees to 0.05 degrees, Fig. 2c), so Hellas does appear to have a flatter region near the edge of the analyzed area. Both slope ranges are small relative to the range of slopes on Earth, which is more like a factor of 20 (Fig. 2a). So my question for the authors is why – in terms of quantitative criteria – does Hellas not qualify as having a shelf, whereas the northern lowlands do? A related minor point: Fig. ED10 only shows slope, not slope and curvature, yet the authors cite this figure in their response to support the statement that Hellas does not have coupled slope and curvature minima. I also find it difficult to see the blue lakes and yellow canyons in ED10a due to the blue and yellow colors in the elevation basemap.

We thank the reviewer for this thoughtful comment and agree that a quantitative comparison between the northern lowlands and a large impact basin such as Hellas is a useful way to frame the null hypothesis. However, in response to Referee #1's concern that the newly added Hellas (Extended Data Fig. 10) was under-developed for an appeal-stage revision—and given that Hellas itself may have hosted lakes or seas—we have removed Extended Data Fig. 10 and the associated Hellas analysis and discussion from the revised manuscript. Accordingly, the manuscript no longer makes the claim that Hellas does or does not qualify as having a shelf, and we no longer rely on Hellas to support the null-hypothesis test.

We retain the null-hypothesis framing in a concise form (impact-basin formation and subsequent modification by fluvial activity, volcanic resurfacing, burial, and erosion) but now evaluate it using the global-scale observations already presented in the manuscript. Please see below.

Main

“The same analysis of Mars' topography does not reveal a single distinct minimum in global slope and curvature, as observed on Earth (Fig. 2c, d; Fig. S3). Instead, two coupled minima in slope and curvature are observed on Mars (Extended Data Fig. 1), supported by a Kruskal–Wallis H-test ($p = 1.07 \times 10^{-6}$), indicating less than a 0.00011% probability that the differences in median slope values among the observed elevation ranges occurred by chance. The first minima lie between approximately -1800 m and -3800 m (slope = 0.31° , curvature = 0.00015 , -0.00017), and the second between -3800 m and -5000 m (slope = 0.12° , curvature =

0.0009, -0.0009). The minima are less distinct than on Earth, but this might be expected, given that Mars' ocean, if it existed, has likely been dry for billions of years, with substantial topographic modification driven by wind erosion³⁹, meteor impacts⁴⁰, Hesperian-Amazonian age outflow channels⁴¹, and regolith creep⁴². These geomorphic processes can smooth³⁹⁻⁴², incise or infill the margin and thus broaden and mute the slope and curvature minima without eliminating an underlying shelf-slope break. Long-wavelength vertical motions associated with geophysical processes—such as isostatic or flexural adjustment to ocean loading and unloading, true polar wander or Tharsis-induced deformation¹¹⁻¹²—would act over scales much larger than the mapped shelf width, primarily tilting or gently warping the margin and changing absolute slopes, but are unlikely to remove a relative slope and curvature minimum between shelf and slope. The processes creating basin relief also differ between the two planets. On Mars, an impact crater is likely the primary cause for the northern lowlands⁴³, whereas on Earth, plate tectonics allowed for chemical differentiation between oceanic and continental crust^{44,45}. “

1.2, 1.3: Sedimentary and stratigraphic features: These datasets are informative additions that specifically address the point that a shelf could be an ocean-generated overprint on an impact basin rather than an original feature of the impact basin.

We thank the reviewer for this positive assessment and are glad that the sedimentary and stratigraphic datasets shown in Fig. 5a,b help clarify our interpretation that the shelf could represent an ocean-generated overprint on an impact-formed surface, rather than an original feature of impact basin formation.

Original comment 2. Post-ocean topographic change

The authors' response to this comment – that vertical motions associated with geophysical processes would be too small in amplitude to affect the topographic features they detect on Mars – is plausible, but it does not quantitatively demonstrate that the isostatic response to ocean unloading would not alter the slope and curvature patterns. Since the authors' argument is largely based on slope and curvature measurements, the possibility that deformation could tilt or warp topography seems important. However, I don't think this point should stand in the way of publication, in part because modeling the response to ocean unloading on Mars would require estimates of lithospheric properties that are currently unknown.

I would also find it helpful if the authors could address an apparent contradiction related to this point. As noted above, they suggest that geophysical mechanisms would not generate enough vertical deformation to substantially alter the topography of the hypothesized shelf on Mars. Elsewhere in the paper, however, they argue that more localized geological processes COULD alter the topography of the shelf: “The [slope and curvature] minima are less distinct than on Earth, but this might be expected, given that Mars' ocean, if it existed, has likely been dry for billions of years, with substantial topographic modification driven by wind erosion(36), meteor impacts(37), and regolith creep(38).”

We thank the reviewer for raising this great point. In the revised text, we now explicitly distinguish short-wavelength geomorphic processes (wind erosion, impacts, outflow channels, regolith creep), which can broaden and mute the minima, from long-wavelength geophysical deformation (isostatic/flexural adjustment, true polar wander, Tharsis-related warping), which primarily tilts the margin but is unlikely to remove a relative shelf–slope minimum. We also now treat such deformation as a caveat in our interpretation.

Main text

“The same analysis of Mars’ topography does not reveal a single distinct minimum in global slope and curvature, as observed on Earth (Fig. 2c, d; Fig. S3). Instead, two coupled minima in slope and curvature are observed on Mars (Extended Data Fig. 1), supported by a Kruskal–Wallis H-test ($p = 1.07 \times 10^{-6}$), indicating less than a 0.00011% probability that the differences in median slope values among the observed elevation ranges occurred by chance. The first minima lie between approximately -1800 m and -3800 m (slope = 0.31° , curvature = 0.00015 , -0.00017), and the second between -3800 m and -5000 m (slope = 0.12° , curvature = 0.0009 , -0.0009). The minima are less distinct than on Earth, but this might be expected, given that Mars’ ocean, if it existed, has likely been dry for billions of years, with substantial topographic modification driven by wind erosion³⁹, meteor impacts⁴⁰, Hesperian–Amazonian age outflow channels⁴¹, and regolith creep⁴². These short-wavelength geomorphic processes can smooth^{39–42}, incise or infill the margin and thus broaden and mute the slope and curvature minima without eliminating an underlying shelf–slope break. Long-wavelength vertical motions associated with geophysical processes—such as isostatic or flexural adjustment to ocean loading and unloading, true polar wander or Tharsis-induced deformation^{11–12}—would act over scales much larger than the mapped shelf width, primarily tilting or gently warping the margin and changing absolute slopes, but are unlikely to remove a relative slope and curvature minimum between shelf and slope. The processes creating basin relief also differ between the two planets. On Mars, an impact crater is likely the primary cause for the northern lowlands⁴³, whereas on Earth, plate tectonics allowed for chemical differentiation between oceanic and continental crust^{44,45}.”

Original comment: Are there two minima in slope and curvature on Mars?

In their response, the authors cite the K-W test in Fig. ED1 (they write Fig. ED3, but I think they mean ED1). I agree that the histogram in ED1a appears to show at least three (perhaps four) elevation zones with distinct slope distributions. However, I’m not convinced that the K-W test used in ED1 is the right choice for detecting breaks in slope. My understanding is that K-W tests for significantly different medians between samples that are not normally distributed, but it doesn’t say where the boundaries between the samples should be drawn. The fact that the three highlighted regions have significantly different medians (Fig. ED1b) does not necessarily mean that there are breaks in slope between those regions. For example, wouldn’t an inclined plane divided into three non-overlapping elevation regions yield the same result, even though the slope is the same everywhere?

We thank the reviewer for this comment. We agree that the Kruskal–Wallis test is not, by itself, a method for detecting where breaks in slope occur. In the revised Methods, we clarify that we first identify elevation zones from the continuous slope–curvature–elevation profiles (Fig. 2; Extended Data Fig. 1), and then use the Kruskal–Wallis test only to quantify that the slope distributions in these *a priori* elevation

bands are statistically distinct. We now state explicitly that the Kruskal–Wallis test is used to assess the distinctness of the elevation zones, not to locate the breaks themselves.

Main text

To test whether surface steepness differs between elevation zones, we first computed median slope and curvature values at 200 m elevation intervals on both Earth and Mars. The resulting martian profiles show an intermediate-elevation, low-slope, low-curvature interval between –1800 m and –3800 m, bounded by higher slopes and curvature at elevations > –1800 m and lower slopes and curvature at elevations < –3800 m (Fig. 2; Extended Data Fig. 1). On this basis, we defined three a priori elevation bands: > –1800 m, –1800 to –3800 m, and < –3800 m. We then applied a Kruskal–Wallis H-test⁷⁷ to these three elevation bands to quantify whether their slope distributions have different medians, without assuming normality. The test revealed a highly significant difference in median slopes ($H = 27.50$, $p = 1.07 \times 10^{-6}$; Extended Data Fig. 1), indicating that the slope populations of the three elevation zones are statistically distinct. The Kruskal–Wallis test is used here only to assess the distinctness of elevation zones defined from the slope–curvature–elevation relationship, not to locate the breaks themselves.

Original comment: Do shelves on Earth correspond to sea-level lowstands?

I appreciate the authors’ new analysis. Earth’s continental shelf depths are a bit of a puzzle! But I must be missing something about Fig. ED6d. In ED6b, there are many purple dots (elevations around –2000 m), whereas points at lower elevations (yellow to red) are less abundant. But ED6d indicates the opposite: that pixels with lower elevations are much more abundant. What does the label “Transgression Regression” mean? (Those are opposites, and what are they labeling?) Are both maps equal-area, such that pixel counts are proportional to area? Polar stereographic projections are not equal-area.

We thank the reviewer for this helpful comment. In panel **b**, we show all cells within the candidate shelf band (–1800 to –3800 m) that satisfy our low-slope criterion, coloured by elevation using a single purple–to–yellow–to–red scale (purple = shallower, yellow/red = deeper) in polar stereographic projection. At higher elevations within the band, these low-slope cells are relatively sparse and laterally dispersed, so they appear to cover a broad area despite being fewer in number. At greater depths, the low-slope cells are more tightly clustered and are also compressed toward the pole by the projection, so they appear narrower even though they are more numerous. Panel **d**, by contrast, is a histogram of all shelf-band cells computed from an equal-area raster, so pixel counts there are proportional to true area and correctly show that the deepest part of the band occupies the largest area.

The label “Transgression / Regression” was intended to highlight the elevation range of relative water-level change inferred from deltaic deposits at Aeolis Dorsa and Hypanis. We have replaced this with “Transgression / Regression inferred from deltaic deposits (Aeolis Dorsa and Hypanis)” and clarified that the deeper part of the band may record regressive deposition during

ocean desiccation, consistent with Zhurong rover observations. Extended Data Fig. 6 and its caption have been revised accordingly.

Extended Data Fig. 6 | Elevation ranges across the terrestrial and martian shelves.

(a,c) Elevation distribution across the mapped terrestrial shelf. These plots show how sea-level fall during the Last Glacial Maximum concentrated most shelf pixels into the glacial lowstand band; pixels at deeper elevations largely reflect additional processes such as tectonic subsidence. (b,d) Elevation distribution of the detected shelf-like pixels on Mars, compared with the elevations of the mapped Arabia shoreline (red) and Deuteronilus shoreline (blue). In (b), shelf-band pixels (-1800 to -3800 m) that satisfy the low-slope criterion are coloured by elevation using a single purple-to-yellow-to-red scale and shown in polar stereographic projection for visualization. Panel (d) shows the corresponding elevation histogram computed from an equal-area raster, so pixel counts are proportional to area and demonstrate that the deepest part of the band occupies the largest area. The shaded band marks the elevation range of relative sea-level change inferred from transgressive and regressive deltaic deposits at Aeolis Dorsa and Hypanis (Fig. 5; Extended Data Fig. 7). Pixels below this elevation range may represent regressive depositional environments associated with shoreline retreat, or surfaces modified by wave action during ocean desiccation, consistent with recent reports of coastal deposits up to ~500 m below the Deuteronilus shoreline⁵². Together, these observations indicate that the transgressive-regressive phases captured by preserved deltas account for only a small

fraction of the detected shelf-like pixels, and that prolonged shoreline retreat during ocean drying likely produced additional, subtler depositional surfaces at lower elevations⁵².

Minor points:

629-632: Duplicate sentence but with different citations.

Addressed. Thank you.

Fig. ED1: Caption should state that this is for Mars.

Addressed. Thank you.

Referee #3 (Remarks to the Author):

Comments to authors.

This is the second time that I have reviewed this manuscript, which investigates whether a paleo-coastal shelf, formed by an ancient ocean(s), could exist in the northern lowlands of Mars, and what topographic signature that it would leave behind.

I really appreciate all the efforts that these authors have made to address the comments and issues that the other two reviewers and I raised. One previous concern I had was that the authors have not considered alternative hypotheses to the coastal shelf one and that there was regional complexities to the interpretation. While the authors have added significant new material, for example, concerning the remnant mounds in Chryse Planitia (<https://www.nature.com/articles/s41561-024-01634-8>), and the comparative topographic study of Hellas (as the non-ocean, null hypothesis), I still don't consider this point to be fully addressed. One outstanding point I have concerns outflow channels. From my original review:

We thank the reviewer for taking the time to review our manuscript a second time and for their thoughtful and constructive assessment. We are grateful for the acknowledgement of our efforts to address the earlier comments, and we agree that carefully considering alternative hypotheses and regional complexities is essential. In the revised manuscript, we further clarify how we treat alternative explanations, including the role of outflow channels, lava flows, and other post-impact processes, and we expand our discussion of these points. We address the reviewer's outstanding concern about outflow channels in detail in the responses and revisions below.

“Similarly, Hesperian-Amazonian age outflow channels, likely caused by megaflooding events, in the Chryse Planitia region have had a major impact on the topography (i.e., Ares Valles, Kasei Valles). These channel systems have both eroded the terrain and deposited huge volumes of material in this area of Mars.”

From the response doc:

“We also acknowledge the role of outflow channels in shaping the northern lowlands

during the Hesperian and Amazonian. While these channels contributed both erosion and deposition, the key shelf-related deposits we highlight are older, with many stratigraphically and chronologically constrained to the Noachian–early Hesperian (e.g., 4.1–3.7 Ga), preceding the major outflow events. We now discuss the role of burial and erosion that may have further modified the shelf signal. These points are also reflected in the updated limitations section, where we explicitly address the potential overprinting of the original surface by later erosion and resurfacing events.”

“A third source of uncertainty is long-term burial, exhumation, and erosion³⁹. While these processes may have introduced regional variability, they are unlikely to alter the broader topographic patterns we identify at the global scale.”

The last highlighted sentence is a good point, but Chryse Planitia is one the two key regions making the case that the northern lowlands is a coastal shelf. There are several major outflow channel systems surrounding Chryse Planitia, which deposited huge volumes of material here via megaflooding. Indeed, the USGS geologic map of the northern lowlands attributes much of the material here to the outflow channels (<https://pubs.usgs.gov/sim/2005/2888/>). There is significant overlap between these outflow channel deposits and the regions that the authors are defining as coastal shelf. The authors have still not explained why the topographic signature they are seeing here is not because of the outflow channels.

We thank the reviewer for emphasizing the importance of Hesperian–Amazonian outflow channels and their impact on the topography of Chryse Planitia. We fully agree that these megafloods likely contributed to locally flattening the surface there, although it is difficult to quantify their exact contribution or determine whether their role is major or minor, especially given that similarly flat, low-slope surfaces are observed in other key regions (for example, Aeolis Dorsa) that do not record large outflow floods. In the revised manuscript, we now explicitly include outflow channels among the short-wavelength geomorphic processes that can smooth, incise or infill the margin and thereby broaden and mute the slope–curvature minima without eliminating an underlying shelf–slope break (please see the main text and below). We also added a dedicated limitation paragraph that specifically addresses outflow-channel erosion and deposition along the dichotomy boundary, noting that these floods likely flattened the surface in Chryse Planitia but that comparable low-slope surfaces occur elsewhere (for example, Aeolis Dorsa and other segments of the proposed shelf), where deltaic deposits and shoreline-consistent elevations independently record sea-level change. Together, these revisions clarify that outflow channels are an important local modifier—particularly in Chryse Planitia—but are unlikely to be the primary cause of the broader, basin-wide shelf-like pattern we document across the northern lowlands.

Abstract

Planet-wide interpretations of shorelines suggest that Mars once hosted an early ocean covering one-third of its surface¹⁻⁹. However, the elevations of these shorelines deviate from an equipotential surface by several kilometers, challenging that interpretation^{3,7,10-12}. Here, we investigate whether a planet that once hosted an ocean should be expected to leave discernible shorelines. We show that on Earth, the most prominent topographic signature of a global ocean is not a shoreline. Rather, it is a band of low slope and curvature values that comprise coastal

plains and the continental shelf, with an elevation range of -410 to -15 m. When applying a similar analysis to the martian surface, we observe a comparably flat zone between approximately -1,800 m and -3,800 m elevation, potentially marking a partially preserved martian coastal shelf. While other processes, such as lava flows¹³, might explain flat regions locally, a coastal-shelf best explains the circumglobal band of flat topography, in addition to river delta deposits^{4,14-17}, coastal deposits¹⁸, thick sequences of layered rock^{19,20}, and aqueously altered minerals^{20,21} all observed within the putative coastal shelf zone. Our results support the presence of an ancient ocean on Mars, and indicate that topographic shelves rather than shorelines are better indicators of long-lived oceans.

Main text

“The same analysis of Mars' topography does not reveal a single distinct minimum in global slope and curvature, as observed on Earth (Fig. 2c, d; Fig. S3). Instead, two coupled minima in slope and curvature are observed on Mars (Extended Data Fig. 1), supported by a Kruskal–Wallis H-test ($p = 1.07 \times 10^{-6}$), indicating less than a 0.00011% probability that the differences in median slope values among the observed elevation ranges occurred by chance. The first minima lie between approximately -1800 m and -3800 m (slope = 0.31° , curvature = 0.00015, -0.00017), and the second between -3800 m and -5000 m (slope = 0.12° , curvature = 0.0009, -0.0009). The minima are less distinct than on Earth, but this might be expected, given that Mars' ocean, if it existed, has likely been dry for billions of years, with substantial topographic modification driven by wind erosion³⁹, meteor impacts⁴⁰, Hesperian-Amazonian age outflow channels⁴¹, and regolith creep⁴². These short-wavelength geomorphic processes can smooth³⁹⁻⁴², incise or infill the margin and thus broaden and mute the slope and curvature minima without eliminating an underlying shelf–slope break. Long-wavelength vertical motions associated with geophysical processes—such as isostatic or flexural adjustment to ocean loading and unloading, true polar wander or Tharsis-induced deformation¹¹⁻¹²—would act over scales much larger than the mapped shelf width, primarily tilting or gently warping the margin and changing absolute slopes, but are unlikely to remove a relative slope and curvature minimum between shelf and slope. The processes creating basin relief also differ between the two planets. On Mars, an impact crater is likely the primary cause for the northern lowlands⁴³, whereas on Earth, plate tectonics allowed for chemical differentiation between oceanic and continental crust^{44,45}. “

Limitation

“A fourth source of limitation is the erosion and sediment redistribution along the dichotomy boundary by Hesperian-aged outflow floods, which likely deposited substantial volumes of sediment along the northern dichotomy, particularly in Chryse Planitia. These outflow events probably contributed to locally flattening the surface there. However, similarly flat, low-slope surfaces are also present at other key sites, such as Aeolis Dorsa^{37,38}—rich in stacked deltaic deposits—and along the remaining segments of the proposed shelf. This broader distribution, together with independent evidence for sea-level changes recorded by deltaic deposits at Hypanis³⁶, suggests that although Hesperian-aged outflow floods helped flatten the surface in Chryse Planitia, they were likely not the primary cause of surface flattening across the northern lowlands.”